# CoT Information: Improved Sample Complexity under Chain-of-Thought Supervision

**Awni Altabaa**
Statistics & Data Science
Yale University
awni.altabaa@yale.edu

**Omar Montasser**
Statistics & Data Science
Yale University
omar.montasser@yale.edu

**John Lafferty**
Statistics & Data Science
Yale University
john.lafferty@yale.edu

## Abstract

Learning complex functions that involve multi-step reasoning poses a significant challenge for standard supervised learning from input-output examples. Chain-of-thought (CoT) supervision, which augments training data with intermediate reasoning steps to provide a richer learning signal, has driven recent advances in large language model reasoning. This paper develops a statistical theory of learning under CoT supervision. Central to the theory is the *CoT information*, which measures the additional discriminative power offered by the chain-of-thought for distinguishing hypotheses with different end-to-end behaviors. The main theoretical results demonstrate how CoT supervision can yield significantly faster learning rates compared to standard end-to-end supervision, with both upper bounds and information-theoretic lower bounds characterized by the CoT information.

## 1 Introduction

"Chain-of-thought" (CoT) reasoning has been a driving force behind recent advances in the capabilities of large language models. While chain-of-thought began as a prompting technique [1–3], CoT-supervised training is now an important component of the post-training pipeline for large language models, and has been found to be highly effective in recent empirical research [4–6].

This paper proposes new concepts in statistical learning theory that are aimed at gaining insight into chain-of-thought learning. Consider the following concrete example of chain-of-thought, to ground the theoretical framework to be developed. The input $x$ is the sequence "`Which is larger, 8.9 or 8.10?`" and the intended answer $y$ is "`8.9`". When asked to answer directly, earlier systems (e.g., GPT-4) might respond incorrectly [e.g., 7]. However, newer models trained with chain-of-thought supervision will typically first output a CoT $z$, represented as a sequence of tokens, enabling the model to arrive at the correct answer. For example, the CoT might be, "`Compare the numbers as decimals, with 8.9 written as 8.90 and where 8.10 already has two decimal places. Then compare them digit by digit. The numbers agree in the first digit. But 9 is larger than 1 in the tenths place, so 8.9 is larger than 8.10.`". At test time, the CoT $z$ serves as an explanation of the answer. During training, however, the CoT $z$ plays the role of a natural language description of the execution trace of an algorithm—a step-by-step procedure that is to be learned. In this way, CoT is used as a rich, additional supervised learning signal that goes beyond standard input-output ("end-to-end") supervision.

The focus of our theory is to describe how this additional information impacts the statistical complexity of CoT-supervised learning. A key contribution of the paper is to identify a quantity that we call the *chain-of-thought information*, denoted by $\mathcal{I}^{\mathrm{CoT}}_{\mathcal{D}, h_\star}(\varepsilon; \mathcal{H})$. As we show, the CoT information characterizes the statistical complexity of CoT-supervised learning and captures the additional discrimination power granted to the learning algorithm by observing the chain-of-thought. In particular, the CoT information governs how the end-to-end error of the learned algorithm scales with the number of CoT training examples. Specifically, in the standard setting of PAC learning for bi-

39th Conference on Neural Information Processing Systems (NeurIPS 2025).

nary classification, the sample complexity scales as $m = \mathcal{O}\left(d/\varepsilon\right)$ in the realizable setting, where $d$ describes the size or complexity of the hypothesis space (e.g., the VC dimension), and $\varepsilon$ is the target classification error. In contrast, under CoT supervision, we show that the sample complexity scales according to $m = \mathcal{O}(d/\mathcal{I}^{\mathrm{CoT}}_{\mathcal{D},h_\star}(\varepsilon; \mathcal{H}))$. A case where the chain-of-thought is highly informative will have $\mathcal{I}^{\mathrm{CoT}}_{\mathcal{D},h_\star}(\varepsilon; \mathcal{H}) \gg \varepsilon$, which translates into favorable sample complexity. By establishing information-theoretic lower bounds, it is shown that the CoT information thus provides a fundamental measure of the value of this type of non-classical supervision. Because the construction of CoT training sets can be a time-consuming and expensive process, the theoretical framework developed in this paper may ultimately be of practical relevance, contributing to a formal understanding and quantification of the *value* of chain-of-thought supervision.

The remainder of the paper is organized as follows. Section 2 introduces an abstract model of chain-of-thought supervised learning, together with formal definitions of the learning objective and the key notions of risk, which we will use in our investigation. In Section 3, we motivate and introduce the CoT information measure, establish its fundamental properties, and, as an initial pedagogical result, show that it can be used to capture the improved sample complexity of CoT-supervised learning in the setting of finite-cardinality hypothesis classes. Section 4 extends this analysis to infinite hypothesis classes, as well as to the agnostic setting where the data are not assumed to be generated by a member of the class. In Section 5, two types of information-theoretic lower bounds are established that, together with the upper bounds, lends further support to considering the CoT information as a fundamental characterization of the value of chain-of-thought supervision. Section 6 concludes with a summary of further extensions that are presented in the appendix, a discussion of related work, and directions for future research that are suggested by the results of this paper.

## 2    Preliminaries: A Model of Chain-of-Thought Supervised Learning

The standard statistical learning problem is formulated as the problem of selecting a distinguished member of a function class $\mathcal{F} : \mathcal{X} \to \mathcal{Y}$, mapping from an input space $\mathcal{X}$ to an output space $\mathcal{Y}$. The learner observes a dataset of input-output examples $\{(x_i, y_i)\}_{i \in [m]}$ and seeks to identify the ground truth function $f_\star \in \mathcal{F}$ (in the realizable setting) or compete with its closest approximation in $\mathcal{F}$ (in the agnostic setting). A learning algorithm in the standard ("end-to-end") setting is a mapping $\mathcal{A} : (\mathcal{X} \times \mathcal{Y})^* \to \mathcal{Y}^{\mathcal{X}}$ from input-output datasets to predictors.

When the target function class $\mathcal{F}$ is highly complex—such as functions representing multi-step reasoning processes—learning from input-output examples alone can be statistically intractable. To overcome this difficulty, a natural approach is to provide the learner with increased supervision through the step-by-step execution of the target function on the input. To formulate this, we assume that each example observed by the learner includes not only the input $x$ and output $y$, but also an auxiliary observation $z$ that represents information about the function's execution on $x$.

A *chain-of-thought (CoT) hypothesis class* $\mathcal{H}$ is a family of functions $h : \mathcal{X} \to \mathcal{Y} \times \mathcal{Z}$. For each $h \in \mathcal{H}$, an input $x \in \mathcal{X}$ yields $h(x) = (y, z)$, where $y \in \mathcal{Y}$ is the output and $z \in \mathcal{Z}$ is the corresponding CoT. We denote the components of $h$ returning only the output as its end-to-end restriction, $h^{\mathrm{e2e}} : \mathcal{X} \to \mathcal{Y}$, and the component returning only the CoT as its CoT restriction, $h^{\mathrm{CoT}} : \mathcal{X} \to \mathcal{Z}$. In the chain-of-thought learning setting, the learner observes a dataset $\{(x_i, y_i, z_i)\}_{i \in [m]}$ and seeks to learn the underlying end-to-end function. A *chain-of-thought learning algorithm* is a mapping $\mathcal{A} : (\mathcal{X} \times \mathcal{Y} \times \mathcal{Z})^* \to \mathcal{Y}^{\mathcal{X}}$ from CoT datasets to predictors.

A key example captured by this framework is autoregressive sequence models, generating the CoT sequentially as $z_t = f(x_1, ..., x_n, z_1, ..., z_{t-1})$ until a final output $y = f(x_1, ..., x_n, z_1, ..., z_{t(x)})$ is generated. In this case, the spaces $\mathcal{X}, \mathcal{Y}, \mathcal{Z}$ would correspond to spaces of variable-length sequences over some vocabulary. Sequence models like transformers [8] are an important way to *implement* such hypothesis classes in a way that allows for CoT supervision. However, the details of any such implementation are not important for our theoretical treatment.

***Types of risk.*** It will be crucial to distinguish between two notions of risk: End-to-end risk and the chain-of-thought risk. Let $\mathcal{D}$ be a distribution over $\mathcal{X}$. For a reference hypothesis $h_\star \in \mathcal{H}$ and a predictor $h \in (\mathcal{Y} \times \mathcal{Z})^{\mathcal{X}}$, we define these risks as follows:

$$\mathcal{R}^{\mathrm{e2e}}_{\mathcal{D}}(h) = \mathop{\mathbb{P}}_{x \sim \mathcal{D}}\left[h^{\mathrm{e2e}}(x) \neq h^{\mathrm{e2e}}_\star(x)\right], \quad \mathcal{R}^{\mathrm{CoT}}_{\mathcal{D}}(h) = \mathop{\mathbb{P}}_{x \sim \mathcal{D}}\left[h(x) \neq h_\star(x)\right].$$

Table 1: A comparison of analysis techniques for studying learning with chain-of-thought.

| Method | Hypothesis class | Sample complexity (realizable) |
|---|---|---|
| E2E supervision | Finite $\mathcal{H}$ | $\log|\mathcal{H}|/\varepsilon$ |
| | General $\mathcal{H}$ | $\mathrm{VC}(\mathcal{L}^{\mathrm{e2e}}(\mathcal{H}))/\varepsilon$ |
| bounding CoT risk | Finite $\mathcal{H}$ | $\log|\mathcal{H}|/\varepsilon$ |
| | General $\mathcal{H}$ | $\mathrm{VC}(\mathcal{L}^{\mathrm{CoT}}(\mathcal{H}))/\varepsilon$ |
| using CoT Information | Finite $\mathcal{H}$ | $\log|\mathcal{H}|/\mathcal{I}^{\mathrm{CoT}}_{\mathcal{D},h_\star}(\varepsilon;\mathcal{H})$ |
| | General $\mathcal{H}$ | $\mathrm{VC}(\mathcal{L}^{\mathrm{CoT}}(\mathcal{H}))/\mathcal{I}^{\mathrm{CoT}}_{\mathcal{D},h_\star}(\varepsilon;\mathcal{H})$ |

That is, $\mathcal{R}^{\mathrm{e2e}}_{\mathcal{D}}(h)$ is the probability that the predictor's end-to-end *output* is incorrect, whereas $\mathcal{R}^{\mathrm{CoT}}_{\mathcal{D}}(h)$ is the probability that *either* the output $h^{\mathrm{e2e}}(x)$ or the CoT $h^{\mathrm{CoT}}(x)$ disagrees with $h_\star$. A key characteristic of the chain-of-thought supervised learning setting is that the training objective is the CoT loss, whereas the testing evaluation metric is the end-to-end risk. This asymmetry has important information-theoretic implications, which are a main focus of this work.

## 2.1 Interlude: The problem of linking the end-to-end and chain-of-thought risks

Before introducing the CoT information measure and our main theoretical result, we first motivate a key aspect of the analysis, which is specific to the CoT setting. In CoT learning, the learner observes training examples $S = \{(x_i, y_i, z_i)\}_{i \in [m]}$ and seeks to identify the *input-output* relationship using information from *both* the output $y_i$ and CoT labels $z_i$. That is, although the CoT error is used as a signal during training, only errors in the final output $y$ are penalized at test time. Consequently, to derive sharp statistical rates, it is necessary to link the two risk functions precisely.

To explain, recall that standard statistical learning theory characterizes the statistical complexity of learning from input-output examples *without* chain-of-thought supervision. For example, focusing on the realizable case for clarity, standard results in PAC learning [e.g., 9] show that the sample complexity to obtain end-to-end error $\varepsilon$ scales as $d/\varepsilon$, where $d$ is a complexity measure such as log-cardinality or the VC dimension of the end-to-end loss class $\mathcal{L}^{\mathrm{e2e}}(\mathcal{H})$. Intuitively, the $\varepsilon$-dependence can be understood in terms of the amount of information per sample, as $\mathcal{O}(1/\varepsilon)$ samples are required to distinguish between two hypotheses whose outputs disagree on a subset of measure $\varepsilon$ in the input space. Matching information-theoretic lower bounds validate that these are the optimal learning rates for the standard E2E-supervised setting.

In the CoT-supervised setting, the learning algorithm potentially has access to more information by observing the CoT, and thus faster rates of convergence are expected. The theoretical challenge lies in capturing this added information as improved rates in the analysis. Standard learning theory results cannot be directly applied to the CoT setting due to the mismatch between the training objective and the evaluation metric. One approach to address this challenge, which is taken by Joshi et al. [10], is to side-step this asymmetry by noting that the end-to-end error is always upper bounded by the CoT error, with $\mathcal{R}^{\mathrm{e2e}}_{\mathcal{D}}(h) \leq \mathcal{R}^{\mathrm{CoT}}_{\mathcal{D}}(h)$, and to instead establish a guarantee on the CoT risk, allowing the use of standard results in learning theory. In particular, one can define the CoT loss class for the hypothesis class $\mathcal{H}$ as a function class over $\mathcal{X} \times \mathcal{Y} \times \mathcal{Z}$ according to

$$\mathcal{L}^{\mathrm{CoT}}(\mathcal{H}) = \left\{ \ell^{\mathrm{CoT}}_h : (x,y,z) \mapsto \mathbf{1}\{h(x) \neq (y,z)\} \mid h \in \mathcal{H} \right\}.$$

Then, appealing to standard results in PAC learning [e.g., Vapnik's "General Learning" framework 9], one can learn $\mathcal{H}$ to obtain a *CoT risk* of $\varepsilon$ with a sample complexity $m(\varepsilon) = \mathcal{O}(\mathrm{VC}(\mathcal{L}^{\mathrm{CoT}}(\mathcal{H}))/\varepsilon)$, which in turn guarantees that the end-to-end risk is also bounded by $\varepsilon$.

This method of analysis leads to a sample complexity with the same $1/\varepsilon$ rate that we see in the end-to-end supervision setting, despite the increased amount of information per sample. In particular, this does not imply improved sample complexity over standard end-to-end supervision in the case of finite-cardinality classes (c.f. Table 1). In the general case, improved sample complexity hinges on whether or not the inequality $\mathrm{VC}(\mathcal{L}^{\mathrm{CoT}}(\mathcal{H})) \ll \mathrm{VC}(\mathcal{L}^{\mathrm{e2e}}(\mathcal{H}))$ holds, which is *a priori* unclear, even if it is possible to construct artificial classes for which this holds [10]. This suboptimality stems from the fact that this approach does not distinguish between the two types of risk and does not explicitly measure the amount of information encoded in the chain-of-thought. As a consequence, this approach cannot achieve matching information-theoretic lower bounds. Moreover, it is unclear

whether it is meaningful to apply this type of analysis to the agnostic setting, where the distribution over input-output-CoT examples is not realizable by the CoT hypothesis class.

# 3 Key Idea: The CoT Information Measure

We now describe a new approach that explicitly accounts for the additional information provided in the CoT supervision for distinguishing between hypotheses with different end-to-end behaviors.

**Definition 1** (CoT information). *For a CoT hypothesis class $\mathcal{H} \subset (\mathcal{Y} \times \mathcal{Z})^{\mathcal{X}}$ and distribution $\mathcal{D}$ over $\mathcal{X}$, we define the CoT information measures as follows:*

$$\mathcal{I}_{\mathcal{D}}^{\mathrm{CoT}}(h_1, h_2) = -\log \mathbb{P}_{\boldsymbol{x} \sim \mathcal{D}} \left[ h_1^{\mathrm{CoT}}(\boldsymbol{x}) = h_2^{\mathrm{CoT}}(\boldsymbol{x}), h_1^{\mathrm{e2e}}(\boldsymbol{x}) = h_2^{\mathrm{e2e}}(\boldsymbol{x}) \right]$$

$$\mathcal{I}_{\mathcal{D},h_\star}^{\mathrm{CoT}}(\varepsilon; \mathcal{H}) = \inf_{h \in \Delta_{\mathcal{D}}^{\mathrm{e2e}}(\varepsilon; \mathcal{H}, h_\star)} \mathcal{I}_{\mathcal{D}}^{\mathrm{CoT}}(h_\star, h).$$

*where the infimum is over $\Delta_{\mathcal{D}}^{\mathrm{e2e}}(\varepsilon; \mathcal{H}, h_\star)$, the set of hypotheses that disagree with the end-to-end behavior (i.e., output) of $h_\star$ with probability at least $\varepsilon$,*

$$\Delta_{\mathcal{D}}^{\mathrm{e2e}}(\varepsilon; \mathcal{H}, h_\star) := \left\{ h \in \mathcal{H} : \mathbb{P}_{\boldsymbol{x} \sim \mathcal{D}} \left[ h_\star^{\mathrm{e2e}}(\boldsymbol{x}) \neq h^{\mathrm{e2e}}(\boldsymbol{x}) \right] > \varepsilon \right\}.$$

The relative CoT information between two hypotheses $\mathcal{I}_{\mathcal{D}}^{\mathrm{CoT}}(h_1, h_2)$ quantifies how effectively the observed CoT behavior distinguishes the two hypotheses. In particular, the probability $\mathbb{P}[h_1^{\mathrm{CoT}}(x) = h_2^{\mathrm{CoT}}(x), h_1^{\mathrm{e2e}}(x) = h_2^{\mathrm{e2e}}(x)] \in (0, 1)$ represents the proportion of inputs on which a pair of hypotheses have matching behavior on *both* the CoT and the end-to-end output, rendering them indistinguishable from these observations. The relative CoT information between a pair of hypotheses is the negative logarithm of this probability; thus, $\mathcal{I}_{\mathcal{D}}^{\mathrm{CoT}}(h_1, h_2)$ takes values in $[0, \infty)$.

The CoT information of a *hypothesis class* $\mathcal{H}$, relative to the reference hypothesis $h_\star$, is a function of the error level $\varepsilon$, denoted $\mathcal{I}_{\mathcal{D},h_\star}^{\mathrm{CoT}}(\varepsilon; \mathcal{H})$. It is defined as the minimal relative CoT information between $h_\star$ and every alternative hypothesis $h \in \Delta_{\mathcal{D}}^{\mathrm{e2e}}(\varepsilon; \mathcal{H}, h_\star)$ which disagrees with $h_\star$'s end-to-end output on more than an $\varepsilon$ fraction of the inputs. A large $\mathcal{I}_{\mathcal{D},h_\star}^{\mathrm{CoT}}(\varepsilon; \mathcal{H})$ thus ensures high distinguishability (via CoT) between $h_\star$ and any such "bad" alternative.

A primary message of this work is that the CoT information characterizes the $\varepsilon$-dependence of sample complexity in Chain-of-Thought supervised learning by quantifying the informativeness of CoT supervision. The CoT information can be much larger than $\varepsilon$, yielding rapid learning under CoT supervision. The intuition is that when two hypotheses differ in terms of their end-to-end behavior, even with small probability, they will typically differ in terms of their computational traces (i.e., CoT) with high probability. Consequently, CoT supervision allows these differing hypotheses to be distinguished far more rapidly than by observing input-output samples alone.

## 3.1 Properties of the CoT information

The following result outlines key properties of the CoT information measure. Among these, the property $\mathcal{I}_{\mathcal{D},h_\star}^{\mathrm{CoT}}(\varepsilon; \mathcal{H}) \geq \varepsilon$ is particularly important. As will be demonstrated, this implies that, in the realizable setting, CoT supervision is never detrimental, information-theoretically. The proof of these properties is given in Appendix A.

**Lemma 1.** *Let $\mathcal{H} \subset (\mathcal{Y} \times \mathcal{Z})^{\mathcal{X}}$ be a CoT hypothesis class. Then the CoT information $\mathcal{I}_{\mathcal{D},h_\star}^{\mathrm{CoT}}(\varepsilon; \mathcal{H})$ satisfies the following properties:*

1. $\mathcal{I}_{\mathcal{D},h_\star}^{\mathrm{CoT}}(\varepsilon; \mathcal{H}) \geq \varepsilon$.

2. $\mathcal{I}_{\mathcal{D},h_\star}^{\mathrm{CoT}}(\varepsilon; \mathcal{H})$ *is monotonically increasing in $\varepsilon$.*

3. $\mathcal{I}_{\mathcal{D},h_\star}^{\mathrm{CoT}}(\varepsilon; \mathcal{H})$ *is monotonically decreasing in $\mathcal{H}$ (under the subset relation).*

Before proceeding with bounding sample complexity in terms of CoT information, we note how the measure behaves in extreme boundary conditions. First, let us consider an example where the CoT annotations are entirely independent of the end-to-end behavior. In particular, consider a CoT hypothesis class with a product structure $\mathcal{H} = \mathcal{F}^{\mathrm{CoT}} \times \mathcal{F}^{\mathrm{e2e}}$, where $\mathcal{F}^{\mathrm{CoT}} \subset \mathcal{Z}^{\mathcal{X}}, \mathcal{F}^{\mathrm{e2e}} \subset \mathcal{Y}^{\mathcal{X}}$. In this case, we would expect no statistical advantage from observing the CoT—this is captured by the CoT information measure, which coincides with the "end-to-end information" in this case.

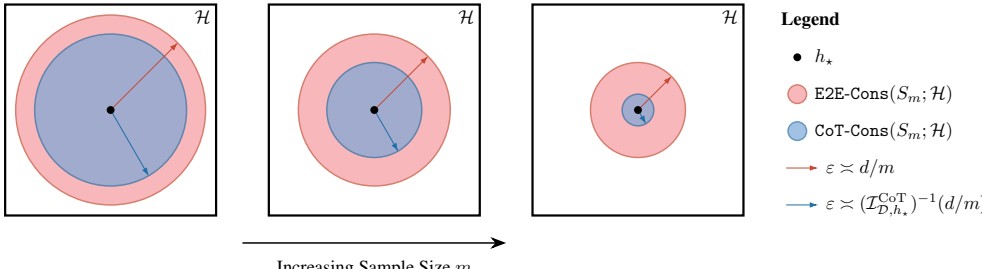

Increasing Sample Size $m$

Figure 1: Illustration of the statistical advantage of CoT supervision in terms of the geometry of the CoT consistency rule with respect to *end-to-end* error. CoT supervision enables the construction of a tighter consistency set when the CoT is informative (i.e., $\mathcal{I}_{\mathcal{D},h_\star}^{\mathrm{CoT}}(\varepsilon;\mathcal{H}) > \varepsilon$), which leads to smaller end-to-end error and more sample-efficient learning.

At the other extreme, consider the case where the CoT from *any* single example reveals the entire target function. For example, let $\mathcal{F} \subset \mathcal{Y}^{\mathcal{X}}$ and consider the CoT hypothesis class $\mathcal{H} = \{h_f : x \mapsto (f, f(x)) : f \in \mathcal{F}\}$. In this case, $\mathcal{I}_{\mathcal{D},h_\star}^{\mathrm{CoT}}(\varepsilon;\mathcal{H}) = \infty$, which corresponds to the fact that a single example is sufficient to attain zero error. Finally, consider the problem of learning a regular language with CoT supervision. Here, we take the output $y$ to indicate whether or not the string $x$ is in the language, but we let $z$ be the sequence of states visited in a DFA representing the language as it processes $x$; see Figure 2. Appendix B provides a more detailed discussion of these examples.

### 3.2 Improved sample complexity via CoT information

To illustrate the main ideas and intuitions underpinning this paper's results, we next prove a sample complexity bound for CoT-supervised learning with *finite* hypothesis classes in the *realizable* setting. While other proofs are deferred to the appendix for clarity and brevity, this particular result is proven here due to its simplicity and pedagogical value.

The learning rule we consider is *chain-of-thought consistency*, $\mathtt{CoT\text{-}Cons}(S;\mathcal{H})$: given a sample $S$, the learner returns any hypothesis in $\mathcal{H}$ which is consistent with the sample $S = \{(x_i, y_i, z_i)\}_{i \in [m]}$ in terms of both outputs and the chain-of-thought.

**Result 1** (Learning with Chain-of-Thought Supervision). *Let $\mathcal{H} \subset (\mathcal{Y} \times \mathcal{Z})^{\mathcal{X}}$ be a finite CoT hypothesis class. For any distribution $\mathcal{D}$ over $\mathcal{X} \times \mathcal{Y} \times \mathcal{Z}$ realized by some $h_\star \in \mathcal{H}$, the CoT consistency learning rule has a sample complexity of*

$$m(\varepsilon,\delta) = \frac{\log|\mathcal{H}| + \log(1/\delta)}{\mathcal{I}_{\mathcal{D},h_\star}^{\mathrm{CoT}}(\varepsilon;\mathcal{H})}.$$

*That is, for any $m \geq m(\varepsilon,\delta)$, with probability at least $1 - \delta$ over $S \sim \mathcal{D}^m$, any hypothesis $h$ that is CoT consistent on $S$ will have end-to-end risk satisfying $\mathcal{R}_{\mathcal{D}}^{\mathrm{e2e}}(h) \leq \varepsilon$.*

*Proof.* We aim to bound the probability of the "bad event"

$$\{\exists h \in \mathcal{H} : \mathcal{R}_{\mathcal{D}}^{\mathrm{e2e}}(h) > \varepsilon, \widehat{\mathcal{R}}_S^{\mathrm{CoT}}(h) = 0\}$$

over the draw of $(x_1, \ldots, x_m) \overset{\mathrm{i.i.d.}}{\sim} \mathcal{D}$. We highlight that the training loss is the empirical *CoT* risk, $\widehat{\mathcal{R}}_S^{\mathrm{CoT}}(h)$, whereas the test metric is the *end-to-end* risk $\mathcal{R}_{\mathcal{D}}^{\mathrm{e2e}}(h)$.

Fix any $h \in \mathcal{H}$ with end-to-end error larger than $\varepsilon$, $\mathcal{R}_{\mathcal{D}}^{\mathrm{e2e}}(h) = \mathbb{P}_{x \sim \mathcal{D}}[h^{\mathrm{e2e}}(x) \neq \mathrm{e2e}(h_\star)(x)] > \varepsilon$ (i.e., $h \in \Delta_{\mathcal{D}}^{\mathrm{e2e}}(\varepsilon;\mathcal{H},h_\star)$). We bound the probability that $h$ is CoT consistent on $S$, $h \in \mathtt{CoT\text{-}Cons}(S;\mathcal{H}) = \{h \in \mathcal{H} : \widehat{\mathcal{R}}_S^{\mathrm{CoT}}(h) = 0\}$, as follows

$$\begin{aligned}
\mathbb{P}_{S \sim \mathcal{D}^{\otimes m}}[h \in \mathtt{CoT\text{-}Cons}(S;\mathcal{H})] &= \mathbb{P}_{S \sim \mathcal{D}^{\otimes m}}\left[\forall i, h^{\mathrm{CoT}}(x_i) = h_\star^{\mathrm{CoT}}(x_i), h^{\mathrm{e2e}}(x_i) = (^{\mathrm{e2e}}h_\star)(x_i)\right] \\
&= \mathbb{P}_{x \sim \mathcal{D}}\left[h^{\mathrm{CoT}}(x) = h_\star^{\mathrm{CoT}}(x), h^{\mathrm{e2e}}(x) = h_\star^{\mathrm{e2e}}(x)\right]^m \\
&\overset{(a)}{=} \left(\exp\left(-\mathcal{I}_{\mathcal{D}}^{\mathrm{CoT}}(h_\star, h)\right)\right)^m \\
&\overset{(b)}{\leq} \exp\left(-m \cdot \mathcal{I}_{\mathcal{D},h_\star}^{\mathrm{CoT}}(\varepsilon;\mathcal{H})\right),
\end{aligned}$$

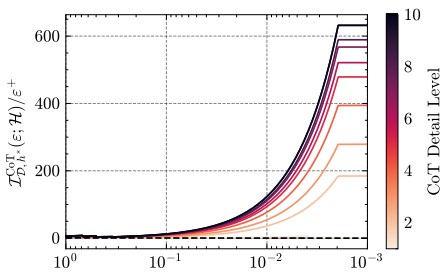 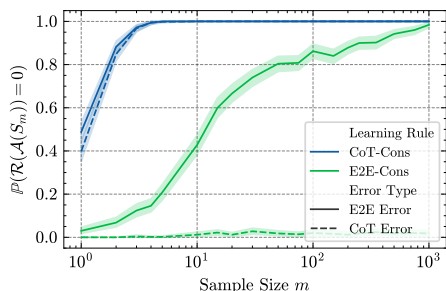

Figure 2: Simulation results for learning regular languages, where the CoT is the sequence of states $z$ visited by a DFA processing an input string $x$. **Right.** CoT supervision yields a $10^2$ to $10^3$-fold gain in sample efficiency. **Left.** This empirical gain aligns with theoretical predictions from the CoT information measure, which for this setup yields a limiting ratio $\lim_{\varepsilon \to 0} \mathcal{I}_{\mathcal{D}, h_\star}^{\text{CoT}}(\varepsilon; \mathcal{H})/\varepsilon \approx 600$.

where step (a) is by the definition of the relative CoT information between a pair of hypotheses, and step (b) is by the definition of $\mathcal{I}_{\mathcal{D}, h_\star}^{\text{CoT}}(\varepsilon; \mathcal{H})$ and the fact that $h \in \Delta_{\mathcal{D}}^{\text{e2e}}(\varepsilon; \mathcal{H}, h_\star)$.

Choosing $m = \frac{\log|\mathcal{H}| + \log(1/\delta)}{\mathcal{I}_{\mathcal{D}, h_\star}^{\text{CoT}}(\varepsilon; \mathcal{H})}$ implies that for each hypothesis $h \in \Delta_{\mathcal{D}}^{\text{e2e}}(\varepsilon; \mathcal{H}, h_\star)$ with end-to-end error larger than $\varepsilon$, the probability that it is in the CoT consistency set is bounded by

$$\mathbb{P}_{S \sim \mathcal{D}^{\otimes m}}[h \in \texttt{CoT-Cons}(S; \mathcal{H})] \leq \frac{\delta}{|\mathcal{H}|}.$$

Applying a union bound over $\mathcal{H}$ yields

$$\mathbb{P}_{S \sim \mathcal{D}^{\otimes m}}\left[\exists h \in \mathcal{H} : \mathcal{R}_{\mathcal{D}}^{\text{e2e}}(h) > \varepsilon, \widehat{\mathcal{R}}_S^{\text{CoT}}(h) = 0\right] \leq \delta$$

to complete the proof. $\qquad\square$

This result demonstrates that, for CoT learning, the $\varepsilon$-dependence of the sample complexity is $\mathcal{O}(1/\mathcal{I}_{\mathcal{D}, h_\star}^{\text{CoT}}(\varepsilon; \mathcal{H}))$, contrasting with the typical rate of $\mathcal{O}(1/\varepsilon)$. Intuitively, the ratio $\mathcal{I}_{\mathcal{D}, h_\star}^{\text{CoT}}(\varepsilon; \mathcal{H})/\varepsilon \geq 1$ quantifies the relative value of a CoT training example compared with an end-to-end training example. Figure 2 previews simulation results exploring CoT information in the context of learning a regular language, where the CoT is the sequence of states from the underlying deterministic finite automaton (DFA). Its left panel depicts the ratio $\mathcal{I}_{\mathcal{D}, h_\star}^{\text{CoT}}(\varepsilon; \mathcal{H})/\varepsilon$ as a function of $\varepsilon$. The plot can be interpreted as follows: For this hypothesis class $\mathcal{H}$ and distribution $\mathcal{D}$ (uniform), each CoT example is roughly 600 times more valuable than a single end-to-end example. The right panel presents empirical learning curves for the `CoT-Cons` and `E2E-Cons` rules, illustrating the statistical advantage of CoT supervision—an advantage theoretically captured by the CoT information measure. Further simulation results are presented in Appendix C.

## 4 Guarantees for CoT-Supervised Learning: Upper Bounds

This section extends our exploration of statistical upper bounds to infinite hypothesis classes and the agnostic learning setting, thereby further elucidating the statistical advantage of CoT supervision.

### 4.1 The realizable setting: Extension to infinite classes

Result 1 established a sample complexity bound determined by two key factors: the term $1/\mathcal{I}_{\mathcal{D}, h_\star}^{\text{CoT}}(\varepsilon; \mathcal{H})$, which captures the information per CoT-supervised sample, and the log-cardinality of the class, $\log|\mathcal{H}|$, which reflects its size or dimension. We now extend this result to infinite classes, replacing the log-cardinality term with the VC dimension of the CoT loss class. As before, the upper bound is achieved by the CoT consistency learning rule.

**Result 2** (Learning infinite classes under CoT supervision). *Let $\mathcal{H} \subset (\mathcal{Y} \times \mathcal{Z})^{\mathcal{X}}$ be a CoT hypothesis class. For any distribution $\mathcal{D}$ over $\mathcal{X} \times \mathcal{Y} \times \mathcal{Z}$ realized by some $h_\star \in \mathcal{H}$, the CoT consistency learning rule has a sample complexity of*

$$m(\varepsilon, \delta) = \mathcal{O}\left(\left(\frac{1}{\mathcal{I}_{\mathcal{D}, h_\star}^{\text{CoT}}(\varepsilon; \mathcal{H}, h_\star)} + 1\right)\left(\text{VC}(\mathcal{L}^{\text{CoT}}(\mathcal{H})) \cdot \log\left(\frac{1}{\mathcal{I}_{\mathcal{D}, h_\star}^{\text{CoT}}(\varepsilon; \mathcal{H})} + 1\right) + \log(1/\delta)\right)\right).$$

*That is, for any $m \geq m(\varepsilon, \delta)$, with probability at least $1 - \delta$ over $S \sim \mathcal{D}^m$, any hypothesis $h$ that is CoT consistent on $S$ will have end-to-end risk satisfying $\mathcal{R}_{\mathcal{D}}^{\text{e2e}}(h) \leq \varepsilon$.*

The proof is provided in Appendix D.1. The result follows from a lemma that relates the CoT risk of any *proper* CoT learning rule (i.e., one that returns a predictor in the hypothesis class) to its performance with respect to the end-to-end error.

## 4.2 The agnostic setting

The previous results assume that the data distribution $\mathcal{D}$ over $\mathcal{X} \times \mathcal{Y} \times \mathcal{Z}$ is *realizable* by the CoT hypothesis class $\mathcal{H}$. Such an assumption can be stringent, particularly in the presence of noise. This section, therefore, addresses the *agnostic* setting, where no restriction is made on the distribution; the goal, instead, is to compete with the best hypothesis in the class $\mathcal{H}$ in terms of *end-to-end risk*.

In the agnostic setting, a natural learning rule is *CoT empirical risk minimization*, which selects a hypothesis that minimizes the *empirical CoT risk*: $\texttt{CoT-ERM}(S; \mathcal{H}) = \arg\min_{h \in \mathcal{H}} \widehat{\mathcal{R}}_S^{\text{CoT}}(h)$.

Recall that CoT supervision never hurts in the realizable setting since $\mathcal{I}_{\mathcal{D}, h_\star}^{\text{CoT}}(\varepsilon; \mathcal{H}) \geq \varepsilon$ for any hypothesis class. The picture is more complicated in the agnostic setting. In particular, CoT supervision can be harmful or distracting, and discarding the CoT annotation and learning from only the input-output examples can be preferable, as the following example shows. The issue arises when the CoT hypothesis class $\mathcal{H}$ is not aligned with the data distribution, especially when $\mathcal{H}$ can fit the end-to-end behavior but not the CoT behavior.

**Example.** Consider a CoT hypothesis class $\mathcal{H} : \mathcal{X} \to \mathcal{Y} \times \mathcal{Z}$ and suppose $\mathcal{D}$ is a distribution over $\mathcal{X} \times \mathcal{Y} \times \mathcal{Z}$ for which the output component is realizable by $\mathcal{H}$ but the CoT component is not realizable. In particular, it is easy to construct examples for which $\inf_{h \in \mathcal{H}} \mathcal{R}_{\mathcal{D}}^{\text{e2e}}(h) = 0$ while $\inf_{h \in \mathcal{H}} \mathcal{R}_{\mathcal{D}}^{\text{CoT}}(h) = 1$. Clearly, in such cases, the CoT-ERM learning rule provides no guarantees whatsoever since $\texttt{CoT-ERM}(S; \mathcal{H}) = \mathcal{H}$ for any $S$ supported by $\mathcal{D}$. In contrast, E2E-ERM enjoys the standard PAC learning guarantees, with a sample complexity $\mathcal{O}\left(1/\varepsilon \cdot \text{VC}(\mathcal{L}^{\text{e2e}}(\mathcal{H}))\right)$.

Thus, CoT supervised learning in the agnostic setting requires a different notion of CoT information, which captures how well-aligned the data distribution is to the hypothesis class, and whether fitting the CoT aligns with fitting the end-to-end behavior. This uses an *excess* risk variant of the CoT information, defined in the following result, which extends our results to the agnostic setting.

**Result 3** (Agnostic learning under CoT supervision)**.** *Let $\mathcal{H} \subset (\mathcal{Y} \times \mathcal{Z})^{\mathcal{X}}$ be a CoT hypothesis class. For any distribution $\mathcal{D}$ over $\mathcal{X} \times \mathcal{Y} \times \mathcal{Z}$, the CoT-ERM learning rule has a sample complexity of*

$$m(\varepsilon, \delta) = \mathcal{O}\left(\frac{\text{VC}(\mathcal{L}^{\text{CoT}}(\mathcal{H})) + \log(1/\delta)}{\widetilde{\mathcal{I}}_{\mathcal{D}}^{\text{CoT}}(\varepsilon; \mathcal{H})^2}\right),$$

*where $\widetilde{\mathcal{I}}_{\mathcal{D}}^{\text{CoT}}(\varepsilon; \mathcal{H})$, the agnostic CoT information, is defined via excess risks as*

$$\widetilde{\mathcal{I}}_{\mathcal{D}}^{\text{CoT}}(\varepsilon; \mathcal{H}) := \inf\left\{\mathcal{E}_{\mathcal{D}}^{\text{CoT}}(h) \,:\, h \in \mathcal{H}, \mathcal{E}_{\mathcal{D}}^{\text{e2e}}(h) \geq \varepsilon\right\},$$

*where $\mathcal{E}_{\mathcal{D}}^{\text{e2e}}(h) := \mathcal{R}_{\mathcal{D}}^{\text{e2e}}(h) - \inf_h \mathcal{R}_{\mathcal{D}}^{\text{e2e}}(h)$ and $\mathcal{E}_{\mathcal{D}}^{\text{CoT}}(h) := \mathcal{R}_{\mathcal{D}}^{\text{CoT}}(h) - \inf_h \mathcal{R}_{\mathcal{D}}^{\text{CoT}}(h)$ are the excess CoT and end-to-end risks, respectively. That is, for any $m \geq m(\varepsilon, \delta)$, with probability at least $1 - \delta$ over the draw of $S \sim \mathcal{D}^m$, the excess end-to-end risk is bounded as $\mathcal{R}_{\mathcal{D}}^{\text{e2e}}(\hat{h}) \leq \inf_h \mathcal{R}_{\mathcal{D}}^{\text{e2e}}(h) + \varepsilon$, where $\hat{h} \in \texttt{CoT-ERM}(S; \mathcal{H})$.*

The proof is presented in Appendix D.2. Note that, unlike in the realizable case, we do not necessarily have the lower bound $\widetilde{\mathcal{I}}_{\mathcal{D}}^{\text{CoT}}(\varepsilon; \mathcal{H}) \geq \varepsilon$. For instance, in the motivating example above, $\widetilde{\mathcal{I}}_{\mathcal{D}}^{\text{CoT}}(\varepsilon; \mathcal{H}) = 0$. However, CoT supervision yields an advantage when the *excess* CoT risk dominates the excess end-to-end risk (i.e., $\mathcal{R}_{\mathcal{D}}^{\text{CoT}}(h) - \mathcal{R}_\star^{\text{CoT}} > \mathcal{R}_{\mathcal{D}}^{\text{e2e}}(h) - \mathcal{R}_\star^{\text{e2e}}$).

## 5 Information Theoretic Lower Bounds for CoT Supervised Learning

This section establishes information-theoretic lower bounds on sample complexity, further validating the CoT information $\mathcal{I}_{\mathcal{D}, h_\star}^{\text{CoT}}(\varepsilon; \mathcal{H})$ as a fundamental measure of statistical complexity for learning with CoT supervision. In general, the statistical complexity of a learning problem depends on several parameters, including the size or complexity of the hypothesis class (e.g., $\text{VC}(\mathcal{H})$ in binary classification) and the error parameter (e.g., $1/\varepsilon$ or $1/\varepsilon^2$ for the realizable and agnostic settings, respectively). Different types of lower bounds scale accordingly with one or both of these factors. Our main focus in this work is on the dependence of the sample complexity on the error parameter,

which corresponds to the amount of information encoded in the CoT supervision for discriminating between hypotheses with different end-to-end behavior.

We begin with a lower bound demonstrating that CoT information $\mathcal{I}_{\mathcal{D},h_\star}^{\mathrm{CoT}}(\varepsilon; \mathcal{H})$ characterizes the $\varepsilon$ dependence of sample complexity. The essence of the result is to lower bound the minimum number of samples required to reliably distinguish a pair of hypotheses with a given end-to-end disagreement, reducing the learning problem to a binary hypothesis testing problem [11], and relating the total variation distance between distributions over $\mathcal{X} \times \mathcal{Y} \times \mathcal{Z}$ induced by a pair of hypotheses to the relative CoT information between them.

**Result 4** (Lower bound via CoT information). *Let $\mathcal{H} \subset (\mathcal{Y} \times \mathcal{Z})^{\mathcal{X}}$ be a CoT hypothesis class and let $\mathcal{D}$ be a distribution on $\mathcal{X}$. Let $\boldsymbol{x}_1, \ldots, \boldsymbol{x}_m \sim \mathcal{D}$ be an i.i.d sample from $\mathcal{D}$. For any $h_\star \in \mathcal{H}$ and $\varepsilon > 0$, if the sample size satisfies*

$$m < \frac{\log(1/\delta)}{\mathcal{I}_{\mathcal{D},h_\star}^{\mathrm{CoT}}(\varepsilon; \mathcal{H})}$$

*then with probability at least $\delta$, there exists $h \in \mathcal{H}$ with end-to-end error at least $\varepsilon$ which is indistinguishable from $h_\star$ on the sample. Moreover, the expected error of any algorithm $\mathcal{A}$ satisfies*

$$\sup_{h_\star \in \mathcal{H}} \mathbb{E}_{S \sim P_{h_\star}^{\otimes m}} \left[ \mathcal{R}_{\mathcal{D},h_\star}^{\mathrm{e2e}}(\mathcal{A}(S)) \right] \geq \frac{1}{2} \sup_{\substack{h_\star \in \mathcal{H} \\ \varepsilon > 0}} \varepsilon \cdot \exp(-m \cdot \mathcal{I}_{\mathcal{D},h_\star}^{\mathrm{CoT}}(\varepsilon; \mathcal{H})).$$

This result validates the CoT information as characterizing the $\varepsilon$-dependence of the rate. However, a weakness of two-point methods is that they do not scale with the size of the hypothesis space. The following result addresses this by reducing the learning problem to that of testing *multiple* hypotheses, using a packing of the hypothesis space with respect to the *end-to-end error*. We then use Fano's inequality to lower bound the probability of error in terms of a mutual information, and relate this mutual information to the CoT information. To apply Fano's method in this way, we extend the framework to allow the observed $\boldsymbol{z}$ to be a stochastic function of the hypothesis CoT.

**Result 5** (Lower bound via Fano's method). *Let $\mathcal{H} \subset (\mathcal{Y} \times \mathcal{Z})^{\mathcal{X}}$ be a CoT hypothesis class and let $\mathcal{D}$ be a distribution over $\mathcal{X}$. Suppose that $x_1, \ldots, x_m \sim \mathcal{D}$. Let $Q \in \mathcal{P}(\mathcal{Y} \times \mathcal{Z} \mid \mathcal{Y} \times \mathcal{Z})$ be a noisy channel from $h(x) = (y, z)$ to observations $\bar{y}, \bar{z}$. Let $C_Q = \max_{a,b} D_{\mathrm{KL}}(Q(\cdot \mid a) \| Q(\cdot \mid b))$ be the capacity factor of the channel. The learner observes the noisy sample $S = \{(x_i, \bar{y}_i, \bar{z}_i)\}_{i=1}^m$. Define the pseudo-metric $d_{\mathcal{D}}^{\mathrm{e2e}}(h_1, h_2) = \mathbb{P}_x[h_1^{\mathrm{e2e}}(x) \neq h_2^{\mathrm{e2e}}(x)]$, and let $M(\varepsilon; \mathcal{H}, d_{\mathcal{D}}^{\mathrm{e2e}})$ be the $\varepsilon$-packing number of $\mathcal{H}$ with respect to this pseudo-metric. Then, for any algorithm $\mathcal{A}$, we have that*

$$m \leq \frac{\log M(\mathcal{H}, d_{\mathcal{D}}^{\mathrm{e2e}}, \varepsilon)}{2 \cdot \left( C_Q \cdot \sup_\pi \mathbb{E}_{h_1, h_2 \sim \pi} \left[ \mathcal{I}_{\mathcal{D}}^{\mathrm{CoT}}(h_1, h_2) \right] + \log 2 \right)},$$

*implies large error for some $h_\star \in \mathcal{H}$ with high probability, i.e.,*

$$\sup_{h_\star \in \mathcal{H}} \mathbb{P}_{S \sim P_{h_\star}^{\otimes m}} \left[ \mathcal{R}_{\mathcal{D},h_\star}^{\mathrm{e2e}}(\mathcal{A}(S)) \geq \frac{\varepsilon}{2} \right] \geq 1/2.$$

Here, $C_Q$ is a bound on the capacity of the channel that adds noise to the chain-of-thought. This lower bound relates the probability of large error to the CoT information measure, like the previous result, but also scales with the size of the hypothesis space, as measured by its packing number. Additionally, the result also models noise in the learning process by observing the CoT through a noisy channel. The proofs of both results, along with further discussion, are presented in Appendix E.

# 6 Discussion and Related Work

## 6.1 Further explorations

We describe additional results not included in the main paper, and defer to the appendix for details.

***Learning with mixed CoT and E2E supervision.*** In practice, obtaining CoT-annotated examples can be a costly and labor-intensive process, limiting their quantity, whereas input-output examples without CoT annotation can be relatively cheap and plentiful. This motivates a need for learning algorithms that can make use of both types of examples. Appendix F.1 studies learning from datasets with a mix of E2E and CoT supervision.

***CoT learning with inductive priors.*** Encoding prior knowledge about solution structure is critical for learning complex functions, such as those representing multi-step reasoning processes, particularly from limited data. Appendix F.2 explores chain-of-thought learning with inductive priors.

***Transfer learning and out-of-distribution generalization.*** Chain-of-thought supervision has significant implications for out-of-distribution generalization, as it guides the learning algorithm toward solutions that exhibit the correct step-by-step reasoning, potentially enabling robust generalization to novel input instances. In Appendix F.3, we define a variant of the CoT Information measure, $\mathcal{I}_{\mathcal{D}_{tr} \rightarrow \mathcal{D}_{test}}^{CoT}(\varepsilon; \mathcal{H})$, that captures transfer learning under CoT supervision. We also present a result on learning under CoT supervision with distribution shift, supported by experimental simulations.

## 6.2 Related work

Early usage of the term "chain-of-thought" referred to empirical prompting techniques that conditioned large language models to generate a series of intermediate reasoning steps before returning the final answer [1–3, 12]. Such prompting often employs in-context learning, where CoT examples are provided within the model's context before it processes the input [3]. Today, the term *chain-of-thought* takes a broader meaning, as it now comprises a core component of the training of large language models [4–6].

Several works have sought to theoretically understand the advantages of the chain-of-thought paradigm by analyzing the representational capacity of neural sequence models with and without chain-of-thought [13–16]. For example, Pérez et al. [13] show that Transformers can simulate Turing machines by generating CoT tokens, and Merrill and Sabharwal [14] extend this analysis by providing a more refined characterization of function classes in terms of the number of CoT steps.

While these studies demonstrate the *existence* of neural network models capable of computing a given function via a specific chain-of-thought, they do not address the statistical question of whether such models can be efficiently *learned* from data. Our work focuses on these statistical learning aspects, a direction also pursued by a few recent studies. For example, Malach [17] studies the problem of learning autoregressive next-token prediction on CoT datasets. The core idea of this work is to express a CoT function as a composition of $T$ (a fixed number) different sequence-domain functions, $\mathcal{H} = \mathcal{H}_1 \times \cdots \times \mathcal{H}_T$, where each $\mathcal{H}_t : \mathcal{X} \times \Sigma^{t-1} \rightarrow \Sigma$ maps the input and CoT generated so far $(\boldsymbol{x}, z_1, ..., z_{t-1})$ to the next CoT symbol $z_t$, with the $T$-th CoT symbol serving as the final output. With this formulation, Malach [17] proposes learning each $\mathcal{H}_t$ independently (with independent parameters), enabling the direct application of standard PAC results. While this approach simplifies the analysis, its assumption of independently learned functions at each iteration is a notable limitation, which does not accurately reflect real-world settings.

Building on this work, Joshi et al. [10] considers a time-invariant composition of sequence-domain functions, where the function at each iteration remains the same. Their analysis relies on bounding the *CoT error* using standard PAC learning tools based on the VC dimension of the CoT loss class, noting that the CoT error provides an upper bound on the end-to-end error (cf. Section 2.1 and the second row of Table 1). Moreover, Joshi et al. [10] construct a synthetic autoregressive class exhibiting a gap between the VC dimension of the end-to-end loss class and CoT loss class, implying a statistical advantage for CoT supervision. While our results also involve the CoT loss class, thus inheriting the advantages of such class complexity differences, the focus of our analysis is on the content of *information* per CoT-supervised sample. This is represented in the dependence of the sample complexity on the target error $\varepsilon$, captured by the CoT information measure. This provides a more complete description of the statistical advantage of CoT supervision in statistical learning. We contend that this information-theoretic analysis, centered on CoT information rather than solely on loss class complexity, identifies a more fundamental source of statistical advantage in CoT-supervised learning. This view is supported by our lower bound results and the close agreement between our theory and simulations.

We close the discussion of related work by noting that learning with chain-of-thought supervision can be framed as a *transfer learning* problem [e.g., 18–20], where the "source task" involves learning the mapping $x \mapsto (z, y)$ that jointly predicts the output $y$ and the auxiliary chain-of-thought signal $z$, while the "target task" is the *end-to-end* prediction task $x \mapsto y$. When the CoT information is large (i.e., $\mathcal{I}_{\mathcal{D}, h_\star}^{CoT}(\varepsilon) > \varepsilon$), the source distribution allows learning the target task more rapidly than the target distribution itself. This is sometimes referred to as *super transfer* [21].

### 6.3 Conclusion and future work

This work provides a theoretical analysis of learning with chain-of-thought (CoT) supervision, introducing the CoT information measure to characterize its statistical advantages via both upper and lower bounds. This opens several promising directions for future theoretical study of CoT learning.

The upper bounds obtained in this work are based on analyzing natural but relatively simple learning rules: CoT-consistency in the realizable setting, and CoT-ERM in the agnostic setting. Investigating the optimality of these algorithms and exploring the design of optimal learning strategies remain key open questions. This may be especially relevant in the agnostic setting, where the alignment between the data distribution and the CoT hypothesis class is critical. For instance, future work could explore adaptive learning rules that balance the optimization of the CoT error and end-to-end error to avoid over-optimizing the CoT when the hypothesis class is poorly aligned with the data. While we use the VC dimension of CoT loss class as the measure of complexity or size of the hypothesis class, it will also be important to consider other measures of model complexity, including covering numbers, local Radamacher complexities, and one-inclusion graphs [22–27].

Furthermore, while our current lower bound results address the realizable setting, establishing corresponding lower bounds for the agnostic setting remains an important open problem. Additionally, future research could investigate the formulation of different structural conditions, such as low-noise assumptions [28, 29] in the CoT setting, to achieve faster learning rates. Developing more sophisticated probabilistic analysis, beyond the standard formulation of agnostic learning, holds promise for more faithfully capturing the complexities of training language models with chain-of-thought reasoning traces, which are often inherently probabilistic.

## Acknowledgements

We thank Dana Angluin and Peter Bartlett for helpful comments on this work. This research was supported by the funds provided by the National Science Foundation and by DoD OUSD (R&E) under Cooperative Agreement PHY-2229929 (The NSF AI Institute for Artificial and Natural Intelligence).

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

# A  Proofs of Properties of the CoT Information ([Section 3.1](#))

**Lemma** (Restatement: Properties of the CoT-information). *Let $\mathcal{H} \subset (\mathcal{Y} \times \mathcal{Z})^{\mathcal{X}}$ be a CoT hypothesis class.*

1. *The CoT information is always larger than the "end-to-end information":*
   *For any $h_1, h_2 \in \mathcal{H}$, $\mathcal{I}_{\mathcal{D}}^{\mathrm{CoT}}(h_1, h_2) \geq \mathbb{P}_{x \sim \mathcal{D}}[h_1^{\mathrm{e2e}}(x) \neq h_2^{\mathrm{e2e}}(x)]$. Moreover,*
   $\mathcal{I}_{\mathcal{D}, h_\star}^{\mathrm{CoT}}(\varepsilon; \mathcal{H}) \geq \varepsilon$, $\forall \varepsilon \in [0, 1], \forall h_\star \in \mathcal{H}$.

2. $\mathcal{I}_{\mathcal{D}, h_\star}^{\mathrm{CoT}}(\cdot; \mathcal{H})$ *is monotonically increasing in $\varepsilon$:*
   *For any $\mathcal{H}, h_\star \in \mathcal{H}$, and $\varepsilon_1 \leq \varepsilon_2$, $\mathcal{I}_{\mathcal{D}, h_\star}^{\mathrm{CoT}}(\varepsilon_1; \mathcal{H}) \leq \mathcal{I}_{\mathcal{D}, h_\star}^{\mathrm{CoT}}(\varepsilon_2; \mathcal{H})$.*

3. $\mathcal{I}_{\mathcal{D}, h_\star}^{\mathrm{CoT}}(\varepsilon; \cdot)$ *is monotonically decreasing in the hypothesis class:*
   *For any hypothesis classes $\mathcal{H} \subseteq \mathcal{H}'$ and $h_\star \in \mathcal{H}$, $\mathcal{I}_{\mathcal{D}, h_\star}^{\mathrm{CoT}}(\varepsilon; \mathcal{H}) \geq \mathcal{I}_{\mathcal{D}, h_\star}^{\mathrm{CoT}}(\varepsilon; \mathcal{H}')$.*

*Proof.*

*Property 1.* To prove the first claim, take any $h_1, h_2 \in \mathcal{H}$, and observe that

$$
\begin{aligned}
\mathcal{I}_{\mathcal{D}}^{\mathrm{CoT}}(h_1, h_2) &:= -\log \mathbb{P}_{\boldsymbol{x} \sim \mathcal{D}} \left[ h_1^{\mathrm{CoT}}(\boldsymbol{x}) = h_2^{\mathrm{CoT}}(\boldsymbol{x}), h_1^{\mathrm{e2e}}(\boldsymbol{x}) = h_2^{\mathrm{e2e}}(\boldsymbol{x}) \right] \\
&\geq -\log \mathbb{P}_{\boldsymbol{x} \sim \mathcal{D}} \left[ h_1^{\mathrm{e2e}}(\boldsymbol{x}) = h_2^{\mathrm{e2e}}(\boldsymbol{x}) \right] \\
&= -\log \left( 1 - \mathbb{P}_{\boldsymbol{x} \sim \mathcal{D}} \left[ h_1^{\mathrm{e2e}}(\boldsymbol{x}) \neq h_2^{\mathrm{e2e}}(\boldsymbol{x}) \right] \right) \\
&\geq \mathbb{P}_{\boldsymbol{x} \sim \mathcal{D}} \left[ h_1^{\mathrm{e2e}}(\boldsymbol{x}) \neq h_2^{\mathrm{e2e}}(\boldsymbol{x}) \right],
\end{aligned}
$$

where the final inequality is by the identity $-\log(1 - x) \geq x$.

We now show the second claim,

$$
\begin{aligned}
\mathcal{I}_{\mathcal{D}, h_\star}^{\mathrm{CoT}}(\varepsilon; \mathcal{H}) &:= \min_{h \in \Delta_{\mathcal{D}}^{\mathrm{e2e}}(\varepsilon; \mathcal{H}, h_\star)} \mathcal{I}_{\mathcal{D}}^{\mathrm{CoT}}(h_\star, h) \\
&\geq \min_{h \in \Delta_{\mathcal{D}}^{\mathrm{e2e}}(\varepsilon; \mathcal{H}, h_\star)} \mathbb{P}_{\boldsymbol{x} \sim \mathcal{D}} \left[ h^{\mathrm{e2e}}(\boldsymbol{x}) \neq h_\star^{\mathrm{e2e}}(\boldsymbol{x}) \right] \\
&\geq \varepsilon
\end{aligned}
$$

The final inequality follows because $\mathbb{P}_{\boldsymbol{x} \sim \mathcal{D}} \left[ h^{\mathrm{e2e}}(\boldsymbol{x}) \neq h_\star^{\mathrm{e2e}}(\boldsymbol{x}) \right] \geq \varepsilon$, $\forall h \in \Delta_{\mathcal{D}}^{\mathrm{e2e}}(\varepsilon; \mathcal{H}, h_\star)$, by definition.

*Property 2.* This follows from the fact that $\Delta_{\mathcal{D}}^{\mathrm{e2e}}(\varepsilon; \mathcal{H}, h_\star) := \{h \in \mathcal{H} : \mathbb{P}[h^{\mathrm{e2e}}(\boldsymbol{x}) \neq \widetilde{h}^{\mathrm{e2e}}(\boldsymbol{x})] > \varepsilon\}$ is decreasing in $\varepsilon$. For $\varepsilon_1 \leq \varepsilon_2$, we have $\Delta_{\mathcal{D}}^{\mathrm{e2e}}(\varepsilon_1; \mathcal{H}, h_\star) \supseteq \Delta_{\mathcal{D}}^{\mathrm{e2e}}(\varepsilon_2; \mathcal{H}, h_\star)$, and hence

$$
\mathcal{I}_{\mathcal{D}, h_\star}^{\mathrm{CoT}}(\varepsilon_1; \mathcal{H}) := \min_{h \in \Delta_{\mathcal{D}}^{\mathrm{e2e}}(\varepsilon_1; \mathcal{H}, h_\star)} \mathcal{I}_{\mathcal{D}}^{\mathrm{CoT}}(h, h_\star) \leq \min_{h \in \Delta_{\mathcal{D}}^{\mathrm{e2e}}(\varepsilon_2; \mathcal{H}, h_\star)} \mathcal{I}_{\mathcal{D}}^{\mathrm{CoT}}(h, h_\star) =: \mathcal{I}_{\mathcal{D}, h_\star}^{\mathrm{CoT}}(\varepsilon_2; \mathcal{H}).
$$

*Property 3.* This property similarly follows from the fact that $\Delta_{\mathcal{D}}^{\mathrm{e2e}}(\varepsilon; \mathcal{H}, h_\star)$ is increasing in $\mathcal{H}$: $\Delta_{\mathcal{D}}^{\mathrm{e2e}}(\varepsilon; \mathcal{H}, h_\star) \subset \Delta_{\mathcal{D}}^{\mathrm{e2e}}(\varepsilon; \mathcal{H}', h_\star)$ for $\mathcal{H} \subseteq \mathcal{H}'$. Thus,

$$
\mathcal{I}_{\mathcal{D}, h_\star}^{\mathrm{CoT}}(\varepsilon; \mathcal{H}) := \min_{h \in \Delta_{\mathcal{D}}^{\mathrm{e2e}}(\varepsilon; \mathcal{H}, h_\star)} \mathcal{I}_{\mathcal{D}}^{\mathrm{CoT}}(h, h_\star) \geq \min_{h \in \Delta_{\mathcal{D}}^{\mathrm{e2e}}(\varepsilon; \mathcal{H}', h_\star)} \mathcal{I}_{\mathcal{D}}^{\mathrm{CoT}}(h, h_\star) =: \mathcal{I}_{\mathcal{D}, h_\star}^{\mathrm{CoT}}(\varepsilon; \mathcal{H}').
$$

$\square$

# B  Simple Examples of CoT Hypothesis classes and their CoT Information

In this section, we provide a more detailed discussion on the illustrative examples presented in [Section 2](#). The first two examples represent the two extremes on the informativeness of CoT supervision and serve as sanity checks to confirm that the CoT information captures the expected statistical complexity in each case. The next example considers a hypothesis class where the CoT supervision includes $T$ independent samples of the input-output function, and shows that the CoT information scales linearly with $T$ as expected. Finally, we consider CoT hypothesis classes based on models of computation such as finite-state machines, where the CoT is taken to be the state trajectory of the computational process.

**Example 1** (Uninformative CoT yields small CoT information). In some cases, the CoT annotations may be entirely "independent" from the end-to-end behavior, and hence uninformative for the purposes of learning with respect to the end-to-end error. We will see that the CoT information $\mathcal{I}_{\mathcal{D},h_\star}^{\mathrm{CoT}}(\varepsilon;\mathcal{H})$ captures this. We will model the "independence" between the CoT and the end-to-end behavior via a hypothesis class with a *product structure*. Let $\mathcal{F}^{\mathrm{e2e}} \subset \mathcal{Y}^{\mathcal{X}}$ be a function class from inputs $\mathcal{X}$ to outputs $\mathcal{Y}$ and let $\mathcal{F}^{\mathrm{CoT}} \subset \mathcal{Z}^{\mathcal{X}}$ be a function class from inputs $\mathcal{X}$ to the CoT space $\mathcal{Z}$. We consider a CoT hypothesis class $\mathcal{H} = \mathcal{F}^{\mathrm{CoT}} \times \mathcal{F}^{\mathrm{e2e}}$. where

$$\mathcal{H} = \{h_{g,f} : x \mapsto (g(x), f(x)) \mid g \in \mathcal{F}^{\mathrm{CoT}}, f \in \mathcal{F}^{\mathrm{e2e}}\}.$$

Let $h_\star = (g_\star, f_\star) \in \mathcal{H} = \mathcal{F}^{\mathrm{CoT}} \times \mathcal{F}^{\mathrm{e2e}}$. Let $\bar{f} \in \mathcal{F}^{\mathrm{e2e}}$ be the end-to-end hypothesis with smallest disagreement with $f_\star$ among hypothesis with end-to-end error at least $\varepsilon$:

$$\bar{f} = \arg\min_{f \in \mathcal{F}^{\mathrm{e2e}}} \left\{ \mathbb{P}_{x \sim \mathcal{D}} [f_\star(x) \neq f(x)] \; : \; \mathbb{P}_{x \sim \mathcal{D}} [f_\star(x) \neq f(x)] > \varepsilon \right\}.$$

Let $\varepsilon^+ := \min\{\mathbb{P}[f_\star(x) \neq f(x)] \; : \; \mathbb{P}[x \sim \mathcal{D}]f_\star(x) \neq f(x) > \varepsilon\}$. By the product-structure definition of $\mathcal{H}$, there exists a hypothesis $\bar{h} := (g_\star, \bar{f}) \in \Delta_{\mathcal{D}}^{\mathrm{e2e}}(\varepsilon;\mathcal{H})$ such that $\mathbb{P}[h_\star(x) \neq \bar{h}(x)] = \mathbb{P}[h_\star^{\mathrm{e2e}}(x) \neq \bar{h}^{\mathrm{e2e}}(x)] = \varepsilon^+$, and hence $\mathcal{I}_{\mathcal{D},h_\star}^{\mathrm{CoT}}(\varepsilon;\mathcal{H},h_\star) = -\log(1-\varepsilon^+)$. Thus, there is no statistical advantage in observing the CoT annotations.

**Example 2** (Fully Informative CoT yields infinite CoT information). Recall the definition of the CoT information as

$$\mathcal{I}_{\mathcal{D},h_\star}^{\mathrm{CoT}}(\varepsilon;\mathcal{H}) := \inf \left\{ -\log \mathbb{P}_x [h(x) = h_\star(x)] : h \in \Delta_{\mathcal{D}}^{\mathrm{e2e}}(\varepsilon;\mathcal{H},h_\star) \right\}.$$

This can be infinite when $\forall h \in \Delta_{\mathcal{D}}^{\mathrm{e2e}}(\varepsilon;\mathcal{H},h_\star)$, $\mathbb{P}_x[h(x) = h_\star(x)] = 0$. This occurs in the extreme case where a single CoT annotation uniquely identifies the end-to-end behavior of the hypothesis (i.e., on every input in the support of $\mathcal{D}$, each hypothesis has a unique CoT). To illustrate this, let $\mathcal{F}^{\mathrm{e2e}} \subset \mathcal{Y}^{\mathcal{X}}$ be a class of functions from the input space $\mathcal{X}$ to the output space $\mathcal{Y}$. Consider the CoT hypothesis class $\mathcal{H} = \{h_f : x \mapsto (f, f(x)) \; : \; f \in \mathcal{F}^{\mathrm{e2e}}\}$. In this extreme example, a single sample is enough to learn the function perfectly. This is captured by the CoT information since $\forall h_1 \neq h_2 \in \mathcal{H}$, we have $\mathbb{P}[h_1(x) = h_2(x)] = 0$ and hence $\mathcal{I}_{\mathcal{D},h_\star}^{\mathrm{CoT}}(\varepsilon;\mathcal{H}) = \infty$.

**Example 3** (CoT Information captures i.i.d. examples in CoT). In this example, we consider a setting where the chain-of-thought represents i.i.d. observations from the end-to-end function, as a toy model that allows us to vary the informativeness of the CoT supervision for a fixed end-to-end function class. We will confirm that the CoT information implies the sample complexity rates that we would expect. Consider the CoT hypothesis class $\mathcal{H}^{(T)}$ where the CoT encodes $T$ independent observations, defined as follows:

$$\mathcal{H}^{(T)} := \left\{ h_f^{(T)} : (x_1, \ldots, x_T) \mapsto (\underbrace{(f(x_1), \ldots, f(x_T))}_{\boldsymbol{z}=(z_1,\ldots,z_T)}, \underbrace{f(x_T)}_{y}) \; \middle| \; f \in \mathcal{F} \right\}.$$

Here, $\mathcal{F}$ is a function class from $\bar{\mathcal{X}}$ to $\bar{\mathcal{Y}}$, and $\mathcal{X} = \bar{\mathcal{X}}^T, \mathcal{Z} = \bar{\mathcal{Y}}^T, \mathcal{Y} = \bar{\mathcal{Y}}$. Let $\mathcal{D} = \bar{\mathcal{D}}^{\otimes T}$ for some distribution $\bar{\mathcal{D}}$ over $\bar{\mathcal{X}}$. Fix $h_\star = h_{f_\star}$ and let $h = h_f \in \Delta_{\mathcal{D}}^{\mathrm{e2e}}(\varepsilon;\mathcal{H},h_\star)$. We have

$$\mathcal{I}_{\mathcal{D}}^{\mathrm{CoT}}(h_\star, h) = -\log \mathbb{P}_{x \sim \mathcal{D}} [h_\star(x) = h(x)] = -\log \mathbb{P}_{x_1, \ldots, x_T \overset{\mathrm{i.i.d.}}{\sim} \bar{\mathcal{D}}} [\forall t \in [T] : f_\star(x_t) = f(x_t)]$$

$$\geq -\log(1-\varepsilon)^T = T \cdot (-\log(1-\varepsilon)) \geq T \cdot \varepsilon.$$

This in turn implies that $\mathcal{I}_{\mathcal{D},h_\star}^{\mathrm{CoT}}(\varepsilon;\mathcal{H}) \geq T \cdot \varepsilon$. That is, one CoT sample is worth $T$ end-to-end samples, and the CoT sample complexity is smaller by a factor of $T$. This is what we would expect for this example since a CoT example consists of $T$ independent samples.

**Example 4** (Learning Regular Languages with State-Trajectory CoT). Let $\mathcal{H}$ be the class of Finite-State Machines with common state space $\mathcal{S}$ and operating over an alphabet $\Sigma$. That is, $\mathcal{H} = \{h_\delta : \delta \in \mathcal{T}\}$, where $\mathcal{T}$ is the set of transition functions $\mathcal{T} = \mathcal{S}^{\mathcal{S} \times \Sigma}$. The Chain-of-Thought observed by the learner is the sequence of states visited by the DFA during its execution: for an input $x = (x_1, ..., x_n) \in \Sigma^n$, the CoT of $h_\delta$ is $\boldsymbol{z} = (z_1, ..., z_n)$, where $z_{t+1} = \delta(z_t, x_t)$. Observing the CoT can be interpreted as providing the learner with an input-dependent partial observation of the

DFA's underlying transition function. Once the learner has identified all components of the transition function (or all components that are necessary to specify the input-output behavior), the learning objective is achieved. We can use this interpretation to lower-bound the CoT information. Let $\Delta(h_1, h_2) = \{(s, x) \in \mathcal{S} \times \Sigma : \delta_1(s, x) \neq \delta_2(s, x)\}$ be the set of state-symbol pairs on which $h_1$ and $h_2$'s transition functions differ. Then, we have

$$1 - \mathop{\mathbb{P}}_{\boldsymbol{x} \sim \mathcal{D}} \left[ h_\star^{\mathrm{CoT}}(\boldsymbol{x}) = h^{\mathrm{CoT}}(\boldsymbol{x}), h_\star^{\mathrm{e2e}}(\boldsymbol{x}) = h^{\mathrm{e2e}}(\boldsymbol{x}) \right]$$
$$\geq \mathop{\mathbb{P}}_{\boldsymbol{x} \sim \mathcal{D}} \left[ h_\star^{\mathrm{CoT}}(\boldsymbol{x}) \neq h^{\mathrm{CoT}}(\boldsymbol{x}) \right]$$
$$= \mathop{\mathbb{P}}_{\boldsymbol{x} \sim \mathcal{D}} \left[ \exists t \in [n] : z_t \neq z_t^* \right]$$
$$= \mathop{\mathbb{P}}_{\boldsymbol{x} \sim \mathcal{D}} \left[ h_\star \text{ visits any } (s, a) \in \Delta(h, h_\star) \right]$$

Suppose that $h_\star$'s transition graph is $\ell$-connected in the sense that for every state $s \in \mathcal{S}$ which is reachable by some input supported by $\mathcal{D}$, $\exists \ell' \leq \ell, a_1, \ldots, a_{\ell'} \in \Sigma$ such that $s_{\ell'} = s$ where $s_{t+1} = \delta^*(s_t, a_t), s_1 = s_{init}$. Then, if e.g. $\mathcal{D} = \mathrm{Unif}(\Sigma^n), n \geq \ell$, the above calculation implies that for all $h \neq h_\star$,

$$\mathop{\mathbb{P}}_{\boldsymbol{x} \sim \mathcal{D}} \left[ h_\star^{\mathrm{CoT}}(\boldsymbol{x}) = h^{\mathrm{CoT}}(\boldsymbol{x}), h_\star^{\mathrm{e2e}}(\boldsymbol{x}) = h^{\mathrm{e2e}}(\boldsymbol{x}) \right] \leq 1 - |\Sigma|^{-(\ell+1)}.$$

Thus, the CoT information is lower bounded as

$$\min_{\varepsilon > 0} \mathcal{I}_{\mathcal{D}, h_\star}^{\mathrm{CoT}}(\varepsilon; \mathcal{H}) \geq |\Sigma|^{-(\ell+1)}.$$

Note that this bound may be loose since it only counts a single trajectory that can be used to distinguish between the pair of hypotheses. But, its strength is that it lower bounds the CoT information at all error levels $\varepsilon$, and hence upper bounds the sample complexity of achieving *zero* error. A rich literature exists on learning regular languages [e.g. 30–35].

**Example 5** (Learning Shuffle Ideals by Observing Computational Trace of Finite State Machines)**.** The class of shuffle ideals is a simple subclass of regular languages that has been studied in the context of efficient PAC learning [36, 37]. For a string $u \in \Sigma^n$, the shuffle ideal generated by $u$ is the language $\Sigma^* u_1 \Sigma^* u_2 \Sigma^* \cdots \Sigma^* u_n \Sigma^*$ consisting of all strings which contain $u$ as a subsequence. The class of shuffle ideals of strings of length $n$ can be represented by finite state automata with $n + 1$ states. For a string $u \in \Sigma^n$, it's shuffle ideal is recognized by the finite state machine with the following transition function

$$\delta(s, a) = \begin{cases} s + 1, & \text{if } a = u_s \\ s & \text{otherwise.} \end{cases}$$

The acceptance state is $s = n + 1$. This finite state machine has a state space with a sequential structure, with each state "looking for" a particular symbol. When that symbol is observed, the state progresses to the next. A string is accepted if state $n + 1$ is reached, signifying that all symbols in the string $u$ are observed in the correct order. Due to the structure of this hypothesis class, each state has exactly one symbol that causes it to progress. Thus, to learn the FSA perfectly, it is enough to learn which symbol each state accepts. In fact, due to the sequential structure of this class of finite-state machines, this information is revealed on a single trajectory from a positive example. Thus, the CoT information can be bounded in terms of the probability of observing a positive example. In particular, for $h_\star \neq h \in \mathcal{H}$, we have

$$\mathop{\mathbb{P}}_{x \sim \mathcal{D}} \left[ h_\star^{\mathrm{CoT}}(x) \neq h^{\mathrm{CoT}}(x) \right] \geq \mathop{\mathbb{P}}_{x \sim \mathcal{D}} \left[ h_\star^{\mathrm{e2e}}(x) = \mathrm{accept} \right],$$

and hence $\min_{\varepsilon > 0} \mathcal{I}_{\mathcal{D}, h_\star}^{\mathrm{CoT}}(\varepsilon; \mathcal{H}) \geq \mathop{\mathbb{P}}_{x \sim \mathcal{D}} \left[ h_\star^{\mathrm{e2e}}(x) = \mathrm{accept} \right]$.

**Example 6** (Turing Machines)**.** It is possible to consider learning Turing Machines with CoT-supervision in a manner similar to Example 4. Recall that a Turing machine is specified by a transition function $\delta : \mathcal{S} \times \Sigma \to \mathcal{S} \times \Sigma \times \{\pm 1\}$, mapping the current state $s$ and observed symbol $\sigma$ on the current position in the tape to the next state $s'$, the symbol to be written $\gamma$, and the direction to move the tape $d$. We may consider Turing machines with chain-of-thought-supervision, where the CoT is the trajectory of states, written symbols, and tape movements: $\boldsymbol{z} = (\langle s_1, \gamma_1, d_1 \rangle, \ldots, \langle s_{t(x)}, \gamma_{t(x)}, d_{t(x)} \rangle)$, where $t(x)$ is the halting time of the Turing machine, and can depend on the input $x$ and the instance $h_\star$. Similar to the case of DFAs, observing the

CoT reveals an input-dependent partial specification of the underlying transition function. To lower bound the CoT information, one can consider analogous "connectivity" conditions to those discussed in Example 4. For example, one such condition is that for every $s \in \mathcal{S}$, there exists an input prefix $w \in \Sigma^{\leq \ell}$ such that the Turing machine $h_\star$ lands in state $s$ when reading $w$ and starting at $s_{init}$.

## C   Simulations

This section presents numerical simulations empirically exploring the CoT information measure for simple CoT hypothesis classes and its ability to predict sample complexity gains from CoT-supervised learning.

### C.1   Deterministic finite automata

A deterministic finite automaton (DFA), a type of finite state machine (FSM), is a foundational model of computation. A DFA defines a function from the space of variable-length strings to a binary classification: accept or reject. DFAs recognize exactly the class of regular languages [38].

CoT learning is often used as a means of providing supervision on the intermediate computation of a reference algorithm to be learned. In the following experiments, we use DFAs as a model of computation to study this type of supervision.

We consider a CoT hypothesis class $\mathcal{H} \subset (\mathcal{Y} \times \mathcal{Z})^{\mathcal{X}}$ based on deterministic finite automata (DFA). We take the input space to be $\mathcal{X} = \Sigma^n$ (or $\Sigma^*$) for some alphabet $\Sigma$. As with Example 4, the hypothesis class $\mathcal{H}$ is the set of all DFAs with state space $\mathcal{S}$ operating over the alphabet $\Sigma$. The output $y = h^{\text{e2e}}(\boldsymbol{x})$ is the acceptance or rejection of the string $\boldsymbol{x} \in \mathcal{X} = \Sigma^n$, and the chain-of-thought $\boldsymbol{z} = (z_1, ..., z_n) = h^{\text{CoT}}(\boldsymbol{x}) \in \mathcal{S}^n$ is the sequence of states the DFA visits during its execution.

Recall that a DFA is specified by a transition function $\delta : \mathcal{S} \times \Sigma \to \mathcal{S}$ that maps the current state and current symbol to the next step, an initial state $s_{\text{init}} \in \mathcal{S}$, and an acceptance state $s_{\text{accept}} \in \mathcal{S}$. The final output of the DFA is $\mathbf{1}\{z_n = s_{\text{accept}}\}$.

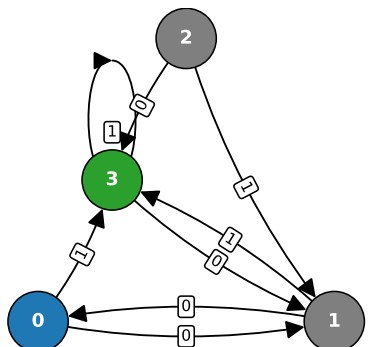

Figure 3: The transition graph of $h_\star$. This DFA is identified with the regular language that checks if the final symbol is a one or a zero.

In these simulations, we fix the size of the state space $\mathcal{S}$ and vocabulary $\Sigma$, as well as choose an initial state and acceptance state, and generate $\mathcal{H}$ as the set of all automata operating on those spaces. We place a uniform distribution over the input space $\mathcal{D} = \text{Unif}(\mathcal{X}) = \text{Unif}(\Sigma^n)$. We then choose $h_\star \sim \text{Unif}(\mathcal{H})$ randomly, plotted to the right, and numerically compute $\mathcal{I}^{\text{CoT}}_{\mathcal{D}}(h_\star, h)$ and $\mathcal{I}^{\text{CoT}}_{\mathcal{D},h_\star}(\varepsilon; \mathcal{H})$. Figure 4 depicts the simulation results for these DFA experiments.

**Value of CoT example *vs* E2E example.** Figures 4a and 4b depict the ratio between the CoT information $\mathcal{I}^{\text{CoT}}_{\mathcal{D},h_\star}(\varepsilon; \mathcal{H})$ and $\varepsilon$ as a function of $\varepsilon$. This ratio can be interpreted as the *value* of one CoT example compared to an end-to-end example, since the learning rate for E2E-supervision scales like $\log|\mathcal{H}|/\varepsilon$ whereas the rate for CoT supervision scales like $\log|\mathcal{H}|/\mathcal{I}^{\text{CoT}}_{\mathcal{D},h_\star}(\varepsilon; \mathcal{H})$. We clip $\varepsilon$ in this ratio to the minimal non-zero end-to-end error $\varepsilon^* = \min\{\mathcal{R}^{\text{e2e}}_{\mathcal{D},h_\star}(h) : \mathcal{R}^{\text{e2e}}_{\mathcal{D},h_\star}(h) \neq 0\}$, and denote this clipped value by $\varepsilon^+$. This is because to achieve a target error smaller than this critical level, $\varepsilon < \varepsilon^*$, we only need $\mathcal{O}(1/\varepsilon^*)$ samples, not $\mathcal{O}(1/\varepsilon)$ samples. In particular, achieving zero error is possible with $\mathcal{O}(1/\varepsilon^*)$ samples. The quantity $\lim_{\varepsilon \to 0} \mathcal{I}^{\text{CoT}}_{\mathcal{D},h_\star}(\varepsilon; \mathcal{H})/\varepsilon^+$ has the special interpretation of the ratio of the number of samples needed to achieve zero error under CoT supervision compared to E2E supervision.

**Varying input length.** In Figure 4a we vary the input sequence length $n$ in the input distribution $\mathcal{D}$. That is, we take $\mathcal{D}_n = \text{Unif}(\Sigma^n)$, and we compute the CoT information as a function of $\varepsilon$ for that distribution, varying $n$. We observe that the CoT information is increasing relative to $\varepsilon$ as the input length $n$ increases. An intuitive explanation for this is that longer inputs allow a bigger portion of the DFA's state transition map to be explored in a single example. Figure 4c depicts $\lim_{\varepsilon \to 0} \mathcal{I}^{\text{CoT}}_{\mathcal{D}_n}(\varepsilon; \mathcal{H})/\varepsilon^+$ as a function of the sequence length $n$. We see that this increases rapidly with $n$, suggesting that, for large $n$, the probability of a CoT trajectory agreeing for a pair of hypotheses with very different end-to-end behavior is vanishingly small. For $n = 10$, we see that

this value is roughly 600. By our theory (e.g., Result 1), this would suggest a $600\times$ improvement in sample complexity for learning with zero target error. (Indeed, this is supported by our numerical simulations on learning with CoT-Cons and E2E-Cons, as depicted in Figures 4e and 4f, which we will discuss further a bit later.)

**Varying CoT detail level.** Now, we consider fixing the input length to $n = 10$ and instead varying the level of detail in the CoT annotations. We do this by varying the proportion of the state trajectory that is revealed to the learner, denoted by $T \in [n]$. For each $T$, we run a simulation where the CoT trajectory revealed to the learner is the first $T$ symbols of the state trajectory. As expected, the CoT information monotonically increases with $T$. In Figure 4b we plot the ratio of CoT information to $\varepsilon$ as a function of $\varepsilon$, varying the level of detail $T$, and in Figure 4d we plot $\lim_{\varepsilon \to 0} \mathcal{I}^{\mathrm{CoT}}_{\mathcal{D},h_\star}(\varepsilon; \mathcal{H})/\varepsilon^+$. While this is monotonically increasing in $T$, it begins to plateau as $T$ increases, suggesting that there are diminishing returns when it comes to distinguishing hypotheses via their CoT trajectories—most of the information is revealed in the earlier portions of the CoT.

**Empirical sample complexity of CoT-Cons and E2E-Cons.** Next, we directly evaluate the sample complexity of CoT-supervised learning compared to E2E-supervised learning by running simulations using the CoT-Cons and E2E-Cons learning rules. We vary the sample size $m$, randomly draw a dataset $S_m$ of size $m$, and apply each learning rule to return a predictor $\mathcal{A}(S_m)$, then compute the generalization loss $\mathcal{R}^{\mathrm{e2e}}_{\mathcal{D}}(\mathcal{A}(S_m))$ of the returned predictor. The CoT-Cons and E2E-Cons learning rules are implemented by constructing the respective consistency sets and returning a predictor uniformly at random. We repeat this for 500 independent trials to estimate the distribution of $\mathcal{R}^{\mathrm{e2e}}_{\mathcal{D}}(\mathcal{A}(S_m))$ as a function of the sample size $m$ for each learning rule. Figure 4e depicts the empirical sample complexity. It is computed by computing the empirical average of the loss for each sample size $m$, and plotting the first sample size at which each target error level $\varepsilon$ is attained. Giving a complementary view, Figure 4f plots the empirical probability (over the random draw of the sample $S_m$) of returning a predictor with zero loss as a function of the sample size $m$. Across both figures, we see a gain in sample efficiency from CoT supervision of the order of $10^2 - 10^3$, which agrees with the theoretical predictions via the CoT information, $\lim_{\varepsilon \to 0} \mathcal{I}^{\mathrm{CoT}}_{\mathcal{D},h_\star}(\varepsilon; \mathcal{H})/\varepsilon^+ \approx 600$.

## C.2 Iterated linear thresholds

In practice, a common way of implementing CoT supervision is to consider a sequence model class (e.g., transformers) and to train the model to generate the CoT as a sequence token-by-token, before returning the final output. In this section, we consider another CoT hypothesis class that simulates a simple form of this autoregressive generation. In particular, we consider a sequence model class that generates tokens as a linear function of a fixed-size window of the history.

Fix a window size $d$, and let $w \in \{-1, 0, 1\}^d$ be a set of weights over this window. For a binary sequence $\boldsymbol{x} = (x_1, ..., x_n) \in \{0, 1\}^n$, we define the function $f_w : \boldsymbol{x} \mapsto (\boldsymbol{x}, z) \in \{0, 1\}^{|x|+1}$ as the function that returns a sequence with the symbol $z$ appended to $\boldsymbol{x}$, where $z$ is computed by applying a threshold to the $w$-weighted linear combination of the prior $d$ symbols,

$$f_w : \boldsymbol{x} \mapsto (\boldsymbol{x}, z) \in \{0, 1\}^{|x|+1}, \quad z = \mathbf{1}\left\{\sum_{i=0}^{d-1} w_i \cdot x_{n-i} \geq 0\right\}.$$

The CoT hypothesis class is defined by iterating $f_w$ $T$ times, taking the produced sequence as the CoT, and the final symbol as the output. That is, $\mathcal{H} = \{h_w : w \in \{-1, 0, 1\}^d\}$, where $h_w$ is defined as

$$h_w^{\mathrm{CoT}} : \boldsymbol{x} \mapsto (z_1, ..., z_T), \quad h_w^{\mathrm{e2e}} : \boldsymbol{x} \mapsto z_T$$
$$(\boldsymbol{x}, (z_1, \ldots, z_T)) = \underbrace{(f_w \circ \cdots \circ f_w)}_{T \text{ times}}(\boldsymbol{x})$$

This represents a simple type of autoregressive CoT hypothesis class, similar to the one studied in Joshi et al. [10]. In this section, we carry out a series of numerical simulations to explore what the CoT information says about CoT-supervised learning for such a class. We take the window size to be $d = 8$ and the number of iterations to be $T = 16$. The experimental results are depicted in Figure 5.

In Figure 5c we plot the CoT information $\mathcal{I}^{\mathrm{CoT}}_{\mathcal{D},h_\star}(\varepsilon; \mathcal{H})$, which depicts the monotonicity in $\varepsilon$ shown in Lemma 1. We observe that $\lim_{\varepsilon \to 0} \mathcal{I}^{\mathrm{CoT}}_{\mathcal{D},h_\star}(\varepsilon; \mathcal{H}) \approx 0.32 > \varepsilon^* \approx 0.05$, and

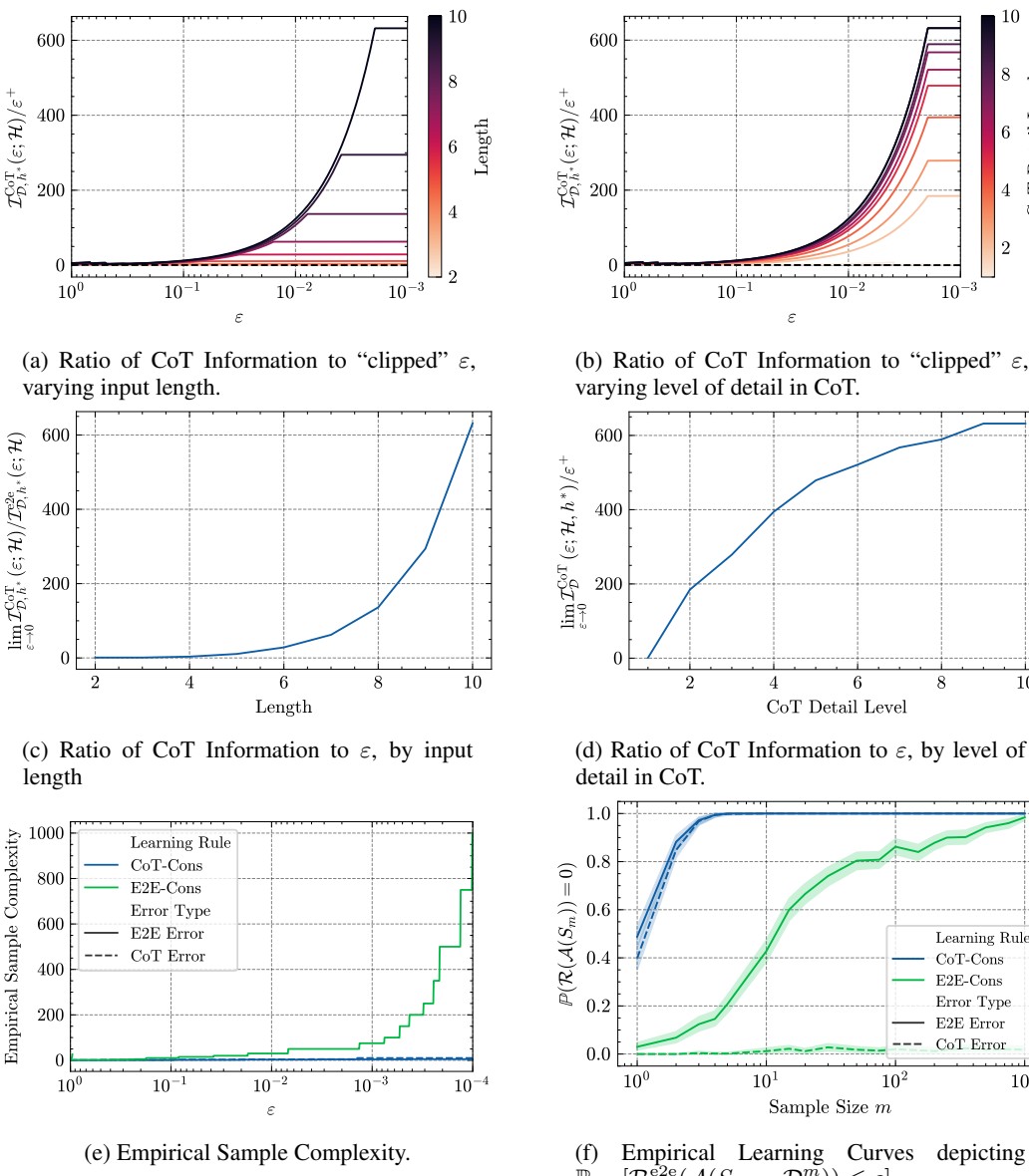

(a) Ratio of CoT Information to "clipped" $\varepsilon$, varying input length.

(b) Ratio of CoT Information to "clipped" $\varepsilon$, varying level of detail in CoT.

(c) Ratio of CoT Information to $\varepsilon$, by input length

(d) Ratio of CoT Information to $\varepsilon$, by level of detail in CoT.

(e) Empirical Sample Complexity.

(f) Empirical Learning Curves depicting $\mathbb{P}_{S^m}[\mathcal{R}_{\mathcal{D}}^{\mathrm{e2e}}(\mathcal{A}(S_m \sim \mathcal{D}^m)) \leq \varepsilon]$.

Figure 4: Numerical experiments for deterministic finite automata CoT hypothesis class.

$\lim_{\varepsilon \to 0} \mathcal{I}_{\mathcal{D}, h_\star}^{\mathrm{CoT}}(\varepsilon; \mathcal{H})/\varepsilon^+ \approx 6$. Consequently, our theory would suggest a $6\times$ gain in sample efficiency from CoT supervision. This matches remarkably well with experimental learning results depicted in Figures 5d to 5f. For example, Figure 5e depicts a roughly 5 fold improvement in sample complexity for `CoT-Cons` compared to `E2E-Cons` at the smallest target error levels.

# D  Proofs for Section 4: Upper Bounds

## D.1  Proof of Result 2

**Result** (Restatement: Learning Infinite Classes under CoT-Supervision). *Let $\mathcal{H} \subset (\mathcal{Y} \times \mathcal{Z})^{\mathcal{X}}$ be a CoT hypothesis class. For any distribution $\mathcal{D}$ over $\mathcal{X} \times \mathcal{Y} \times \mathcal{Z}$ realized by some $h_\star \in \mathcal{H}$, the CoT-consistency learning rule has a sample complexity of*

$$m(\varepsilon, \delta) = \mathcal{O}\left(\left(\frac{1}{\mathcal{I}_{\mathcal{D}, h_\star}^{\mathrm{CoT}}(\varepsilon; \mathcal{H})} + 1\right)\left(\mathrm{VC}(\mathcal{L}^{\mathrm{CoT}}(\mathcal{H})) \cdot \log\left(\frac{1}{\mathcal{I}_{\mathcal{D}, h_\star}^{\mathrm{CoT}}(\varepsilon; \mathcal{H})} + 1\right) + \log(1/\delta)\right)\right)$$

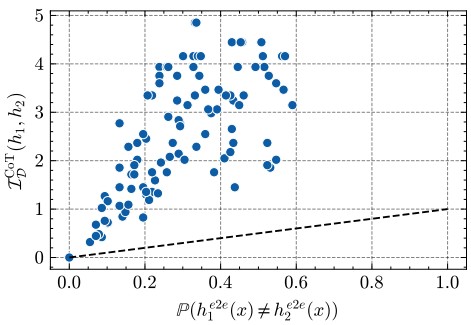

(a) Relativve CoT information between pairs of hypotheses plotted against their end-to-end disagreement.

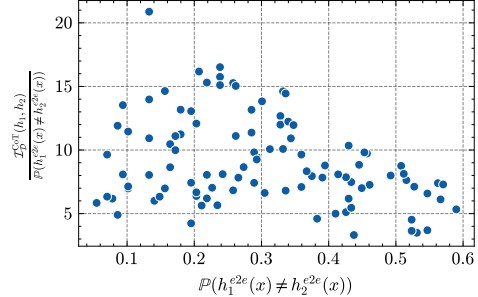

(b) The ratio of pairwise CoT information to end-to-end disagreement, plotted against the end-to-end disagreement.

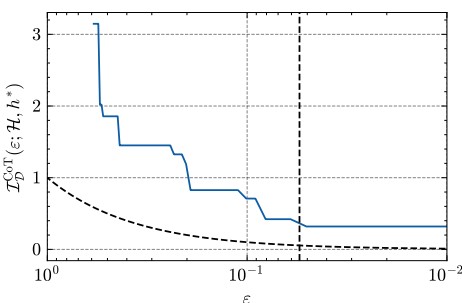

(c) CoT information $\mathcal{I}_{\mathcal{D},h_\star}^{\mathrm{CoT}}(\varepsilon;\mathcal{H})$ as a function of $\varepsilon$.

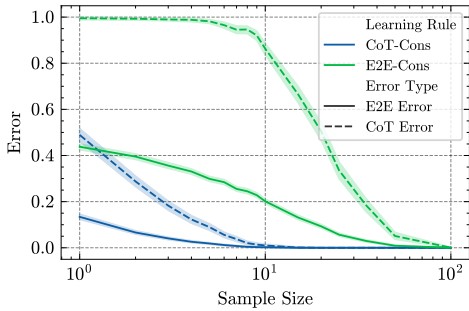

(d) Learning curves with and without CoT supervision.

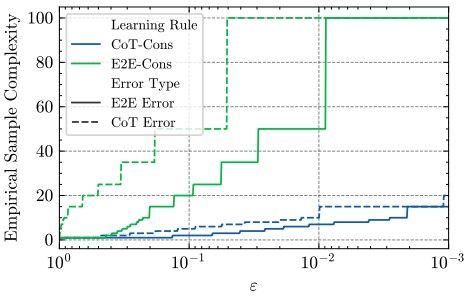

(e) Empirical sample complexity.

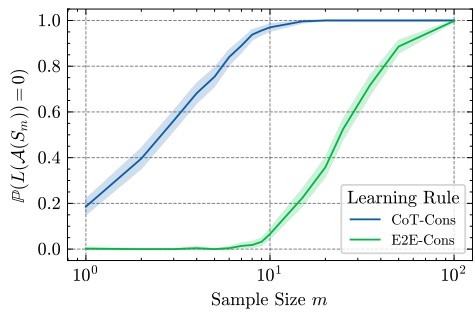

(f) Empirical probability of each learning rule returning a predictor with zero error.

Figure 5: Numerical experiments for iterated linear thresholds CoT hypothesis class.

*That is, for any $m \geq m(\varepsilon, \delta)$, we have that with probability at least $1 - \delta$ over $S \sim \mathcal{D}^m$,*

$$\forall h \in \texttt{CoT-Cons}(S; \mathcal{H}), \text{ we have } \mathcal{R}_{\mathcal{D}}^{\text{e2e}}(h) \leq \varepsilon.$$

The key to proving this result will be to establish the following lemma, which relates the performance of any *proper* CoT learner with respect to the CoT error to its performance with respect to the end-to-end error. As before, the intuition is that achieving small CoT error implies *very small* end-to-end error, because the CoT error measures any algorithmic errors, not only errors in the answer (which might be reachable via an incorrect algorithm). This relationship is captured by the CoT information.

Recall that a *proper* learner for $\mathcal{H}$ is defined as a learning algorithm that returns a predictor *in the hypothesis class*. In general, an improper learner may return any predictor, not necessarily in the hypothesis class (and this can have some computational advantages).

**Lemma 2** (Relating CoT performance to E2E performance via CoT Information). *Any proper CoT-learner $\mathcal{A} : (\mathcal{X} \times \mathcal{Y} \times \mathcal{Z})^* \to \mathcal{H}$ which achieves CoT-error $\varepsilon$ with sample complexity $m_{\mathcal{A}}^{\text{CoT}}(\varepsilon, \delta)$ also achieves end-to-end error $\varepsilon$ with sample complexity $m_{\mathcal{A}}^{\text{e2e}}(\varepsilon, \delta) \leq m_{\mathcal{A}}^{\text{CoT}}(\gamma(\varepsilon)^-, \delta)$, where $\gamma : (0, 1) \to (0, 1)$ is defined as*

$$\gamma(\varepsilon) := \inf \left\{ \mathcal{R}_{\mathcal{D}}^{\text{CoT}}(h) : h \in \Delta_{\mathcal{D}}^{\text{e2e}}(\varepsilon; \mathcal{H}) \right\}.$$

*Here, $m(\varepsilon^-, \delta)$ denotes the limit to $\varepsilon$ from below. Moreover, $\gamma$ can be related to the CoT information as follows*

$$\max \left( \frac{\mathcal{I}_{\mathcal{D}, h_\star}^{\text{CoT}}(\varepsilon; \mathcal{H})}{1 + \mathcal{I}_{\mathcal{D}, h_\star}^{\text{CoT}}(\varepsilon; \mathcal{H})}, \varepsilon \right) \leq \gamma(\varepsilon) \leq \min(\mathcal{I}_{\mathcal{D}, h_\star}^{\text{CoT}}(\varepsilon; \mathcal{H}), 1).$$

*Proof.* Let $\mathcal{A}$ be a proper CoT learner for $\mathcal{H}$ (i.e., it returns a hypothesis in $\mathcal{H}$) with CoT-error sample complexity $m_{\mathcal{A}}^{\text{CoT}}(\varepsilon, \delta)$. Let $m \geq m_{\mathcal{A}}^{\text{CoT}}(\gamma(\varepsilon), \delta)$ and let

$$S = \{(x_1, y_1, z_1), \ldots, (x_m, y_m, z_m)\} \overset{\text{i.i.d.}}{\sim} \mathcal{D}$$

be an i.i.d dataset drawn from the distribution $\mathcal{D}$. Note that we fold the hypothesis $h_\star$ into $\mathcal{D}$ for notational convenience, and we have $y_i, z_i = h_\star(x_i)$. By the assumption on the CoT-error sample complexity of $\mathcal{A}$, we have that with probability at least $1 - \delta$, $\hat{h} = \mathcal{A}(S)$ satisfies $\mathcal{R}_{\mathcal{D}}^{\text{CoT}}(\hat{h}) \leq \gamma(\varepsilon)^- < \gamma(\varepsilon)$.

To show that $\hat{h}$ has end-to-end error smaller than $\varepsilon$, we proceed by contradiction. Suppose we are in the event $\{S : \mathcal{R}_{\mathcal{D}}^{\text{CoT}}(\mathcal{A}(S)) < \gamma(\varepsilon)\}$ and that $\mathcal{R}_{\mathcal{D}}^{\text{e2e}}(\hat{h}) > \varepsilon$. This implies $\hat{h} \in \Delta_{\mathcal{D}}^{\text{e2e}}(\varepsilon; \mathcal{H})$ and hence

$$\gamma(\varepsilon) := \inf \left\{ \mathcal{R}_{\mathcal{D}}^{\text{CoT}}(h) : h \in \Delta_{\mathcal{D}}^{\text{e2e}}(\varepsilon, \mathcal{H}) \right\} \leq \mathcal{R}_{\mathcal{D}}^{\text{CoT}}(\hat{h}) \leq \gamma(\varepsilon)^- < \gamma(\varepsilon).$$

This yields a contradiction. Thus, on the event $\{S : \mathcal{R}_{\mathcal{D}}^{\text{CoT}}(\mathcal{A}(S)) < \gamma(\varepsilon)\}$, which occurs with probability at least $1 - \delta$, we have $\mathcal{R}_{\mathcal{D}}^{\text{e2e}}(\hat{h}) \leq \varepsilon$. This proves the first part of the lemma.

We now proceed to relate $\gamma$ to the CoT information. Note that the CoT information can be written in terms of $\gamma$ as follows

$$\begin{aligned}
\mathcal{I}_{\mathcal{D}, h_\star}^{\text{CoT}}(\varepsilon; \mathcal{H}) &:= \inf_{h \in \Delta_{\mathcal{D}}^{\text{e2e}}(\varepsilon; \mathcal{H})} \left\{ -\log \mathbb{P}_{x, y, z} [h(x) = (y, z)] \right\} \\
&= -\log \sup_{h \in \Delta_{\mathcal{D}}^{\text{e2e}}(\varepsilon; \mathcal{H})} \left( 1 - \mathcal{R}_{\mathcal{D}}^{\text{CoT}}(h) \right) \\
&= -\log \left( 1 - \inf_{h \in \Delta_{\mathcal{D}}^{\text{e2e}}(\varepsilon; \mathcal{H})} \mathcal{R}_{\mathcal{D}}^{\text{CoT}}(h) \right) \\
&= -\log(1 - \gamma(\varepsilon)).
\end{aligned}$$

The identity $-\log(1 - x) \geq x$ gives $\mathcal{I}_{\mathcal{D}, h_\star}^{\text{CoT}}(\varepsilon; \mathcal{H}) \geq \gamma(\varepsilon)$. The identity $-\log(1 - x) \leq \frac{x}{1-x}$ gives $\mathcal{I}_{\mathcal{D}, h_\star}^{\text{CoT}}(\varepsilon; \mathcal{H}) \leq \gamma(\varepsilon)/(1 - \gamma(\varepsilon))$ which can be rearranged to give $\gamma(\varepsilon) \geq \mathcal{I}_{\mathcal{D}, h_\star}^{\text{CoT}}(\varepsilon; \mathcal{H})/(1 + \mathcal{I}_{\mathcal{D}, h_\star}^{\text{CoT}}(\varepsilon; \mathcal{H}))$. Finally, note that $\gamma(\varepsilon) \geq \varepsilon$ by definition since $\mathcal{R}_{\mathcal{D}}^{\text{CoT}}(h) \geq \mathcal{R}_{\mathcal{D}}^{\text{e2e}}(h), \forall h$. $\square$

Note that the restriction that the CoT-learning algorithm $\mathcal{A}$ is *proper* was crucial in the proof above. In particular, we used $\hat{h} \in \mathcal{H}$ in order to derive the contradiction.

For a CoT hypothesis class $\mathcal{H} \subset (\mathcal{Y} \times \mathcal{Z})^{\mathcal{X}}$, recall that we define the CoT loss class over $\mathcal{X} \times \mathcal{Y} \times \mathcal{Z} \to \{0,1\}$ as the $0-1$ class

$$\mathcal{L}^{\mathrm{CoT}}(\mathcal{H}) := \left\{ \ell_h^{\mathrm{CoT}} : (x,y,z) \mapsto \mathbf{1}\{h(x) \neq (y,z)\} : \ h \in \mathcal{H} \right\}.$$

The complexity of this loss class will appear in our analysis since we will be analyzing learning algorithms that learn with respect to the CoT loss $\ell^{\mathrm{CoT}}$.

We are now ready to prove the main result.

*Proof of Result 2.* By Lemma 2, a CoT learner $\mathcal{A}$ with a sample complexity of $m_{\mathcal{A}}^{\mathrm{CoT}}(\varepsilon,\delta)$ with respect to the CoT error has a sample complexity with respect to the end-to-end error of at most $m_{\mathcal{A}}^{\mathrm{e2e}}(\varepsilon,\delta) \leq m_{\mathcal{A}}^{\mathrm{CoT}}(\gamma(\varepsilon)^-,\delta)$, where $\gamma$ is defined in the lemma. The CoT-consistency rule enjoys a sample complexity of

$$m_{\mathcal{A}}^{\mathrm{CoT}}(\varepsilon,\delta) = \mathcal{O}\left( \frac{1}{\varepsilon} \cdot \left( \mathrm{VC}(\mathcal{L}^{\mathrm{CoT}}(\mathcal{H})) \cdot \log(1/\varepsilon) + \log(1/\delta) \right) \right).$$

For the end-to-end error, this translates to the sample complexity of

$$m_{\mathcal{A}}^{\mathrm{e2e}}(\varepsilon,\delta) \leq m_{\mathcal{A}}^{\mathrm{CoT}}(\gamma(\varepsilon),\delta) \leq \mathcal{O}\left( \frac{1}{\gamma(\varepsilon)} \cdot \left( \mathrm{VC}(\mathcal{L}^{\mathrm{CoT}}(\mathcal{H})) \cdot \log(1/\gamma(\varepsilon)) + \log(1/\delta) \right) \right)$$

$$\leq \mathcal{O}\left( \frac{1 + \mathcal{I}_{\mathcal{D},h_\star}^{\mathrm{CoT}}(\varepsilon;\mathcal{H})}{\mathcal{I}_{\mathcal{D},h_\star}^{\mathrm{CoT}}(\varepsilon;\mathcal{H})} \cdot \left( \mathrm{VC}(\mathcal{L}^{\mathrm{CoT}}(\mathcal{H})) \cdot \log\left( \frac{1 + \mathcal{I}_{\mathcal{D},h_\star}^{\mathrm{CoT}}(\varepsilon;\mathcal{H})}{\mathcal{I}_{\mathcal{D},h_\star}^{\mathrm{CoT}}(\varepsilon;\mathcal{H})} \right) + \log(1/\delta) \right) \right)$$

$$= \mathcal{O}\left( \left( \frac{1}{\mathcal{I}_{\mathcal{D},h_\star}^{\mathrm{CoT}}(\varepsilon;\mathcal{H})} + 1 \right) \cdot \left( \mathrm{VC}(\mathcal{L}^{\mathrm{CoT}}(\mathcal{H})) \cdot \log\left( \frac{1}{\mathcal{I}_{\mathcal{D},h_\star}^{\mathrm{CoT}}(\varepsilon;\mathcal{H})} + 1 \right) + \log(1/\delta) \right) \right).$$

$\square$

## D.2 Proof of Result 3

**Result** (Restatement: Agnostic Learning under CoT-Supervision). *Let $\mathcal{H} \subset (\mathcal{Y} \times \mathcal{Z})^{\mathcal{X}}$ be a CoT hypothesis class. For any distribution $\mathcal{D}$ over $\mathcal{X} \times \mathcal{Y} \times \mathcal{Z}$, the CoT-ERM learning rule has a sample complexity of*

$$m(\varepsilon,\delta) = \mathcal{O}\left( \frac{\mathrm{VC}(\mathcal{L}^{\mathrm{CoT}}(\mathcal{H})) + \log(1/\delta)}{\widetilde{\mathcal{I}}_{\mathcal{D}}^{\mathrm{CoT}}(\varepsilon;\mathcal{H})^2} \right),$$

*where the agnostic version of the CoT information is defined as follows*

$$\widetilde{\mathcal{I}}_{\mathcal{D}}^{\mathrm{CoT}}(\varepsilon;\mathcal{H}) := \inf\left\{ \mathcal{R}_{\mathcal{D}}^{\mathrm{CoT}}(h) - \mathcal{R}_\star^{\mathrm{CoT}} : h \in \mathcal{H}, \mathcal{R}_{\mathcal{D}}^{\mathrm{e2e}}(h) - \mathcal{R}_\star^{\mathrm{e2e}} \geq \varepsilon \right\},$$

*where $\mathcal{R}_\star^{\mathrm{CoT}} := \inf_{h \in \mathcal{H}} \mathcal{R}_{\mathcal{D}}^{\mathrm{CoT}}(h)$, $\mathcal{R}_\star^{\mathrm{e2e}} := \inf_{h \in \mathcal{H}} \mathcal{R}_{\mathcal{D}}^{\mathrm{e2e}}(h)$. That is, for any $m \geq m(\varepsilon,\delta)$, we have that with probability at least $1 - \delta$ over $S \sim \mathcal{D}^m$, the excess end-to-end risk is bounded as*

$$\forall h \in \mathtt{CoT\text{-}ERM}(S;\mathcal{H}), \text{ we have } \mathcal{R}_{\mathcal{D}}^{\mathrm{e2e}}(h) \leq \mathcal{R}_\star^{\mathrm{e2e}} + \varepsilon.$$

Our aim is to analyze the performance of the CoT-ERM learning rule, which seeks to minimize the CoT-penalized error. This is a natural learning rule to consider in the CoT-supervised setting, and corresponds to optimization procedures that are implemented in practice in CoT learning. Similar to the realizable setting, the key to proving this learning guarantee is to relate the CoT error of a CoT learner to its end-to-end error. This is established in the following lemma, which is an analogue of Lemma 2.

Recall that, for a distribution $\mathcal{D}$ over $\mathcal{X} \times \mathcal{Y} \times \mathcal{Z}$ and a CoT hypothesis class $\mathcal{H} : \mathcal{X} \to \mathcal{Y} \times \mathcal{Z}$, we define the optimal end-to-end and CoT errors achievable by $\mathcal{H}$ as

$$\mathcal{R}_\star^{\mathrm{CoT}} := \inf_{h \in \mathcal{H}} \mathcal{R}_{\mathcal{D}}^{\mathrm{CoT}}(h), \ \mathcal{R}_\star^{\mathrm{e2e}} := \inf_{h \in \mathcal{H}} \mathcal{R}_{\mathcal{D}}^{\mathrm{e2e}}(h).$$

**Lemma 3** (Relating CoT performance to E2E performance in the Agnostic Setting). *Any agnostic proper CoT-learner $\mathcal{A} : (\mathcal{X} \times \mathcal{Y} \times \mathcal{Z})^* \to \mathcal{H}$ which achieves excess CoT-error $\varepsilon$ with sample complexity $m_{\mathcal{A}}^{\mathrm{CoT}}(\varepsilon, \delta)$ also achieves excess end-to-end error $\varepsilon$ with sample complexity $m_{\mathcal{A}}^{\mathrm{e2e}}(\varepsilon, \delta) \leq m_{\mathcal{A}}^{\mathrm{CoT}}(\gamma(\varepsilon)^-, \delta)$, where $\gamma : (0, 1) \to (0, 1)$ is defined as*

$$\gamma(\varepsilon) := \widetilde{\mathcal{I}}_{\mathcal{D}}^{\mathrm{CoT}}(\varepsilon; \mathcal{H}) = \inf \left\{ \mathcal{R}_{\mathcal{D}}^{\mathrm{CoT}}(h) - \mathcal{R}_{\star}^{\mathrm{CoT}} \, : \, h \in \mathcal{H}, \, \mathcal{R}_{\mathcal{D}}^{\mathrm{e2e}}(h) - \mathcal{R}_{\star}^{\mathrm{e2e}} \geq \varepsilon \right\}.$$

*Proof.* Let $\mathcal{A}$ be a proper CoT learner for $\mathcal{H}$ (i.e., it returns a hypothesis in $\mathcal{H}$) with CoT-error sample complexity $m_{\mathcal{A}}^{\mathrm{CoT}}(\varepsilon, \delta)$. Let $m \geq m_{\mathcal{A}}^{\mathrm{CoT}}(\gamma(\varepsilon)^-, \delta)$ and let

$$S = \{(x_1, y_1, z_1), \ldots, (x_m, y_m, z_m)\} \overset{\mathrm{i.i.d.}}{\sim} \mathcal{D}$$

be an i.i.d dataset drawn from the distribution $\mathcal{D}$. By the assumption on the CoT-error sample complexity of $\mathcal{A}$, we have that with probability at least $1 - \delta$, $\hat{h} = \mathcal{A}(S)$ satisfies $\mathcal{R}_{\mathcal{D}}^{\mathrm{CoT}}(\hat{h}) < \mathcal{R}_{\star}^{\mathrm{CoT}} + \gamma(\varepsilon)$.

To show that $\hat{h}$ has end-to-end error smaller than $\varepsilon$, we proceed by contradiction. Suppose we are in the event $\{S : \mathcal{R}_{\mathcal{D}}^{\mathrm{CoT}}(\mathcal{S}(S)) < \mathcal{R}_{\star}^{\mathrm{CoT}} + \gamma(\varepsilon)\}$ and that the end-to-end error is larger than desired $\mathcal{R}_{\mathcal{D}}^{\mathrm{e2e}}(\hat{h}) > \mathcal{R}_{\star}^{\mathrm{e2e}} + \varepsilon$. This implies

$$\gamma(\varepsilon) := \inf \left\{ \mathcal{R}_{\mathcal{D}}^{\mathrm{CoT}}(h) - \mathcal{R}_{\star}^{\mathrm{CoT}} \, : \, h \in \mathcal{H}, \, \mathcal{R}_{\mathcal{D}}^{\mathrm{e2e}}(h) \geq \mathcal{R}_{\star}^{\mathrm{e2e}} + \varepsilon \right\} \leq \mathcal{R}_{\mathcal{D}}^{\mathrm{CoT}}(\hat{h}) - \mathcal{R}_{\star}^{\mathrm{CoT}} < \gamma(\varepsilon).$$

This yields a contradiction. Thus, on the event $\{S : \mathcal{R}_{\mathcal{D}}^{\mathrm{CoT}}(\mathcal{S}(S)) < \mathcal{R}_{\star}^{\mathrm{CoT}} + \gamma(\varepsilon)\}$, which occurs with probability at least $1 - \delta$, we must have $\mathcal{R}_{\mathcal{D}}^{\mathrm{e2e}}(\mathcal{A}(S)) \leq \mathcal{R}_{\star}^{\mathrm{e2e}} + \varepsilon$. $\qquad \square$

We can now prove our main result, which follows by a similar argument to Result 2.

*Proof of Result 3.* By Lemma 3, a CoT learner $\mathcal{A}$ with a sample complexity of $m_{\mathcal{A}}^{\mathrm{CoT}}(\varepsilon, \delta)$ with respect to the CoT error has a sample complexity with respect to the end-to-end error of at most $m_{\mathcal{A}}^{\mathrm{e2e}}(\varepsilon, \delta) \leq m_{\mathcal{A}}^{\mathrm{CoT}}(\gamma(\varepsilon)^-, \delta)$, where $\gamma(\varepsilon) = \widetilde{\mathcal{I}}_{\mathcal{D}}^{\mathrm{CoT}}(\varepsilon; \mathcal{H})$ is the agnostic version of the CoT information. The CoT-ERM rule enjoys a sample complexity of

$$m_{\mathcal{A}}^{\mathrm{CoT}}(\varepsilon, \delta) = \mathcal{O}\left( \frac{1}{\varepsilon^2} \cdot \left( \mathrm{VC}(\mathcal{L}^{\mathrm{CoT}}(\mathcal{H})) + \log(1/\delta) \right) \right).$$

For the end-to-end error, this translates to the sample complexity

$$m(\varepsilon, \delta) = \mathcal{O}\left( \frac{\mathrm{VC}(\mathcal{L}^{\mathrm{CoT}}(\mathcal{H})) + \log(1/\delta)}{\widetilde{\mathcal{I}}_{\mathcal{D}}^{\mathrm{CoT}}(\varepsilon; \mathcal{H})^2} \right),$$

$\qquad \square$

# E  Proofs of Section 5: Lower Bounds

## E.1  Proof of Result 4

We will break down Result 4 into several statements and prove each separately.

**Result** (First Part of Result 4). *Let $\mathcal{H} \subset (\mathcal{Y} \times \mathcal{Z})^{\mathcal{X}}$ be a CoT hypothesis class and let $\mathcal{D}$ be a distribution on $\mathcal{X}$. Let $x_1, \ldots, x_m \sim \mathcal{D}$ be an i.i.d sample from $\mathcal{D}$. For any $h_\star \in \mathcal{H}$ and $\varepsilon > 0$, we have that*

$$m < \frac{\log(1/\delta)}{\mathcal{I}_{\mathcal{D}, h_\star}^{\mathrm{CoT}}(\varepsilon; \mathcal{H})}$$

*implies that with probability at least $\delta$, there exists $h \in \mathcal{H}$ with end-to-end error at least $\varepsilon$ which is indistinguishable from $h_\star$ on this sample.*

*Proof.* Fix $h_\star \in \mathcal{H}$ and $\varepsilon \in [0, 1]$. Let $\bar{h} \in \arg\min_{h \in \Delta_{\mathcal{D}}^{\mathrm{e2e}}(\varepsilon; \mathcal{H}, h_\star)} \mathcal{I}_{\mathcal{D}}^{\mathrm{CoT}}(h_\star, h)$. (If the infimum is not attained, let $\bar{h}$ be an $\eta$-minimizer and take $\eta \to 0$.) Then, by definition, we have that $\bar{h}$ has end-to-end error at least $\varepsilon$ and $\mathcal{I}_{\mathcal{D}, h_\star}^{\mathrm{CoT}}(\varepsilon; \mathcal{H}) = \mathcal{I}_{\mathcal{D}}^{\mathrm{CoT}}(h_\star, \bar{h})$. Thus, the probability that $h_\star$ and $\bar{h}$ agree on a random input $x$ with respect to both the CoT and E2E behavior can be expressed as

$$\mathbb{P}_{x \sim \mathcal{D}} \left[ \bar{h}^{\mathrm{CoT}}(x) = h_\star^{\mathrm{CoT}}(x), \, \bar{h}^{\mathrm{e2e}}(x) = h_\star^{\mathrm{e2e}}(x) \right] = \exp(-\mathcal{I}_{\mathcal{D}, h_\star}^{\mathrm{CoT}}(\varepsilon; \mathcal{H})).$$

Now, we compute the probability that $h_\star$ and $\bar{h}$ are indistinguishable on the CoT-annotated sample of $m$ points $x_1, \ldots, x_m \overset{\text{i.i.d.}}{\sim} \mathcal{D}$:

$$\underset{x_1,\ldots,x_m \overset{\text{i.i.d.}}{\sim} \mathcal{D}}{\mathbb{P}} \left[ \forall i \in [m], \bar{h}^{\text{CoT}}(x_i) = h_\star^{\text{CoT}}(x_i), \, \bar{h}^{\text{e2e}}(x_i) = h_\star^{\text{e2e}}(x_i) \right]$$

$$= \left( \underset{x \sim \mathcal{D}}{\mathbb{P}} \left[ \bar{h}^{\text{CoT}}(x) = h_\star^{\text{CoT}}(x), \, \bar{h}^{\text{e2e}}(x) = h_\star^{\text{e2e}}(x) \right] \right)^m$$

$$= \exp(-m \cdot \mathcal{I}_{\mathcal{D},h_\star}^{\text{CoT}}(\varepsilon; \mathcal{H})).$$

This occurs with probability at least $\delta$ when

$$m \leq \frac{\log(1/\delta)}{\mathcal{I}_{\mathcal{D},h_\star}^{\text{CoT}}(\varepsilon; \mathcal{H})}.$$

$\square$

The next lower bound result is based on relating the learning problem to binary hypothesis testing and lower bounding the sample complexity of hypothesis testing via the total variation distance. The basic idea of relating the performance of a statistical estimator to the total variation distance is due to LeCam [11]. We also point to Yu [39] for a classic reference on statistical lower bounds, including Le Cam's method, as well as Polyanskiy and Wu [40] for a modern reference.

Henceforth, for a hypothesis $h \in \mathcal{H}$, we will denote by $P_h \in \mathcal{P}(\mathcal{X} \times \mathcal{Y} \times \mathcal{Z})$ the distribution over input-CoT-output tuples where the marginal on $\mathcal{X}$ is the input distribution $\mathcal{D}$ and the distribution over $(y, z)$ given $x$ is the Dirac measure at $h(x)$.

**Result** (Second Part of Result 4). *Let $\mathcal{A} : (\mathcal{X} \times \mathcal{Y} \times \mathcal{Z})^* \to \mathcal{H}$ be any learning algorithm that maps a dataset $S^m = \{(x_i, y_i, z_i)\}_{i=1}^m$ to a predictor $\hat{h}$. Suppose there exists $h_1, h_2 \in \mathcal{H}$ such that $\underset{x \sim \mathcal{D}}{\mathbb{P}} \left[ h_1^{\text{e2e}}(x) \neq h_2^{\text{e2e}}(x) \right] \geq 2\varepsilon$. Assume that $S^m \sim (\mathcal{D} \otimes \delta_{(y,z)=h_\star(x)})^{\otimes m}$ the sample size $m$ is upper bounded as*

$$m < \frac{\log(\frac{1}{2\delta})}{\mathcal{I}_{\mathcal{D}}^{\text{CoT}}(h_1, h_2)}.$$

*Then, we must have*

$$\inf_{h_\star \in \mathcal{H}} \underset{S^m \sim h_\star}{\mathbb{P}} \left[ \mathcal{R}_{\mathcal{D},h_\star}^{\text{e2e}}(\mathcal{A}(S^m)) \geq \varepsilon \right] > \delta.$$

*Moreover, the expected error of any CoT-learning algorithm $\mathcal{A}$ is lower-bounded as,*

$$\sup_{h_\star \in \mathcal{H}} \underset{S^m \sim h_\star}{\mathbb{E}} \left[ \mathcal{R}_{\mathcal{D},h_\star}^{\text{e2e}}(\mathcal{A}(S^m)) \right] \geq \frac{1}{2} \sup_{h_1,h_2 \in \mathcal{H}} \underset{x \sim \mathcal{D}}{\mathbb{P}} \left[ h_1^{\text{e2e}}(x) \neq h_2^{\text{e2e}}(x) \right] \cdot \exp(-m \cdot \mathcal{I}_{\mathcal{D}}^{\text{CoT}}(h_1, h_2))$$

$$\geq \frac{1}{2} \sup_{\substack{h_\star \in \mathcal{H} \\ \varepsilon > 0}} \varepsilon \cdot \exp(-m \cdot \mathcal{I}_{\mathcal{D},h_\star}^{\text{CoT}}(\varepsilon; \mathcal{H})).$$

*Proof.* Let us consider the pseudometric on the hypothesis space $\mathcal{H}$, defined by

$$d_{\mathcal{D}}^{\text{e2e}}(h_1, h_2) = \underset{x \sim \mathcal{D}}{\mathbb{P}} \left[ h_1^{\text{e2e}}(x) \neq h_2^{\text{e2e}}(x) \right],$$

which measures the end-to-end disagreement. Note that $\mathcal{R}_{\mathcal{D}}^{\text{e2e}}(h) = d_{\mathcal{D}}^{\text{e2e}}(h, h_\star)$. Moreover, note that $d_{\mathcal{D}}^{\text{e2e}}$ satisfies

$$d_{\mathcal{D}}^{\text{e2e}}(h_1, h_3) \leq d_{\mathcal{D}}^{\text{e2e}}(h_1, h_2) + d_{\mathcal{D}}^{\text{e2e}}(h_2, h_3).$$

Let $\mathcal{A} : (\mathcal{X} \times \mathcal{Y} \times \mathcal{Z})^* \to \mathcal{Y}^{\mathcal{X}}$ be any learning algorithm. Assume towards a contradiction that $\underset{S \sim P_h^{\otimes m}}{\mathbb{P}} \left[ \mathcal{R}_{\mathcal{D}}^{\text{e2e}}(\mathcal{A}(S)) \geq \varepsilon \right] \leq \delta, \forall h \in \mathcal{H}$. By assumption, there exists a pair of hypotheses $h_1, h_2$ such that $d_{\mathcal{D}}^{\text{e2e}}(h_1, h_2) \geq 2\varepsilon$. Consider the event that the predictor returned by $\mathcal{A}$ is close to $h_0$ in end-to-end behavior, $\mathcal{E} := \{d_{\mathcal{D}}^{\text{e2e}}(h_0, \mathcal{A}(S)) < \varepsilon\}$. We will consider the probability of this event when the data is generated by $h_1$ and $h_2$. By the assumption on the performance of the algorithm, we have that the probability of this event under $h_1$ is bounded as

$$\underset{S \sim P_{h_1}^{\otimes m}}{\mathbb{P}} \left[ d_{\mathcal{D}}^{\text{e2e}}(h_0, \mathcal{A}(S)) < \varepsilon \right] \geq 1 - \delta.$$

On the other hand, under $S \sim P_{h_2}^{\otimes m}$ note that by the triangle inequality of $d_{\mathcal{D}}^{\text{e2e}}$, we have

$$d_{\mathcal{D}}^{\text{e2e}}(h_1, h_2) \leq d_{\mathcal{D}}^{\text{e2e}}(h_1, \mathcal{A}(S)) + d_{\mathcal{D}}^{\text{e2e}}(\mathcal{A}(S), h_1) \iff d_{\mathcal{D}}^{\text{e2e}}(h_0, \mathcal{A}(S) \geq \underbrace{d_{\mathcal{D}}^{\text{e2e}}(h_1, h_2)}_{\geq 2\varepsilon} - \underbrace{d_{\mathcal{D}}^{\text{e2e}}(\mathcal{A}(S), h_1)}_{\geq \varepsilon \text{ w.p. } \leq \delta},$$

where the first bound is by the assumption on $h_1, h_2$ and the second is by the assumption on the performance of the learning algorithm $\mathcal{A}$. This then implies that

$$\mathop{\mathbb{P}}_{S \sim P_{h_2}^{\otimes m}} \left[ d_{\mathcal{D}}^{\text{e2e}}(h_0, \mathcal{A}(S)) < \varepsilon \right] \leq \delta.$$

By the definition of the total variation distance, this then implies that the total variation distance between $P_{h_1}^{\otimes m}$ and $P_{h_2}^{\otimes m}$ must be at least

$$\text{TV}\left( P_{h_1}^{\otimes m}, P_{h_2}^{\otimes m} \right) := \sup_A \left| P_{h_1}^{\otimes m}(A) - P_{h_2}^{\otimes m}(A) \right| \geq P_{h_1}^{\otimes m}(\mathcal{E}) - P_{h_2}^{\otimes m}(\mathcal{E}) \geq 1 - 2\delta.$$

Thus, to derive a contradiction, we will relate the total variation distance to the relative CoT information and choose $m$ small enough such that $\text{TV}\left( P_{h_1}^{\otimes m}, P_{h_2}^{\otimes m} \right) < 1 - 2\delta$. We compute the total variation distance as follows:

$$\text{TV}\left( P_{h_1}^{\otimes m}, P_{h_2}^{\otimes m} \right) := \frac{1}{2} \sum_{(x_{1:m}, y_{1:m}, z_{1:m})} |P_{h_1}(x_{1:m}, y_{1:m}, z_{1:m}) - P_{h_2}(x_{1:m}, y_{1:m}, z_{1:m})|$$

$$= \frac{1}{2} \sum_{x_{1:m}} \mathcal{D}(x_{1:m}) \sum_{y_{1:m}, z_{1:m}} |\mathbf{1}\{(y_{1:m}, z_{1:m}) = h_1(x_{1:m})\} - \mathbf{1}\{(y_{1:m}, z_{1:m}) = h_2(x_{1:m})\}|$$

$$\overset{(a)}{=} \sum_{x_{1:m}} \mathcal{D}(x_{1:m}) \mathbf{1}\{h_1(x_{1:m}) \neq h_2(x_{1:m})\}$$

$$= \mathop{\mathbb{P}}_{x_{1:m} \sim \mathcal{D}^{\otimes m}} [\exists i \in [m] : h_1(x_i) \neq h_2(x_i)]$$

$$= 1 - \mathop{\mathbb{P}}_{x \sim \mathcal{D}} [h_1(x) = h_2(x)]^m$$

$$= 1 - \exp(-m \cdot \mathcal{I}_{\mathcal{D}}^{\text{CoT}}(h_1, h_2))$$

To see step (a), note that the function

$$\Delta\{h_1, h_2\}(x_{1:m}, y_{1:m}, z_{1:m}) := |\mathbf{1}\{(y_{1:m}, z_{1:m}) = h_1(x_{1:m})\} - \mathbf{1}\{(y_{1:m}, z_{1:m}) = h_2(x_{1:m})\}|$$

takes the value 1 either if $h_1(x_{1:m}) = (y_{1:m}, z_{1:m})$ and $h_2(x_{1:m}) \neq (y_{1:m}, z_{1:m})$ or if $h_2(x_{1:m}) = (y_{1:m}, z_{1:m})$ and $h_1(x_{1:m}) \neq (y_{1:m}, z_{1:m})$. Thus, in the sum $\sum_{y_{1:m}, z_{1:m}}$ we only need to consider values of $y_{1:m}, z_{1:m}$ that agree with at least one of $h_1, h_2$.

To guarantee that $\text{TV}(P_{h_1}, P_{h_2}) < 1 - 2\delta$, it is enough to have

$$m < \frac{\log(\frac{1}{2\delta})}{\mathcal{I}_{\mathcal{D}}^{\text{CoT}}(h_1, h_2)}.$$

This proves the first statement.

The second statement, in terms of the expected error, can be proven by a an analogous argument and related to the CoT information via the above calculation of the TV distance. In particular, fix any $h_1, h_2 \in \mathcal{H}$, and consider the predictor returned by the algorithm $\hat{h} = \mathcal{A}(S)$. Convert this predictor to a randomized test as follows

$$\widetilde{h} = \begin{cases} h_1 & \text{w.p. } \frac{d_{\mathcal{D}}^{\text{e2e}}(h_2, \hat{h})}{d_{\mathcal{D}}^{\text{e2e}}(h_1, \hat{h}) + d_{\mathcal{D}}^{\text{e2e}}(h_2, \hat{h})} \\ h_2 & \text{w.p. } \frac{d_{\mathcal{D}}^{\text{e2e}}(h_1, \hat{h})}{d_{\mathcal{D}}^{\text{e2e}}(h_1, \hat{h}) + d_{\mathcal{D}}^{\text{e2e}}(h_2, \hat{h})}. \end{cases}$$

Under $h_1$, we can lower bound the expected error via the triangle inequality as follows

$$\mathop{\mathbb{E}}_{S \sim P_{h_1}^{\otimes m}} \left[ d_{\mathcal{D}}^{\text{e2e}}(\widetilde{h}, h_1) \right] = d_{\mathcal{D}}^{\text{e2e}}(h_1, h_2) \mathop{\mathbb{E}}_{S \sim P_{h_1}^{\otimes m}} \left[ \frac{d_{\mathcal{D}}^{\text{e2e}}(h_1, \hat{h})}{d_{\mathcal{D}}^{\text{e2e}}(h_1, \hat{h}) + d_{\mathcal{D}}^{\text{e2e}}(h_2, \hat{h})} \right] \leq \mathop{\mathbb{E}}_{S \sim P_{h_1}^{\otimes m}} \left[ d_{\mathcal{D}}^{\text{e2e}}(\hat{h}, h_1) \right],$$

where we used the fact that $d_{\mathcal{D}}^{\text{e2e}}(h_1, \hat{h}) + d_{\mathcal{D}}^{\text{e2e}}(h_2, \hat{h}) \geq d_{\mathcal{D}}^{\text{e2e}}(h_1, h_2)$. Similarly, under $h_2$, we have

$$\mathop{\mathbb{E}}_{S \sim P_{h_2}^{\otimes m}} \left[ d_{\mathcal{D}}^{\text{e2e}}(\widetilde{h}, h_2) \right] = d_{\mathcal{D}}^{\text{e2e}}(h_1, h_2) \mathop{\mathbb{E}}_{S \sim P_{h_2}^{\otimes m}} \left[ \frac{d_{\mathcal{D}}^{\text{e2e}}(h_2, \hat{h})}{d_{\mathcal{D}}^{\text{e2e}}(h_1, \hat{h}) + d_{\mathcal{D}}^{\text{e2e}}(h_2, \hat{h})} \right] \leq \mathop{\mathbb{E}}_{S \sim P_{h_2}^{\otimes m}} \left[ d_{\mathcal{D}}^{\text{e2e}}(\hat{h}, h_1) \right].$$

Now, consider the prior $\pi = \frac{1}{2}(\delta_{h_1} + \delta_{h_2})$ and let $h_\star \sim \pi$. Then, we have

$$\sup_{h_\star \in \mathcal{H}} \mathop{\mathbb{E}}_{S \sim P_{h_\star}^{\otimes m}} \left[ d_{\mathcal{D}}^{\text{e2e}}(\mathcal{A}(S), h_\star) \right] \geq \mathop{\mathbb{E}}_{h_\star \sim \pi} \left[ \mathop{\mathbb{E}}_{S \sim P_{h_\star}^{\otimes m}} \left[ d_{\mathcal{D}}^{\text{e2e}}(\mathcal{A}(S), h_\star) \right] \right]$$

$$= \frac{1}{2} \cdot \left( \mathop{\mathbb{E}}_{S \sim P_{h_1}^{\otimes m}} \left[ d_{\mathcal{D}}^{\text{e2e}}(\mathcal{A}(S), h_1) \right] + \mathop{\mathbb{E}}_{S \sim P_{h_2}^{\otimes m}} \left[ d_{\mathcal{D}}^{\text{e2e}}(\mathcal{A}(S), h_2) \right] \right)$$

$$\geq d_{\mathcal{D}}^{\text{e2e}}(h_1, h_2) \cdot \frac{1}{2} \cdot \left( \mathop{\mathbb{P}}_{S \sim P_{h_1}^{\otimes m}} \left[ \widetilde{h} \neq h_1 \right] + \mathop{\mathbb{P}}_{S \sim P_{h_2}^{\otimes m}} \left[ \widetilde{h} \neq h_2 \right] \right)$$

$$\geq \frac{d_{\mathcal{D}}^{\text{e2e}}(h_1, h_2)}{2} \cdot \left( 1 - \text{TV} \left( P_{h_1}^{\otimes m}, P_{h_2}^{\otimes m} \right) \right),$$

where the last inequality follows from the minimum average probability of error in binary hypothesis testing (or, equivalently, the supremum representation of the total variation distance). Now, using the previous calculation of the total variation distance in terms of the CoT information, we have

$$\sup_{h_\star \in \mathcal{H}} \mathop{\mathbb{E}}_{S \sim P_{h_\star}^{\otimes m}} \left[ d_{\mathcal{D}}^{\text{e2e}}(\mathcal{A}(S), h_\star) \right] \geq \sup_{h_1, h_2 \in \mathcal{H}} \mathop{\mathbb{P}}_{x \sim \mathcal{D}} \left[ h_1^{\text{e2e}}(x) \neq h_2^{\text{e2e}}(x) \right] \cdot \exp(-m \cdot \mathcal{I}_{\mathcal{D}}^{\text{CoT}}(h_1, h_2))$$

$$\geq \sup_{h_\star \in \mathcal{H}} \sup_{\substack{\varepsilon > 0 \\ h \in \Delta_{\mathcal{D}}^{\text{e2e}}(\varepsilon; \mathcal{H})}} \mathop{\mathbb{P}}_{x \sim \mathcal{D}} \left[ h_\star^{\text{e2e}}(x) \neq h^{\text{e2e}}(x) \right] \cdot \exp(-m \cdot \mathcal{I}_{\mathcal{D}}^{\text{CoT}}(h_\star, h))$$

$$\overset{(a)}{\geq} \sup_{h_\star \in \mathcal{H}} \sup_{\varepsilon > 0} \varepsilon \cdot \exp(-m \cdot \inf_{h \in \Delta_{\mathcal{D}}^{\text{e2e}}(\varepsilon; \mathcal{H})} \mathcal{I}_{\mathcal{D}}^{\text{CoT}}(h_\star, h))$$

$$\overset{(b)}{=} \sup_{\substack{h_\star \in \mathcal{H} \\ \varepsilon > 0}} \varepsilon \cdot \exp(-m \cdot \mathcal{I}_{\mathcal{D}, h_\star}^{\text{CoT}}(\varepsilon; \mathcal{H})).$$

In step (a) we used the fact that $\mathop{\mathbb{P}}_{x \sim \mathcal{D}} \left[ h_\star^{\text{e2e}}(x) \neq h^{\text{e2e}}(x) \right] \geq \varepsilon, \forall h \in h \in \Delta_{\mathcal{D}}^{\text{e2e}}(\varepsilon; \mathcal{H})$, and in step (b) we used the definition of the CoT information $\mathcal{I}_{\mathcal{D}, h_\star}^{\text{CoT}}(\varepsilon; \mathcal{H}) := \inf_{h \in \mathcal{I}_{\mathcal{D}, h_\star}^{\text{CoT}}(\varepsilon; \mathcal{H})} \mathcal{I}_{\mathcal{D}}^{\text{CoT}}(h, h_\star)$. $\qquad \square$

### E.2  Proof of Result 5

The next upper bound will use information-theoretic tools to establish a lower bound that scales with the size of the hypothesis space. As with the previous lower bound result, the strategy will be to reduce the learning problem into a hypothesis testing problem. However, unlike the previous results, which considered binary hypothesis testing, here we will consider a reduction to multiple hypothesis testing. The main idea is to test between a finite collection of hypotheses whose minimum end-to-end disagreement is $\varepsilon$. If we can show that it is impossible to reliably distinguish between these hypotheses with a given sample size, then this implies that the best learning algorithm must at least incur an end-to-end error proportional to $\varepsilon$.

To state our result, we begin by recalling the definition of an $\varepsilon$-packing.

**Definition.** *Let $\mathbb{X}$ be a set and let $d$ be a (pseudo)metric. A subset $\{x_1, \ldots, x_M\} \subset \mathbb{X}$ is called an $\varepsilon$-packing of $\mathbb{X}$ with respect to $d$ if $\min_{i \neq j} d(x_i, x_j) \geq \varepsilon$. The packing number is defined as the size of the maximum packing, $M(\varepsilon; \mathbb{X}, d) := \max\{m : \exists \ \varepsilon\text{-packing of } \mathbb{X} \text{ of size } m\}$.*

The main information-theoretic tool we will use will be Fano's inequality [39, 41], stated below.

**Lemma** (Fano's Inequality). *Let $W \to X \to Y \to \hat{W}$ be a Markov chain, and assume $W \sim \text{Unif}([M])$. Then,*

$$P_e := \mathbb{P}\left[ W \neq \hat{W} \right] \geq 1 - \frac{I(X; Y) + h_b(P_e)}{\log M} \geq 1 - \frac{I(X; Y) + \log 2}{\log M}.$$

Similar to the previous section, for $h \in \mathcal{H}$, we will denote by $P_h$ the distribution on $\mathcal{X} \times \mathcal{Y} \times \mathcal{Z}$ induced by the hypothesis $h$. As before, the marginal over $\mathcal{X}$ is $\mathcal{D}$ for all $P_h$. However, unlike the previous section, $P_h(y, z \mid x)$ is not a Dirac measure since to first passes through the noisy channel $Q$. The distribution $P_h$ is defined as

$$P_h(x, y, z) = \mathcal{D}(x) \cdot Q(y, z \mid h(x)).$$

We are now ready to prove the result.

**Result** (Restatement of Result 5). *Let $\mathcal{H} \subset (\mathcal{Y} \times \mathcal{Z})^{\mathcal{X}}$ be a CoT hypothesis class and let $\mathcal{D}$ be a distribution over $\mathcal{X}$. Suppose that $x_1, \ldots, x_m \sim \mathcal{D}$. Let $Q \in \mathcal{P}(\mathcal{Y} \times \mathcal{Z} \mid \mathcal{Y} \times \mathcal{Z})$ be a noisy channel from $h(x) = (y, z)$ to observations $\bar{y}, \bar{z}$. Let $C_Q = \max_{a,b} D_{\mathrm{KL}}\left(Q(\cdot \mid a) \| Q(\cdot \mid b)\right)$ be the capacity factor of the channel. The learner observes the noisy sample $S = \{(x_i, \bar{y}_i, \bar{z}_i)\}_{i=1}^m$. Define the pseudo-metric $d_{\mathcal{D}}^{\mathrm{e2e}}(h_1, h_2) = \mathbb{P}_x[h_1^{\mathrm{e2e}}(x) \neq h_2^{\mathrm{e2e}}(x)]$, and let $M(\varepsilon; \mathcal{H}, d_{\mathcal{D}}^{\mathrm{e2e}})$ be the $\varepsilon$-packing number of $\mathcal{H}$ with respect to this pseudo-metric. Then, for any algorithm $\mathcal{A}$ observing the CoT-supervised sample $S$ of size $m$, the probability of having large end-to-end error is lower bounded as*

$$\sup_{h_\star \in \mathcal{H}} \mathbb{P}_{S \sim P_{h_\star}^{\otimes m}} \left[ \mathcal{R}_{\mathcal{D}}^{\mathrm{e2e}}(\mathcal{A}(S)) \geq \frac{\varepsilon}{2} \right] \geq 1 - \frac{m \cdot C_Q \cdot \sup_\pi \mathbb{E}_{h_1, h_2 \sim \pi} \left[ \mathcal{I}_{\mathcal{D}}^{\mathrm{CoT}}(h_1, h_2) \right] + \log 2}{\log M(\mathcal{H}, d_{\mathcal{D}}^{\mathrm{e2e}}, \varepsilon)}.$$

*Proof.* Let $\mathcal{H}' := \{h_1, \ldots, h_M\} \subset \mathcal{H}$ be an $\varepsilon$-packing of $\mathcal{H}$ with respect to the end-to-end distance $d_{\mathcal{D}}^{\mathrm{e2e}}(h_1, h_2) = \mathbb{P}_{x \sim \mathcal{D}}[h_1^{\mathrm{e2e}}(x) \neq h_2^{\mathrm{e2e}}(x)]$, where $M = M(\varepsilon; \mathcal{H}, d_{\mathcal{D}}^{\mathrm{e2e}})$. Consider the prior distributed uniformly on this packing, $\pi = \mathrm{Unif}(\{h_1, \ldots, h_M\})$. For any learning algorithm $\mathcal{A} : (\mathcal{X} \times \mathcal{Y} \times \mathcal{Z})^* \to \mathcal{H}$, consider the modified algorithm $\mathcal{A}' : (\mathcal{X} \times \mathcal{Y} \times \mathcal{Z})^* \to \mathcal{H}'$ which projects $\mathcal{A}$ onto the packing $\mathcal{H}'$. That is,

$$\mathcal{A}'(S) := \arg\min_{h \in \mathcal{H}'} d_{\mathcal{D}}^{\mathrm{e2e}}(\mathcal{A}(S), h).$$

The test error of $\mathcal{A}'$ can be related to the test error of $\mathcal{A}$ via the geometry of $\mathcal{H}$ under the pseudometric $d_{\mathcal{D}}^{\mathrm{e2e}}$. In particular, letting $\hat{h} = \mathcal{A}(S)$ and $\widetilde{h} = \mathcal{A}'(S)$, we have that for all $h \in \mathcal{H}'$,

$$d_{\mathcal{D}}^{\mathrm{e2e}}(h, \widetilde{h}) \leq d_{\mathcal{D}}^{\mathrm{e2e}}(h, \hat{h}) + d_{\mathcal{D}}^{\mathrm{e2e}}(\hat{h}, \widetilde{h}) \leq 2\, d_{\mathcal{D}}^{\mathrm{e2e}}(h, \hat{h}),$$

where we used $d_{\mathcal{D}}^{\mathrm{e2e}}(\hat{h}, \widetilde{h}) \leq d_{\mathcal{D}}^{\mathrm{e2e}}(h, \hat{h}), \forall h \in \mathcal{H}'$, which follows by the definition of $\widetilde{h}$ as the projection of $\hat{h}$ onto $\mathcal{H}'$. Thus, we have that, for any $h \in \mathcal{H}', d_{\mathcal{D}}^{\mathrm{e2e}}(h, \widetilde{h}) \geq \varepsilon \implies d_{\mathcal{D}}^{\mathrm{e2e}}(h, \hat{h}) \geq \varepsilon/2$. Also, note that $h \neq \widetilde{h} \implies d_{\mathcal{D}}^{\mathrm{e2e}}(h, \widetilde{h}) \geq \varepsilon$. Thus, we have,

$$\mathbb{P}\left[h \neq \widetilde{h}\right] \leq \mathbb{P}\left[d_{\mathcal{D}}^{\mathrm{e2e}}(h, \widetilde{h}) \geq \varepsilon\right] \leq \mathbb{P}\left[d_{\mathcal{D}}^{\mathrm{e2e}}(h, \hat{h}) \geq \varepsilon/2\right].$$

By Fano's inequality, we can lower bound $\mathbb{P}[h \neq \widetilde{h}]$, which in turn implies the following lower bound on the probability of $\mathcal{A}$ having large end-to-end error,

$$\mathbb{P}_{h_\star \sim \pi, S \sim P_{h_\star}^{\otimes m}} \left[d_{\mathcal{D}}^{\mathrm{e2e}}(h_\star, \mathcal{A}(S)) \geq \varepsilon/2\right] \geq \mathbb{P}_{h_\star \sim \pi, S \sim P_{h_\star}^{\otimes m}} \left[d_{\mathcal{D}}^{\mathrm{e2e}}(h_\star, \mathcal{A}'(S)) \geq \varepsilon\right]$$

$$\geq \mathbb{P}_{h_\star \sim \pi, S \sim P_{h_\star}^{\otimes m}} \left[h_\star \neq \mathcal{A}'(S)\right]$$

$$\geq 1 - \frac{I(h_\star; S) + \log 2}{\log M(\varepsilon; \mathcal{H}, d_{\mathcal{D}}^{\mathrm{e2e}})}$$

$$\geq 1 - \frac{\sup_{\pi \in \mathcal{P}(\mathcal{H})} I(h_\star; S) + \log 2}{\log M(\varepsilon; \mathcal{H}, d_{\mathcal{D}}^{\mathrm{e2e}})}.$$

The first two inequalities are just restating the implications above, the third inequality is Fano's inequality, and the last inequality simply uses $I(h_\star; S) \leq \sup_\pi I(h_\star; S)$. That is, we bound the mutual information under the uniform prior over the packing, $\pi = \mathrm{Unif}(\mathcal{H}')$, by the supremum of the mutual information over all priors (i.e., the capacity).

Now, we compute the mutual information $I(h_\star; S)$ and relate it to the CoT information. First, let $\bar{P} = \mathbb{E}_{h \in \pi} [P_h]$ denote the mixture of $P_h$ under $\pi$, and note that

$$I(h_\star; S) = \mathbb{E}_{h \sim \pi} \left[ D_{\mathrm{KL}}\left( P_h^{\otimes m} \big\| \bar{P}^{\otimes m} \right) \right] \leq \mathbb{E}_{h_1, h_2 \sim \pi} \left[ D_{\mathrm{KL}}\left( P_{h_1}^{\otimes m} \big\| P_{h_2}^{\otimes m} \right) \right] = m \cdot \mathbb{E}_{h_1, h_2 \sim \pi} \left[ D_{\mathrm{KL}}\left( P_{h_1} \| P_{h_2} \right) \right],$$

where the inequality follows from the convexity of the KL divergence in the second argument and Jensen's inequality, and the last equality is by the chain rule for the KL divergence.

Now, let us compute the KL divergence between the distributions induced by a pair of hypotheses and relate it to the relative CoT information between them. For convenience, let us fold in the output into the CoT and use a bold $z$ to denote $z = (y, z)$.

$$
\begin{aligned}
D_{\mathrm{KL}}\left(P_{h_1} \| P_{h_2}\right) &= \underset{x,z\sim P_{h_1}}{\mathbb{E}}\left[\log \frac{P_{h_1}(x,z)}{P_{h_2}(x,z)}\right] = \underset{x,z\sim P_{h_1}}{\mathbb{E}}\left[\log \frac{\mathcal{D}(x)Q(z \mid h_1(x))}{\mathcal{D}(x)Q(z \mid h_2(x))}\right] \\
&= \underset{x\sim\mathcal{D}}{\mathbb{E}}\left[\sum_{z\in\mathcal{Y}\times\mathcal{Z}} Q(z \mid h_1(x)) \log \frac{Q(z \mid h_1(x))}{Q(z \mid h_2(x))}\right] \\
&= \underset{x\sim\mathcal{D}}{\mathbb{E}}\left[\mathbf{1}\{h_1(x) = h_2(x)\} \sum_{z\in\mathcal{Y}\times\mathcal{Z}} Q(z \mid h_1(x)) \log \frac{Q(z \mid h_1(x))}{Q(z \mid h_2(x))}\right] \\
&\quad + \underset{x\sim\mathcal{D}}{\mathbb{E}}\left[\mathbf{1}\{h_1(x) \neq h_2(x)\} \sum_{z\in\mathcal{Y}\times\mathcal{Z}} Q(z \mid h_1(x)) \log \frac{Q(z \mid h_1(x))}{Q(z \mid h_2(x))}\right] \\
&\overset{(a)}{=} \underset{x\sim\mathcal{D}}{\mathbb{E}}\left[\mathbf{1}\{h_1(x) \neq h_2(x)\} \sum_{z\in\mathcal{Y}\times\mathcal{Z}} Q(z \mid h_1(x)) \log \frac{Q(z \mid h_1(x))}{Q(z \mid h_2(x))}\right] \\
&\overset{(b)}{\leq} \underset{x\sim\mathcal{D}}{\mathbb{E}}\left[\mathbf{1}\{h_1(x) \neq h_2(x)\} \max_{z_1,z_2\in\mathcal{Y}\times\mathcal{Z}} \sum_{z\in\mathcal{Y}\times\mathcal{Z}} Q(z \mid z_1) \log \frac{Q(z \mid z_1)}{Q(z \mid z_2)}\right] \\
&\overset{(c)}{=} C_Q \cdot \underset{x\sim\mathcal{D}}{\mathbb{P}}[h_1(x) \neq h_2(x)] \\
&\overset{(d)}{\leq} C_Q \cdot -\log \underset{x\sim\mathcal{D}}{\mathbb{P}}[h_1(x) = h_2(x)] \\
&\overset{(d)}{=} C_Q \cdot \mathcal{I}_{\mathcal{D}}^{\mathrm{CoT}}(h_1, h_2).
\end{aligned}
$$

Step (a) follows by noting that $\sum_{z} Q(z \mid h_1(x)) \log \frac{Q(z) \mid h_1(x)}{Q(z \mid h_2(x))} = 0$ when $h_1(x) = h_2(x)$. Steps (b) and (c) are simply bounding the KL divergence between the observations under two different hypotheses that differ by the capacity of the channel, $C_Q := \max_{z_1,z_2} D_{\mathrm{KL}}\left(Q(\cdot \mid z_1) \| Q(\cdot \mid z_2)\right)$. Step (d) uses the identity $x \leq -\log(1 - x)$, and step (d) is the definition of the relative CoT information between two hypotheses.

Plugging this into the previous bound proves the result.

$$
\sup_{h_\star\in\mathcal{H}} \underset{S\sim P_{h_\star}^{\otimes m}}{\mathbb{P}}\left[\mathcal{R}_{\mathcal{D}}^{\mathrm{e2e}}(\mathcal{A}(S)) \geq \frac{\varepsilon}{2}\right] \geq \underset{h_\star\sim\pi, S\sim P_{h_\star}^{\otimes m}}{\mathbb{P}}\left[d_{\mathcal{D}}^{\mathrm{e2e}}(h_\star, \mathcal{A}(S)) \geq \varepsilon/2\right]
$$

$$
\geq 1 - \frac{m \cdot C_Q \cdot \sup_\pi \underset{h_1,h_2\sim\pi}{\mathbb{E}}\left[\mathcal{I}_{\mathcal{D}}^{\mathrm{CoT}}(h_1, h_2)\right] + \log 2}{\log M(\mathcal{H}, d_{\mathcal{D}}^{\mathrm{e2e}}, \varepsilon)}.
$$

$\square$

In particular, this result implies that when

$$
m \leq \frac{\log M(\mathcal{H}, d_{\mathcal{D}}^{\mathrm{e2e}}, \varepsilon)}{2 \cdot \left(C_Q \cdot \sup_\pi \underset{h_1,h_2\sim\pi}{\mathbb{E}}\left[\mathcal{I}_{\mathcal{D}}^{\mathrm{CoT}}(h_1, h_2)\right] + \log 2\right)},
$$

the probability that the error is more than $\varepsilon/2$ is at least $1/2$,

$$
\sup_{h_\star\in\mathcal{H}} \underset{S\sim P_{h_\star}^{\otimes m}}{\mathbb{P}}\left[\mathcal{R}_{\mathcal{D}}^{\mathrm{e2e}}(\mathcal{A}(S)) \geq \frac{\varepsilon}{2}\right] \geq \frac{1}{2}.
$$

Now, let us discuss the role of the noisy channel in the setting of this result. The noisy channel models noise in the learning process. For example, errors in human-created CoT annotations, errors

in the output labels, or any other type of noise. For simplicity, let us consider a symmetric channel over $\mathcal{Y} \times \mathcal{Z}$ parameterized by an error level $e$ and defined as

$$Q(\cdot \mid \boldsymbol{z}) = (1 - e) \cdot \delta_x + e \cdot \mathrm{Unif}(\mathcal{Y} \times \mathcal{Z}).$$

We can compute the channel capacity factor $C_Q$ for this channel. Let $\boldsymbol{x} \neq \boldsymbol{y}, \boldsymbol{x}, \boldsymbol{y} \in \mathcal{Y} \times \mathcal{Z}$ be two different symbols to be transmitted through the channel, and suppose the error level is non-zero, $e \in (0, 1]$. For convenience, denote $|\mathcal{Y} \times \mathcal{Z}| = N$.

$$D_{\mathrm{KL}} \left( Q(\cdot \mid \boldsymbol{x}) \| Q(\cdot \mid \boldsymbol{y}) \right)$$

$$= \sum_{\boldsymbol{z}} Q(\boldsymbol{z} \mid \boldsymbol{x}) \log \frac{Q(\boldsymbol{z} \mid \boldsymbol{x})}{Q(\boldsymbol{z} \mid \boldsymbol{y})}$$

$$= Q(\boldsymbol{x} \mid \boldsymbol{x}) \log \frac{Q(\boldsymbol{x} \mid \boldsymbol{x})}{Q(\boldsymbol{x} \mid \boldsymbol{y})} + Q(\boldsymbol{y} \mid \boldsymbol{x}) \log \frac{Q(\boldsymbol{y} \mid \boldsymbol{x})}{Q(\boldsymbol{y} \mid \boldsymbol{y})} + \sum_{\boldsymbol{z} \neq \boldsymbol{x}, \boldsymbol{z} \neq \boldsymbol{y}} Q(\boldsymbol{z} \mid \boldsymbol{x}) \log \frac{Q(\boldsymbol{z} \mid \boldsymbol{x})}{Q(\boldsymbol{z} \mid \boldsymbol{y})}$$

$$= \left( 1 - e + \frac{e}{N} \right) \log \frac{1 - e + e/N}{e/N} + \left( \frac{e}{N} \right) \log \frac{e/N}{1 - e + e/N} + \sum_{\boldsymbol{z} \neq \boldsymbol{x}, \boldsymbol{z} \neq \boldsymbol{y}} \left( \frac{e}{N} \right) \log \frac{e/N}{e/N}$$

$$= \left( 1 - e + \frac{e}{N} \right) \log \left( \frac{1 - e + e/N}{e/N} \right) - \left( \frac{e}{N} \right) \log \left( \frac{1 - e + e/N}{e/N} \right)$$

$$= (1 - e) \log \left( 1 + \frac{N(1 - e)}{e} \right)$$

Intuitively, this is a decreasing function in the error level $e$, decreasing towards 0 as $e \to 1$. This corresponds to the fact that it is harder to distinguish between hypotheses when the observations are more noisy. For $e = 1/100$ and $|\mathcal{Y} \times \mathcal{Z}| = 1,000$, the capacity factor is approximately $C_Q \approx 11.39$.

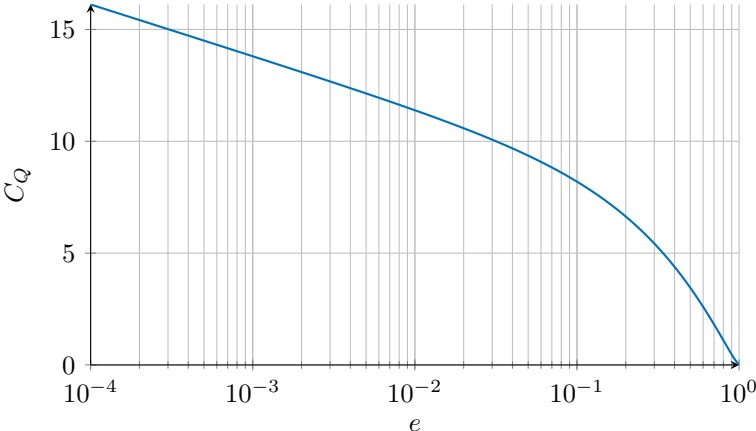

Figure 6: Capacity factor of $Q$ as a function of error level for $|\mathcal{Y} \times \mathcal{Z}| = 1,000$

# F    Other Topics

## F.1    Learning with Mixed Supervision

In many situations, CoT training examples may be difficult or expensive to obtain, for example, because they require manual human annotation. On the other hand, input-output examples without CoT annotation might be much more readily available. In such cases, one might have a dataset that includes a large number of end-to-end input-output examples and a small number of CoT-annotated examples. What types of learning guarantees can we establish in such a setting?

The following result, an extension of Result 1, analyzes the sample complexity of the consistency rule when applied to $m_{\mathrm{e2e}}$ end-to-end examples and $m_{\mathrm{CoT}}$ CoT examples.

**Result 6** (Learning with Mixed Datasets). *Let $\mathcal{H} \subset (\mathcal{Y} \times \mathcal{Z})^{\mathcal{X}}$ be a finite CoT hypothesis class, and let $\mathcal{D}$ be a distribution over $\mathcal{X}$. Consider an i.i.d dataset, $S = S_{\mathrm{e2e}} \cup S_{\mathrm{cot}}$, consisting of $m_{\mathrm{e2e}}$*

*input-output examples and $m_{\text{cot}}$ CoT-annotated examples*

$$S_{\text{e2e}} = \{(x_i^{\text{e2e}}, h_\star^{\text{e2e}}(x_i^{\text{e2e}}))\}_{i \in m_{\text{e2e}}}, S_{\text{cot}} = \{(x_i^{\text{e2e}}, h_\star^{\text{CoT}}(x_i^{\text{cot}}), h_\star^{\text{e2e}}(x_i^{\text{cot}}))\}_{i \in m_{\text{cot}}},$$

*where $x_1^{\text{e2e}}, \ldots, x_{m_{\text{e2e}}}^{\text{e2e}}, x_1^{\text{cot}}, \ldots, x_{m_{\text{cot}}}^{\text{cot}} \overset{i.i.d.}{\sim} \mathcal{D}$. Suppose the number of end-to-end examples is $\gamma$ times the number of CoT examples, so $m_{\text{e2e}} = \gamma \cdot m_{\text{cot}}$ and let $m_{\text{cot}} \equiv m$. Then, the $\text{e2e}-\text{CoT}$-consistency rule has sample complexity with respect to the end-to-end error of*

$$m(\varepsilon, \delta) = \frac{\log|\mathcal{H}| + \log(1/\delta)}{\gamma \cdot \varepsilon + \mathcal{I}_{\mathcal{D}, h_\star}^{\text{CoT}}(\varepsilon; \mathcal{H})}.$$

*That is, for $m \geq m(\varepsilon, \delta)$, with probability at least $1 - \delta$ over the draw of $S = S_{\text{e2e}} \cup S_{\text{cot}}$,*

$$\forall h \in \mathcal{H} \text{ such that } \widehat{\mathcal{R}}_S^{\text{e2e/CoT}}(h) = 0, \text{ we have } \mathcal{R}_{\mathcal{D}}^{\text{e2e}}(h) \leq \varepsilon.$$

*Proof.* We would like to bound the probability of the bad event

$$\{\exists h \in \mathcal{H} : \mathcal{R}_{\mathcal{D}}^{\text{e2e}}(h) > \varepsilon, \ \widehat{\mathcal{R}}_S^{\text{e2e/CoT}}(h) = 0\}$$

over $x_1^{\text{e2e}}, \ldots, x_{m_{\text{e2e}}}^{\text{e2e}}, x_1^{\text{cot}}, \ldots, x_{m_{\text{cot}}}^{\text{cot}} \overset{i.i.d.}{\sim} \mathcal{D}$. Fix any $h \in \mathcal{H}$ with end-to-end error $\mathcal{R}_{\mathcal{D}}^{\text{e2e}}(h) = \mathbb{P}_{x \sim \mathcal{D}}[h^{\text{e2e}}(x) \neq \text{e2e}(h_\star)(x)] > \varepsilon$ (i.e., $h \in \Delta_{\mathcal{D}}^{\text{e2e}}(\varepsilon; \mathcal{H}, h_\star)$). We bound the probability that $h$ is consistent with $S = S_{\text{e2e}} \cup S_{\text{cot}}$, $h \in \text{CoT-Cons}(S; \mathcal{H}) = \{h \in \mathcal{H} : \widehat{\mathcal{R}}_S^{\text{e2e/CoT}}(h) = 0\}$, as follows

$$\mathbb{P}_{S = S_{\text{e2e}} \cup S_{\text{cot}}}[h \in \text{CoT-Cons}(S; \mathcal{H})]$$

$$= \mathbb{P}_{S_{\text{e2e}}}\left[\forall i \in [m_{\text{e2e}}], \ h^{\text{e2e}}(x_i^{\text{e2e}}) = h_\star^{\text{e2e}}(x_i^{\text{e2e}})\right]$$

$$\cdot \mathbb{P}_{S_{\text{cot}}}\left[\forall i \in [m_{\text{cot}}], \ h^{\text{CoT}}(x_i^{\text{cot}}) = h_\star^{\text{CoT}}(x_i^{\text{cot}}), \ h^{\text{e2e}}(x_i^{\text{cot}}) = h_\star^{\text{e2e}}(x_i^{\text{cot}})\right]$$

$$= \mathbb{P}_{x \sim \mathcal{D}}\left[h^{\text{e2e}}(x_i) = h_\star^{\text{e2e}}(x)\right]^{m_{\text{e2e}}} \cdot \mathbb{P}_{x \sim \mathcal{D}}\left[h^{\text{CoT}}(x) = h_\star^{\text{CoT}}(x_i), \ h^{\text{e2e}}(x_i) = h_\star^{\text{e2e}}(x)\right]^{m_{\text{cot}}}$$

$$\overset{(a)}{\leq} (1 - \varepsilon)^{m_{\text{e2e}}} \cdot \left(\exp\left(-\mathcal{I}_{\mathcal{D}}^{\text{CoT}}(h_\star, h)\right)\right)^{m_{\text{cot}}}$$

$$\overset{(b)}{\leq} \exp\left(-m_{\text{e2e}} \cdot \varepsilon\right) \cdot \left(\exp\left(-\mathcal{I}_{\mathcal{D}}^{\text{CoT}}(h_\star, h)\right)\right)^{m_{\text{cot}}}$$

$$\overset{(c)}{\leq} \exp\left(-m_{\text{e2e}} \cdot \varepsilon\right) \cdot \exp\left(-m_{\text{cot}} \cdot \mathcal{I}_{\mathcal{D}, h_\star}^{\text{CoT}}(\varepsilon; \mathcal{H})\right)$$

$$= \exp\left(-m_{\text{e2e}} \cdot \varepsilon - m_{\text{cot}} \cdot \mathcal{I}_{\mathcal{D}, h_\star}^{\text{CoT}}(\varepsilon; \mathcal{H})\right),$$

where step (a) applies $\mathcal{R}_{\mathcal{D}}^{\text{e2e}}(h) > \varepsilon$ (since $h \in \Delta_{\mathcal{D}}^{\text{e2e}}(\varepsilon; \mathcal{H}, h_\star)$) for the first factor and uses the definition of $\mathcal{I}_{\mathcal{D}}^{\text{CoT}}(h_\star, h)$ for the second factor, step (b) is the identity $\log(1-x) \leq -x$ for $x \in (0, 1)$, and step (c) is by the definition of $\mathcal{I}_{\mathcal{D}, h_\star}^{\text{CoT}}(\varepsilon; \mathcal{H})$ and the fact that $h \in \Delta_{\mathcal{D}}^{\text{e2e}}(\varepsilon; \mathcal{H}, h_\star)$.

Now, we write $m_{\text{e2e}} = \gamma \cdot m$, $m_{\text{cot}} = m$, and set $m = \frac{\log|\mathcal{H}| + \log(1/\delta)}{\gamma \cdot \varepsilon + \mathcal{I}_{\mathcal{D}, h_\star}^{\text{CoT}}(\varepsilon; \mathcal{H})}$. This guarantees that the probability that any fixed hypothesis with error larger than $\varepsilon$ is consistent with $S = S_{\text{e2e}} \cup S_{\text{cot}}$ is at most

$$\mathbb{P}_{S = S_{\text{e2e}} \cup S_{\text{cot}}}[h \in \text{CoT-Cons}(S; \mathcal{H})] \leq \frac{\delta}{|\mathcal{H}|}.$$

Applying a union bound over $\mathcal{H}$ then shows that

$$\mathbb{P}_{S = S_{\text{e2e}} \cup S_{\text{cot}}}\left[\exists h \in \mathcal{H} : \mathcal{R}_{\mathcal{D}}^{\text{e2e}}(h) > \varepsilon, \ \widehat{\mathcal{R}}_S^{\text{e2e/CoT}}(h) = 0\right] \leq \delta.$$

$\square$

### F.2 CoT Learning with Inductive Priors

Encoding prior knowledge about solution structure is critical for learning complex functions, such as those representing multi-step reasoning processes, particularly from limited data. This is especially relevant in chain-of-thought settings, which are often applied to learning such reasoning processes.

The chain-of-thought trajectories can be viewed as additional supervision for the intermediate steps of an algorithm, helping to align the learner to the ground-truth algorithm, which may otherwise be very difficult to learn if only input-output examples are observed.

A key idea in learning algorithms is the so-called *Minimum Description Length* (MDL) principle. This encodes a prior or an inductive bias in the learner that favors hypotheses that have a small description length (e.g., thought of as the length of program code or the size of a Turing machine's state space). This is also related to the notion of algorithmic complexity in algorithmic information theory [42, 43].

In this section, we consider a minimum description length type of learning rule for the chain-of-thought setting. This also provides an extension of Result 1 to CoT hypothesis classes that are *countably infinite*.

For a prior $p$ over a hypothesis class $\mathcal{H}$, we define the *chain-of-thought* MDL rule corresponding to the prior $p$ as

$$\mathrm{MDL}_p^{\mathrm{CoT}}(S; \mathcal{H}) = \underset{h \in \texttt{CoT-Cons}(S;\mathcal{H})}{\arg\max} \; p(h). \tag{1}$$

That is, given a CoT dataset $S$, $\mathrm{MDL}_p^{\mathrm{CoT}}(S; \mathcal{H})$ selects the hypothesis that maximizes the prior $p$ among hypotheses that are CoT-consistent with $S$. One way to define such a prior $p$ over $\mathcal{H}$ is through a *prefix-free description language* $d : \mathcal{H} \to \{0,1\}$ which maps a hypothesis to a bitstring description. The prior $p$ can then be defined as $p(h) = 2^{-|d(h)|}$, in which case $\mathrm{MDL}_p^{\mathrm{CoT}}(S; \mathcal{H}) = \arg\min_{h \in \texttt{CoT-Cons}(S;\mathcal{H})} |d(h)|$ selects the minimum-description CoT-consistent hypothesis.

The following result provides a learning guarantee for $\mathrm{MDL}_p^{\mathrm{CoT}}$ in terms of the likelihood of $h_\star$ under the prior $p$ and the CoT-information metric $\mathcal{I}_{\mathcal{D}, h_\star}^{\mathrm{CoT}}(\varepsilon; \mathcal{H})$.

**Result 7** (Learning with Chain-of-Thought and MDL). *Let $\mathcal{H}$ be a countable autoregressive hypothesis class and consider a prior $p$ over $\mathcal{H}$. Let $S$ be an i.i.d. dataset of $m$ examples drawn from a distribution $\mathcal{D}$ over $\mathcal{X}$. Then, with probability at least $1\delta$ over the draw of $S$, any CoT-consistent hypothesis $h \in \texttt{CoT-Cons}(S; \mathcal{H})$ has its end-to-end error bounded by*

$$\mathcal{R}_{\mathcal{D}}^{\mathrm{e2e}}(h) \leq \varepsilon_h := \inf \left\{ \varepsilon > 0 : \mathcal{I}_{\mathcal{D}, h_\star}^{\mathrm{CoT}}(\varepsilon; \mathcal{H}) \geq \frac{\log(1/p(h)) + \log(1/\delta)}{m} \right\}$$

*This in turn implies a sample complexity with respect to the end-to-end error for the autoregressive MDL rule $\mathrm{MDL}_p^{\mathrm{CoT}}$ of*

$$m(\varepsilon, \delta) = \mathcal{O}\left( \frac{\log \frac{1}{p(h_\star)} + \log \frac{1}{\delta}}{\mathcal{I}_{\mathcal{D}, h_\star}^{\mathrm{CoT}}(\varepsilon; \mathcal{H})} \right).$$

*That is, for $m \geq m(\varepsilon, \delta)$, with probability at least $1 - \delta$ over $S = \{\boldsymbol{x}_1, \ldots, \boldsymbol{x}_m\} \overset{i.i.d.}{\sim} \mathcal{D}$,*

$$\forall h \in \mathrm{MDL}_p^{\mathrm{CoT}}(S; \mathcal{H}) \text{ we have } \mathcal{R}_{\mathcal{D}}^{\mathrm{e2e}}(h) \leq \varepsilon.$$

*Proof.* Following the argument in the proof of Result 1, for any fixed $h \in \mathcal{H}$ and any $\varepsilon_h > 0$, the probability that $h$ is CoT-consistent on $S$, $h \in \texttt{CoT-Cons}(S; \mathcal{H}) = \{h \in \mathcal{H} : \widehat{\mathcal{R}}_S^{\mathrm{CoT}}(h) = 0\}$ yet has end-to-end error $\mathcal{R}_{\mathcal{D}}^{\mathrm{e2e}}(h) > \varepsilon_h$, is bounded by

$$\underset{S \sim \mathcal{D}^{\otimes m}}{\mathbb{P}} \left[ \widehat{\mathcal{R}}_S^{\mathrm{CoT}}(h) = 0, \mathcal{R}_{\mathcal{D}}^{\mathrm{e2e}}(h) > \varepsilon_h \right] \leq \exp(-m \cdot \mathcal{I}_{\mathcal{D}}^{\mathrm{CoT}}(h_\star, h)) \leq \exp(-m \cdot \mathcal{I}_{\mathcal{D}, h_\star}^{\mathrm{CoT}}(\varepsilon_h; \mathcal{H})).$$

For each $h \in \mathcal{H}$, we will target an end-to-end error $\varepsilon_h$ in a prior-dependent manner such that

$$\underset{S \sim \mathcal{D}^{\otimes m}}{\mathbb{P}} \left[ \widehat{\mathcal{R}}_S^{\mathrm{CoT}}(h) = 0, \mathcal{R}_{\mathcal{D}}^{\mathrm{e2e}}(h) > \varepsilon_h \right] \leq \delta p(h).$$

This occurs if $\varepsilon_h$ is chosen such that

$$\exp(-m \cdot \mathcal{I}_{\mathcal{D}, h_\star}^{\mathrm{CoT}}(\varepsilon_h; \mathcal{H})) \leq \delta p(h)$$

$$\iff \mathcal{I}_{\mathcal{D}, h_\star}^{\mathrm{CoT}}(\varepsilon_h; \mathcal{H}) \geq \frac{\log(1/p(h)) + \log(1/\delta)}{m}.$$

Thus, we define the target error $\varepsilon_h$ for $h$ as

$$\varepsilon_h := \inf\left\{\varepsilon > 0 : \mathcal{I}_{\mathcal{D},h_\star}^{\mathrm{CoT}}(\varepsilon;\mathcal{H}) \geq \frac{\log(1/p(h)) + \log(1/\delta)}{m}\right\}.$$

Note that this can be viewed in terms of the generalized inverse $(\mathcal{I}_{\mathcal{D},h_\star}^{\mathrm{CoT}}(\cdot;\mathcal{H}))^-$, where the generalized inverse of a function $f$ is defined as $f^-(x) = \inf\{y : f(y) > x\}$. This is well-defined since $\mathcal{I}_{\mathcal{D},h_\star}^{\mathrm{CoT}}(\cdot;\mathcal{H})$ is an increasing function by Lemma 1.

Now, by a union bound, we have that the probability that *any* CoT-consistent hypothesis exceeds its target end-to-end error is bounded by

$$\mathop{\mathbb{P}}_{S\sim\mathcal{D}^{\otimes m}}\left[\exists h \in \mathcal{H} : \widehat{\mathcal{R}}_S^{\mathrm{CoT}}(h) = 0, \mathcal{R}_{\mathcal{D}}^{\mathrm{e2e}}(h) > \varepsilon_h\right]$$

$$\leq \sum_{h\in\mathcal{H}} \mathop{\mathbb{P}}_S\left[h \in \mathtt{CoT\text{-}Cons}(S;\mathcal{H}), \mathcal{R}_{\mathcal{D}}^{\mathrm{e2e}}(h) > \varepsilon_h\right]$$

$$\leq \sum_{h\in\mathcal{H}} \delta p(h) = \delta.$$

Since $h_\star \in \mathtt{CoT\text{-}Cons}(S;\mathcal{H})$, we have that $p(h_\star) \leq p(\mathrm{MDL}_p^{\mathrm{CoT}}(S;\mathcal{H}))$ by definition of $\mathrm{MDL}_p^{\mathrm{CoT}}$. This, in turn, implies that

$$\mathcal{R}_{\mathcal{D}}^{\mathrm{e2e}}(\mathrm{MDL}_p^{\mathrm{CoT}}(S;\mathcal{H})) \leq \inf\left\{\varepsilon > 0 : \mathcal{I}_{\mathcal{D},h_\star}^{\mathrm{CoT}}(\varepsilon;\mathcal{H}) \geq \frac{\log(1/p(h_\star)) + \log(1/\delta)}{m}\right\},$$

with probability at least $1 - \delta$. Noting the property that $f(x) \geq y \implies x \geq f^-(y)$ for generalized inverses, we obtain a sample complexity of

$$m(\varepsilon,\delta) = \mathcal{O}\left(\frac{\log\frac{1}{p(h_\star)} + \log\frac{1}{\delta}}{\mathcal{I}_{\mathcal{D},h_\star}^{\mathrm{CoT}}(\varepsilon;\mathcal{H})}\right).$$

$\square$

**Corollary 1.** *If the prior $p$ is defined as $p(h) = 2^{-|d(h)|}$ in terms of a prefix-free description language $d : \mathcal{H} \to \{0,1\}^*$ satisfying Kraft's inequality $\sum_h 2^{-|d(h)|} \leq 1$, then the sample complexity of autoregressive MDL rule $\mathrm{MDL}_p^{\mathrm{CoT}}$ satisfies*

$$m(\varepsilon,\delta) = \mathcal{O}\left(\frac{|d(h_\star)| + \log\frac{1}{\delta}}{\mathcal{I}_{\mathcal{D},h_\star}^{\mathrm{CoT}}(\varepsilon;\mathcal{H})}\right).$$

One advantage of such MDL-style analysis is obtaining a sample complexity that is instance-dependent (i.e., $h_\star$-dependent). In the standard end-to-end setting, we obtain a sample complexity of $\mathcal{O}(\log(1/p(h_\star))/\varepsilon)$. In the CoT setting, we obtain an instance-dependent description of the sample complexity in both the numerator and denominator through the CoT information $\mathcal{I}_{\mathcal{D},h_\star}^{\mathrm{CoT}}(\varepsilon;\mathcal{H})$.

### F.3 Transfer Learning & Out-of-Distribution Generalization Under CoT Supervision

In many applications where chain-of-thought learning is applied, *out-of-distribution generalization* is a key aspect. This type of generalization requires compositional reasoning abilities, for example, learning a set of generally applicable atomic skills that can be recombined to generalize systematically to novel combinations of known elements. Of particular interest is learning from simple problem instances and generalizing to larger and more complex instances. This is sometimes referred to as *length-generalization* when the notion of increased complexity corresponds to longer inputs.

Chain-of-thought learning has important implications for this type of generalization. In particular, it allows direct supervision on the "atomic skills" and how to combine them to solve problems. This can, in principle, enable systematic generalization by transforming the learning problem from one of learning input-output patterns to one of learning general principles that can be applied beyond the training distribution.

In this section, we explore how the CoT information measure can be extended to analyze generalization under distribution shift.

The following generalized definition of CoT information captures the amount of information revealed about the end-to-end behavior on a test distribution $\mathcal{D}_{\text{test}}$ from observing a CoT-annotated sample drawn from a training distribution $\mathcal{D}_{\text{tr}}$.

**Definition 2** (Relative CoT Information Between a Pair of Distributions). *For a CoT hypothesis class $\mathcal{H} \subset (\mathcal{Y} \times \mathcal{Z})^{\mathcal{X}}$, we define the relative CoT-Information between a distribution $\mathcal{D}_{\text{tr}}$ and $\mathcal{D}_{\text{test}}$ as*

$$\mathcal{I}^{\text{CoT}}_{\mathcal{D}_{\text{tr}} \to \mathcal{D}_{\text{test}}}(\varepsilon; \mathcal{H}, h_\star) = \inf_{h \in \Delta^{\text{e2e}}_{\mathcal{D}_{\text{test}}}(\varepsilon; \mathcal{H}, h_\star)} \left\{ - \log_{x \sim \mathcal{D}_{\text{tr}}} \mathbb{P} \left[ h^{\text{CoT}}(x) = h_\star^{\text{CoT}}(x), h^{\text{e2e}}(x) = h_\star^{\text{e2e}}(x) \right] \right\}.$$

*where the infimum is over $\Delta^{\text{e2e}}_{\mathcal{D}_{\text{test}}}(\varepsilon; \mathcal{H}, h_\star)$, the set of hypotheses that disagree with $h_\star$'s end-to-end behavior (i.e., output) on the test distribution with probability greater than $\varepsilon$,*

$$\Delta^{\text{e2e}}_{\mathcal{D}_{\text{test}}}(\varepsilon; \mathcal{H}, h_\star) := \left\{ h \in \mathcal{H} : \mathbb{P}_{x \sim \mathcal{D}_{\text{test}}} \left[ h_\star^{\text{e2e}}(x) \neq h^{\text{e2e}}(x) \right] > \varepsilon \right\}.$$

Unlike the in-distribution setting (where $\mathcal{D}_{\text{test}} = \mathcal{D}_{\text{tr}}$), $\mathcal{I}^{\text{CoT}}_{\mathcal{D}_{\text{tr}} \to \mathcal{D}_{\text{test}}}(\varepsilon; \mathcal{H}, h_\star) \geq \varepsilon$ does not necessarily hold for arbitrary CoT hypothesis classes and pairs of distributions. However, we do have

$$\mathcal{I}^{\text{CoT}}_{\mathcal{D}_{\text{tr}} \to \mathcal{D}_{\text{test}}}(\varepsilon; \mathcal{H}, h_\star) \geq \inf_{h \in \Delta^{\text{e2e}}_{\mathcal{D}_{\text{test}}}(\varepsilon; \mathcal{H}, h_\star)} \left\{ \mathcal{R}^{\text{e2e}}_{\mathcal{D}_{\text{tr}}}(h) \right\}.$$

With strong CoT supervision on a sufficiently diverse training distribution, we would expect the relative CoT information to be large.

Analogous results to those presented in the main text in terms of the (standard) CoT information also apply in the transfer learning setting via the relative CoT information measure between two distributions defined above. For example, in the finite hypothesis class case, we have the following analogue of Result 1.

**Result 8** (Transfer Learning with Chain-of-Thought Supervision). *For any finite CoT class $\mathcal{H}$ and distributions $\mathcal{D}_{\text{tr}}$ (training) and $\mathcal{D}_{\text{test}}$ (test) over $\mathcal{X}$, the CoT consistency rule has sample complexity with respect to the $\mathcal{D}_{\text{test}}$-end-to-end error of*

$$m(\varepsilon, \delta) = \frac{\log |\mathcal{H}| + \log(1/\delta)}{\mathcal{I}^{\text{CoT}}_{\mathcal{D}_{\text{tr}} \to \mathcal{D}_{\text{test}}}(\varepsilon; \mathcal{H}, h_\star)}.$$

*That is, for $m \geq m(\varepsilon, \delta)$, we have that with probability at least $1 - \delta$ over $S = \{x_1, \ldots, x_m\} \overset{i.i.d.}{\sim} \mathcal{D}_{\text{tr}}$,*

$$\forall h \in \texttt{CoT-Cons}(S; \mathcal{H}), \text{ we have } \mathcal{R}^{\text{e2e}}_{\mathcal{D}_{\text{test}}}(h) \leq \varepsilon.$$

*Proof.* Fix any $h \in \mathcal{H}$ with end-to-end error under $\mathcal{D}_{\text{test}}$ larger than $\varepsilon$, i.e., $\mathcal{R}^{\text{e2e}}_{\mathcal{D}_{\text{test}}}(h) > \varepsilon$ (so $h \in \Delta^{\text{e2e}}_{\mathcal{D}_{\text{test}}}(\varepsilon; \mathcal{H}, h_\star)$). We bound the probability that $h$ is CoT-consistent on $S$, $h \in \texttt{CoT-Cons}(S; \mathcal{H}) = \{h \in \mathcal{H} : \widehat{\mathcal{R}}^{\text{CoT}}_S(h) = 0\}$, as follows

$$\mathbb{P}_{S \sim \mathcal{D}_{\text{tr}}^{\otimes m}} \left[ h \in \texttt{CoT-Cons}(S; \mathcal{H}) \right] = \mathbb{P}_{S \sim \mathcal{D}_{\text{tr}}^{\otimes m}} \left[ \forall i, h^{\text{CoT}}(x_i) = h_\star^{\text{CoT}}(x_i), h^{\text{e2e}}(x_i) = h_\star^{\text{e2e}}(x_i) \right]$$

$$= \mathbb{P}_{x \sim \mathcal{D}_{\text{tr}}} \left[ h^{\text{CoT}}(x) = h_\star^{\text{CoT}}(x_i), h^{\text{e2e}}(x_i) = h_\star^{\text{e2e}}(x) \right]^m$$

$$\leq \exp \left( -m \cdot \mathcal{I}^{\text{CoT}}_{\mathcal{D}_{\text{tr}} \to \mathcal{D}_{\text{test}}}(\varepsilon; \mathcal{H}, h_\star) \right),$$

where we use the definition of the relative CoT information and the fact that $h \in \Delta^{\text{e2e}}_{\mathcal{D}_{\text{test}}}(\varepsilon; \mathcal{H}, h_\star)$.

Choosing $m \geq m(\varepsilon, \delta)$ in the theorem statement guarantees that

$$\mathbb{P}_{S \sim \mathcal{D}_{\text{tr}}^{\otimes m}} \left[ \exists h \in \mathcal{H} : \mathcal{R}^{\text{e2e}}_{\mathcal{D}_{\text{test}}}(h) > \varepsilon, \widehat{\mathcal{R}}^{\text{CoT}}_S(h) = 0 \right] \leq \delta.$$

$\square$

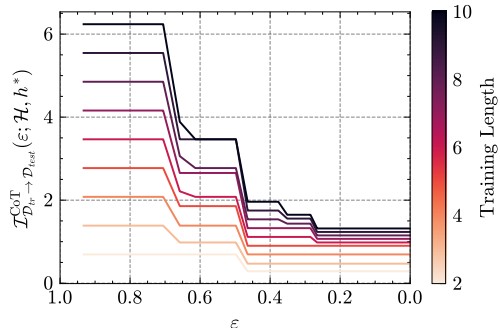
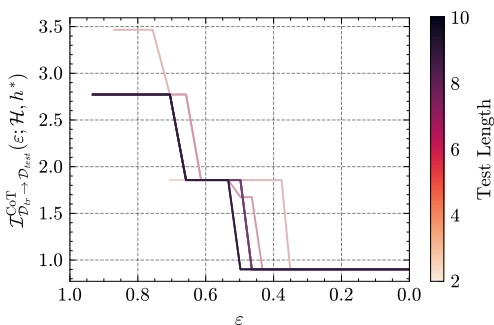

(a) $\mathcal{I}^{\mathrm{CoT}}_{\mathcal{D}_{\mathrm{tr}}\to\mathcal{D}_{\mathrm{test}}}(\varepsilon;\mathcal{H},h_\star)$ with fixed test distribution at length $L = 5$, and varying the training distribution.

(b) $\mathcal{I}^{\mathrm{CoT}}_{\mathcal{D}_{\mathrm{tr}}\to\mathcal{D}_{\mathrm{test}}}(\varepsilon;\mathcal{H},h_\star)$ with fixed training distribution at length $L = 5$, and varying the test distribution.

Figure 7: Simulations exploring $\mathcal{I}^{\mathrm{CoT}}_{\mathcal{D}_{\mathrm{tr}}\to\mathcal{D}_{\mathrm{test}}}(\varepsilon;\mathcal{H},h_\star)$

An analogous result also holds for the infinite hypothesis class setting.

**Example 7** (Length-Generalization in Finite-State Machines). Example 4 shows that

$$\min_{\varepsilon>0}\mathcal{I}^{\mathrm{CoT}}_{\mathcal{D},h_\star}(\varepsilon;\mathcal{H}) \geq |\Sigma|^{-(\ell+1)}$$

for the class of finite-state machines when $h_\star$'s transition graph is $\ell$-connected and $\mathcal{D}$ is a uniform distribution over inputs of length $n \geq \ell$. In fact, the same line of reasoning shows that

$$\min_{\varepsilon>0}\mathcal{I}^{\mathrm{CoT}}_{\mathcal{D}_{\mathrm{tr}}\to\mathcal{D}_{\mathrm{test}}}(\varepsilon;\mathcal{H},h_\star) \geq |\Sigma|^{-(\ell+1)}$$

for any distribution $\mathcal{D}_{\mathrm{test}}$ (i.e., of arbitrary length). The chain-of-thought annotations allow each component of the FSM's transition function to be identified and hence enable generalization to arbitrary test distributions.

Figure 7 depicts simulation results with the DFA example presented in Appendix C, exploring the relative CoT information between a pair of distributions, $\mathcal{I}^{\mathrm{CoT}}_{\mathcal{D}_{\mathrm{tr}}\to\mathcal{D}_{\mathrm{test}}}(\varepsilon;\mathcal{H},h_\star)$ as defined above. Figure 7a depicts the CoT information for a fixed test-distribution, varying the input length in the training distribution, showing that the CoT information curves are increasing with the input length. This suggests that longer, more complex inputs reveal more information about the underlying hypothesis and its input-output behavior. On the other hand, Figure 7b depicts the CoT information for fixed training distribution, with input length $L = 5$, varying the test distribution. We see that the CoT information remains relatively large, even for longer and more complex test inputs. This suggests that, under CoT supervision, the observations reveal enough about the hypothesis to identify its behavior beyond the training distribution.

