# OpenReview forum: "CoT Information: Improved Sample Complexity under Chain-of-Thought Supervision"
_NeurIPS.cc/2025/Conference — NeurIPS 2025 spotlight_

### Official Review · Reviewer_T4Cn · 2025-06-30

**Clarity:** 2
**Significance:** 2
**Originality:** 2
**Rating:** 4
**Confidence:** 2

**Summary:**

This paper focuses on the statistical theory of learning under CoT supervision. It introduces an abstract model of CoT supervised learning, a CoT information measure, covering both finite-cardinality hypothesis classes and infinite hypothesis classes, as well as the agnostic setting. Two types of information-theoretic lower bounds are established, along with their corresponding upper bounds.

**Questions:**

- The paper models CoT supervision z as an intermediate variable between input x and output y, which is a relatively general modeling approach. Are the derived conclusions applicable to general problems with intermediate supervision? Are there more specific modeling methods for CoT itself?

**Ethical Concerns:**

["NO or VERY MINOR ethics concerns only"]

**Final Justification:**

Thanks for the response, my main concerns have been addressed.

**Limitations:**

Yes

**Quality:**

2

**Strengths And Weaknesses:**

Strengths
- S1: The paper addresses a very important problem in CoT theoretical research, providing a detailed analysis of sample complexity under specific CoT problems, which contributes to understanding the underlying mechanisms of CoT.
- S2: The paper provides detailed derivation proofs, covering cases from simple to complex.
Weaknesses
- W1: The paper lacks a specific analysis of its differences from related work, making it difficult to effectively gauge its novelty. It's recommended to add a detailed comparative analysis with existing work and a specific description of the paper's innovative points.
- W2: There's a lack of discussion on the practical applications of the paper's theory, despite some application-oriented theoretical analysis being added in Section 6. Adding insights on how the theory guides practical applications or points of application for solving real-world problems would significantly increase the paper's utility.
- W3: Possibly due to it being a theoretical paper, it contains a large number of symbols, many of which lack detailed explanations (e.g., line 31). This makes it less user-friendly, especially for readers unfamiliar with the relevant theories.

---

> ### Author Rebuttal · Authors · 2025-07-30
>
> We thank the reviewer for their thoughtful and constructive feedback. We are especially grateful for the recognition of the importance of our work in advancing the theoretical understanding of chain-of-thought supervision (S1), as well as the acknowledgment of the rigor and completeness of our proofs across a range of settings (S2). We appreciate the opportunity to clarify the relationship of our contributions to prior work, further articulate their practical implications, and improve the accessibility of our exposition. Below, we address each of the reviewer's concerns in detail.
>
> ---
>
> **W1: Connection to related work**
>
> > W1: The paper lacks a specific analysis of its differences from related work, making it difficult to effectively gauge its novelty. It's recommended to add a detailed comparative analysis with existing work and a specific description of the paper's innovative points.
>
> We'd like to point you to the related work section within the discussion (Section 6.2), which discusses related work and positions our contributions within the literature. We'd also like to point you to Section 2.1 and Table 1, where we motivate how our analytical approach differs from prior approaches.
>
> The most closely related pieces of work are Malach (2024) [citation 16] and Joshi et al. (2025) [citation 10].
> - Malach (2024): This work studies the problem of learning autoregressive next-token prediction on CoT datasets. The core idea of this work is to express a CoT function as a composition of $T$ (a fixed number) different sequence-domain functions and learns each function independently. Since the function at each step is assumed to be independent, this enables the direct application of standard PAC learning results independently at each step. While this approach simplifies the analysis, its assumption of independently learned functions at each iteration is a notable limitation, which does not accurately reflect real-world settings and can limit sample efficiency gains from CoT supervision.
> - More recently, Joshi et al. (2025) studied a modification of Malach (2024)'s setting where the composition of sequence-domain functions is assumed to be time-invariant. Their analysis relies on bounding the *CoT error* (rather than end-to-end error) using standard PAC learning tools based on the VC dimension of the CoT loss class, noting that the CoT error provides an upper bound on the end-to-end error. The main contribution of Joshi et al. (2025) is bounding the VC dimension of the CoT loss class in terms of $T$ and the VC dimension of the iterated sequence-domain function class.
>
> In our work, we take a different analytical approach, focusing on quantifying the information content per CoT-supervised sample. In particular, this appears through the $\epsilon$-dependence of the rate, replacing the $1/\epsilon$ rate of Malach (2024) and Joshi et al. (2025) with the potentially much faster $1/\mathcal{I}^{\mathrm{CoT}}(\epsilon)$ rate.
>
> In addition to the above discussion, we will use the additional content page allowed in the final version of the paper to expand on the discussion of the related work.
>
> ---
>
> **W2: Discussion of practical applications of the theory developed in the paper**
>
> > W2: There's a lack of discussion on the practical applications of the paper's theory, despite some application-oriented theoretical analysis being added in Section 6. Adding insights on how the theory guides practical applications or points of application for solving real-world problems would significantly increase the paper's utility.
>
> While the main goal of this work is to develop a theoretical understanding of the statistical aspects of CoT-supervised learning, we agree on the importance of including a discussion of the practical applications and implications of this theory. Below, we will summarize and point to sections of the paper that pertain to these practical considerations, some of which were deferred to the appendix due to space constraints.
>
> - In Figure 2 in the main paper and Appendix C, we present simulation results where we 1) explicitly compute the CoT information, and 2) empirically evaluate the sample complexity of learning with and without CoT supervision. We show that our theory accurately predicts the gains in sample complexity.
> - Related to this question, we would like to point you to our response to review `zRps` where we discuss further extensions that enable the extension of such empirical exploration to more complex parametric model classes such as neural networks (e.g., Transformers).
> - In Appendix F.1, we study the problem of learning with mixed end-to-end and CoT supervision. This exploration is directly motivated by practical considerations: namely,  the fact that obtaining CoT-annotated examples is often costly and labor-intensive, which limits their availability. Thus, in certain practical scenarios, one might have a large number of end-to-end annotated samples together with a small number of CoT-annotated samples. The results of Appendix F.1 show that our theoretical framework allows us to analyze such scenarios.
> - In Appendix F.3, we study the important practical problem of distribution-shift and out-of-distribution generalization. We present an extension of the framework described in the main paper that enables the analysis of these problems. Our results show that CoT-supervision can enable greater domain adaptation and out-of-distribution generalization compared to end-to-end supervision. This provides a possible theoretical explanation to empirical observations on CoT-supervision improving OOD generalization (e.g., Stechly, Valmeekam, and Kambhampati, 2024; Y. Zhou et al., 2024)
>
> To address your concern, we will include an expanded discussion of practical applications and implications in the final version of the paper.
>
> ---
>
> **W3: Large number of symbols**
>
> > W3: Possibly due to it being a theoretical paper, it contains a large number of symbols, many of which lack detailed explanations (e.g., line 31). This makes it less user-friendly, especially for readers unfamiliar with the relevant theories.
>
> Indeed, due to being a theoretical paper tackling a newly developed area of analysis within statistical learning theory, some new notation and symbols are necessary. That said, we made a conscious effort to choose notation that supports clarity and readability, and we welcome any suggestions you may have for improvement.
>
> Can you clarify what you mean by your comment on line 31? We would be happy to make refinements based on your feedback. This line in the introduction introduces the central quantity in this work, the "Chain-of-Thought Information", which characterizes the statistical complexity of CoT-supervised learning. As this is the introduction section, our intention here is to merely give a *preview* of the main results to be developed in the paper rather than being a formal description. The formal description is provided in Definition 1 within Section 3.
>
> We will make refinements to the presentation to include more exposition and explanation of different notation as it is introduced, as well as to provide intuition and recall/reference definitions where appropriate.
>
> ---
>
> Thank you again for your review and helpful feedback. We hope we were able to address your main concerns and look forward to further discussion.

---

> > ### Comment · Reviewer_T4Cn · 2025-08-04
> >
> > Thanks for you response. My main concerns have been addressed and I raised my score.

---

> > > ### Author Response · Authors · 2025-08-06
> > >
> > > Thank you for your engagement in the review process! We are glad we were able to address your main concerns, and we appreciate you raising your score.
> > >
> > > We've noticed that the updated score isn't showing up on the portal yet. Would you mind checking to ensure the change went through on your end? It's possible there was a glitch in the system.

---

### Official Review · Reviewer_QDtR · 2025-07-03

**Clarity:** 3
**Significance:** 4
**Originality:** 3
**Rating:** 5
**Confidence:** 2

**Summary:**

This paper develops a statistical learning theory framework to analyse the benefits of CoT supervision, introducing a quantity called **CoT information** that quantifies how much additional signal CoT annotations provide for distinguishing correct from incorrect models. The authors show that, unlike standard supervision, which minimises end-to-end loss, CoT-supervised training minimises a richer loss that includes both intermediate reasoning steps and final answers, leading to potentially much lower sample complexity. Through both upper and lower bounds, the paper demonstrates that informative CoT traces can significantly accelerate learning and provide theoretical guarantees in both realisable and agnostic settings.

**Questions:**

1. The paper shows that an "informative" CoT is beneficial. This raises a practical question: what structural properties can make a CoT informative? Have the authors investigate this?
2. In the agnostic setting, the authors admit that CoT supervision can hurt if the hypothesis class poorly fits the CoT traces. How practical is it to design hypothesis classes that jointly fit outputs and reasoning traces?
3. Do the bounds extend to stochastic CoT annotations, such as those from RLHF-annotated intermediate steps or from uncertain model generations?

**Ethical Concerns:**

["NO or VERY MINOR ethics concerns only"]

**Final Justification:**

My concerns have been addressed, and I would like to maintain my original score.

**Limitations:**

See Weakness.

**Quality:**

4

**Strengths And Weaknesses:**

Strengths:
- The paper is exceptionally clear and well-organised.
- CoT prompting and fine-tuning can enhance the LLM performance significantly through empirical results [1,2]; however, it is still unknown why CoT can lead to the performance gain. This paper tried to theoretically understand an important but challenging problem, which is significant.

Weaknesses
- The primary weakness is that the empirical validation, while effective for illustrating the theory, is confined to symbolic tasks (e.g., learning finite automata). While this is a standard practice for theoretical papers, the work would have been even more compelling if it included a small-scale experiment with a neural model (e.g., a small transformer). Such an experiment would help bridge the gap between the theoretical framework and current LLM practices, showcasing how these insights apply to the very models that motivated the research.

Note: I can't validate the theoretical correctness as my primary expertise is not in statistical learning theory. :)

[1] Wei, J., Wang, X., Schuurmans, D., Bosma, M., Xia, F., Chi, E., ... & Zhou, D. (2022). Chain-of-thought prompting elicits reasoning in large language models. Advances in neural information processing systems, 35, 24824-24837.
[2] Shao, Z., Wang, P., Zhu, Q., Xu, R., Song, J., Bi, X., ... & Guo, D. (2024). Deepseekmath: Pushing the limits of mathematical reasoning in open language models. arXiv preprint arXiv:2402.03300.

---

> ### Author Rebuttal · Authors · 2025-07-30
>
> Thank you very much for your thoughtful and constructive review. We appreciate your positive feedback on the presentation and clarity of the paper, as well as the importance of the theoretical contributions to understanding the emerging CoT paradigm.
>
> ---
>
> **Questions**
>
> Thank you for your interesting questions.
>
> > The paper shows that an "informative" CoT is beneficial. This raises a practical question: what structural properties can make a CoT informative? Have the authors investigate this?
>
> Intuitively, the CoT is informative when observing it helps to identify the underlying end-to-end input-output relationship. The exact properties needed for this to hold depend on the underlying task and hypothesis class. For example, in the case of identifying finite state automata (see Fig 2 and Appendix C), the CoT corresponds to observing the state trajectory of the automata, which helps to identify its transition function, ultimately identifying the end-to-end behavior at a much more rapid rate. However, this precise rate can depend on properties of the transition function of $h_\star$ (e.g., a more highly-connected transition graph can be learned more quickly).
>
> > In the agnostic setting, the authors admit that CoT supervision can hurt if the hypothesis class poorly fits the CoT traces. How practical is it to design hypothesis classes that jointly fit outputs and reasoning traces?
>
> The reason that CoT supervision can be harmful in the agnostic setting (at least for CoT-ERM) is that the agnostic setting makes no assumptions whatsoever about the data and whether it aligns at all with the hypothesis class. Thus, it is not surprising that CoT supervision is not necessarily helpful. For example, in the general case, the agnostic setting does not preclude extreme cases where the hypothesis class perfectly captures the end-to-end function but cannot express the CoT trajectory at all. Thus, in order to make sense of the agnostic setting in CoT-supervised learning, we must make structural assumptions that quantify how well-aligned the hypothesis class is to the data distribution in terms of both the CoT and the final output. This is captured by Result 3.
>
> Intuitively, to design hypothesis classes that jointly fit the outputs and reasoning traces, the reasoning traces must actually correspond to ``intermediate steps'' in the underlying model. In other words, models within the hypothesis class that fit the reasoning traces are better at predicting the final output than models that do not fit the reasoning traces.
>
> > Do the bounds extend to stochastic CoT annotations, such as those from RLHF-annotated intermediate steps or from uncertain model generations?
>
> As stated, the results in the paper mainly apply to the deterministic setting. This is partly because we use a PAC-like framework, where hypothesis classes and function outputs are typically deterministic. However, we believe that analogous results would hold in the stochastic setting. A hint towards this is the information-theoretic analysis in Section 5 and Appendix E, where hypotheses in the CoT hypothesis class are stochastic.

---

> > ### Comment · Reviewer_QDtR · 2025-08-03
> >
> > Dear Authors,
> >
> > Thank you for your response. My concerns have been addressed, and I would like to maintain my original score.
> >
> > Best of luck!
> >
> > Kind regards,
> > Reviewer QDtR

---

### Official Review · Reviewer_zRps · 2025-07-06

**Clarity:** 4
**Significance:** 3
**Originality:** 3
**Rating:** 5
**Confidence:** 2

**Summary:**

This paper proposes a theoretical framework for understanding the statistical benefits of chain-of-thought (cot) supervision. The authors introduce a the concept of cot Information, which quantifies how cot annotations can help distinguish between different hypothese classes more effectively than standard input-output supervision. They show that cot supervision can significantly reduce sample complexity, sometimes requiring far fewer examples to reach the same accuracy. The paper provides both tight upper and lower bounds using this newly proposed information measure. Theoretical results are extended to agnostic settings and infinite hypothesis classes. The authors also include case studies, such as regular languages with state trajectories as cot , where cot supervision yields up to 600x gains in sample efficiency. The work positions itself as going beyond prior studies that focused on VC dimension or structural assumptions without explicitly modeling the information gain from chain-of-thought.

**Questions:**

I thank the authors for their novel contribution. Cot information measure is central to the paper, but for much of the non-theory community, it’s unclear how one could estimate or approximate it in practice. If we have a dataset with cot annotations, is there a way to tell whether the cots are informative under your framework? Even a heuristic or proxy would help bridge the gap between the theory and practical applications.

**Ethical Concerns:**

["NO or VERY MINOR ethics concerns only"]

**Limitations:**

The authors addressed some of the work's limitation in the conclusion section.

**Paper Formatting Concerns:**

N/A; This paper seems correct in format to my eyes.

**Quality:**

4

**Strengths And Weaknesses:**

Strength: To my eyes, the core idea of cot information proposed in the paper is novel and interesting. The paper presents a new perspective to measure how helpful chain-of-thought supervision can be for machine learning. The idea of cot information captures how much more easily a model can learn when given reasoning steps, not just final answers. To my eyes, the theoretical results are also correctly constructed to show cot can need far fewer examples, and this is true in both clean and noisy data settings. The idea that cot supervision helps learning because it adds information per example is well motivated and explained. The paper also gives clear examples and simulations that match the theory and make the results easier to understand. The overall presentation is clear and careful, and I think the community will benefit from this paper a lot.

Weakness: Primarily this paper can be deemed lack of experimental diversity: Simulations are instructive but limited in scope; broader tasks (e.g., toy tasks in NLP or vision) would strengthen the case. However, this is not crucial for the topic of this particular conference submission, which focuses on theoretical foundation for LLM prompting techniques.

---

> ### Author Rebuttal · Authors · 2025-07-30
>
> Thank you for your thorough and thoughtful review. We are glad you found the core idea of CoT information to be a novel and interesting contribution, with a clear and well-motivated presentation.
>
> **Question about estimating or approximating the CoT Information in practice**
>
> > *I thank the authors for their novel contribution. Cot information measure is central to the paper, but for much of the non-theory community, it’s unclear how one could estimate or approximate it in practice. If we have a dataset with cot annotations, is there a way to tell whether the cots are informative under your framework? Even a heuristic or proxy would help bridge the gap between the theory and practical applications.*
>
> Thank you for your kind words and for this important question. We have given this some thought and have a few preliminary ideas that we are interested in exploring further in future work.
>
> First, note that the CoT Information is defined in terms of the infimum of a particular negative log probability within some constraint set. Thus, one approach to estimate or approximate this quantity for specific model classes and tasks is to relate this to a constrained optimization problem, which can be approximated by optimizing the Lagrangian corresponding to the parametric model class. For example, suppose one has a parametric model class, like a neural network, $f_\theta: \mathcal{X} \to \Delta(\mathcal{Y} \times \mathcal{Z})$ parametrizing a probability distribution over CoT and output given the input. For example, $f_\theta$ could be a Transformer network that outputs a sequence composed of the CoT followed by the output. We may then seek to estimate the CoT information as the solution to the following optimization problem over the parameters of the model $\theta$:
> $$\min_\theta \max_{\lambda} -\frac{1}{n}\sum_{i} \log f_\theta(y_i, z_i | x_i) + \lambda \cdot (\epsilon - \frac{1}{n} \sum_i f^{e2e}_{\theta}(y_i | x_i))$$
>
> This optimization problem can be approximated via standard gradient-based optimization methods. Here, ${(x_i, y_i, z_i)}_{i}$ is a dataset that includes CoT annotations.
>
> We also note that, as per section 5 on information-theoretic lower bounds, the CoT information can also be related to a mutual information between the joint probability distributions over $(x, y, z)$ induced by different hypotheses. Thus, another approach would be to try to estimate this mutual information.

---

### Comment · Area_Chair_zAmz · 2025-08-01

Dear Reviewers,

The author-reviewer discussion phase has started. If you want to discuss with the authors about more concerns and questions, please post your thoughts by adding official comments as soon as possible.

Thanks for your efforts and contributions to NeurIPS 2025.

Best regards,

Your Area Chair

---

### Decision · Program_Chairs · 2025-09-17

**Decision:**

Accept (spotlight)

**Comment:**

Chain-of-thought concept is very important in the current foundation model's training and inference. In this paper, CoT information is proposed, and, based on this information, a CoT supervision theory is developed. The obtained results can reflect the current development of CoT in practice, and reviewers also admit the merits of this paper. Some concerns (CoT explanations or estimation) were proposed in the initial reviews, yet the authors addressed them during the rebuttal.

Given that this topic is very timely and important, I recommend accepting this paper.